# Testing water fluxes and storage from two hydrology configurations within the ORCHIDEE land surface model across US semi-arid sites

Natasha MacBean[1*], Russell L. Scott[2], Joel A. Biederman[2], Catherine Ottlé[3], Nicolas Vuichard[3], Agnès Ducharne[4], Thomas Kolb[5], Sabina Dore[6], Marcy Litvak[7], David J.P. Moore[8].

[1]Department of Geography, Indiana University, Bloomington, IN 47405, USA.
[2]Southwest Watershed Research Center, United States Agricultural Department, Agricultural Research Service, Tucson, AZ 85719, USA.
[3]Laboratoire des Sciences du Climat et de l'Environnement, LSCE/IPSL, CEA-CNRS-UVSQ, Université Paris-Saclay, Gif-sur-Yvette, F-91191, France.
[4]UMR METIS, Sorbonne Université, CNRS, EPHE, Paris, F-75005, France
[5]School of Forestry, Northern Arizona University, Flagstaff, AZ, 86011, USA.
[6]Hydrofocus, Inc., Davis, CA, 95618, USA.
[7]Department of Biology, University of New Mexico, Albuquerque, NM, 87131, USA.
[8]School of Natural Resources and the Environment, University of Arizona, Tucson, AZ, 85721, USA.

*Correspondence to*: Natasha MacBean (nlmacbean@gmail.com)

## Abstract

Plant activity in semi-arid ecosystems is largely controlled by pulses of precipitation, making them particularly vulnerable to increased aridity that is expected with climate change. Simple bucket-model hydrology schemes in land surface models (LSMs) have had limited ability in accurately capturing semi-arid water stores and fluxes. Recent, more complex, LSM hydrology models have not been widely evaluated against semi-arid ecosystem *in situ* data. We hypothesize that the failure of older LSM versions to represent evapotranspiration, ET, in arid lands is because simple bucket models do not capture realistic fluctuations in upper layer soil moisture. We therefore predict that including a discretized soil hydrology scheme based on a mechanistic description of moisture diffusion will result in an improvement in model ET when compared to data because the temporal variability of upper layer soil moisture content better corresponds to that of precipitation inputs. To test this prediction, we compared ORCHIDEE LSM simulations from 1) a simple conceptual 2-layer bucket scheme with fixed hydraulic parameters, and 2) a 11-layer discretized mechanistic scheme of moisture diffusion in unsaturated soil based on Richards equations, against daily and monthly soil moisture and ET observations, together with data-derived estimates of transpiration/evapotranspiration, T/ET, ratios, from six semi-arid grass, shrub and forest sites in the southwestern USA. The 11-layer scheme also has modified calculations of surface runoff, water limitation, and resistance to bare soil evaporation, E, to be compatible with the more complex hydrology configuration. To diagnose remaining discrepancies in the 11-layer model, we tested two further

configurations: i) the addition of a term that captures bare soil evaporation resistance to dry soil; and ii) reduced bare soil fractional vegetation cover. We found that the more mechanistic 11-layer model results in a better representation of the daily and monthly ET observations. We show that, as predicted, this is because of improved simulation of soil moisture in the upper layers of soil (top ~10cm). Some discrepancies between observed and modelled soil moisture and ET may allow us to prioritize future model development and the collection of additional data. Biases in winter and spring soil moisture at the forest sites could be explained by inaccurate soil moisture data during periods of soil freezing and/or underestimated snow forcing data. Although ET is generally well captured by the 11-layer model, modelled T/ET ratios were generally lower than estimated values across all sites, particularly during the monsoon season. Adding a soil resistance term generally decreased simulated bare soil evaporation, E, and increased soil moisture content, thus increasing transpiration, T, and reducing the negative bias between modelled and estimated monsoon T/ET ratios. This negative bias could also be accounted for at the low elevation sites by decreasing the model bare soil fraction, thus increasing the amount of transpiring leaf area. However, adding the bare soil resistance term and decreasing the bare soil fraction both degraded the model fit to ET observations. Furthermore, remaining discrepancies in the timing of the transition from minimum T/ET ratios during the hot, dry May-June period to high values at the start of the monsoon in July-August may also point towards incorrect modelling of leaf phenology and vegetation growth in response to monsoon rains. We conclude that a discretized soil hydrology scheme and associated developments improves estimates of ET by allowing the modelled upper layer soil moisture to more closely match the pulse precipitation dynamics of these semi-arid ecosystems; however, the partitioning of T from E is not solved by this modification alone.

## 1 Introduction

Semi-arid ecosystems – which cover ~40% of the Earth's terrestrial surface and which include rangelands, shrublands, grasslands, savannas, and seasonally dry forests – are in zones of transition between humid and arid climates and are characterized by sparse, patchy vegetation cover and limited water availability. Moisture availability in these ecosystems is therefore a major control on the complex interactions between vegetation dynamics and surface energy, water, and carbon exchange (Biederman et al., 2017; Haverd et al., 2016). Given the sensitivity to water availability, semi-arid ecosystem functioning may be particularly vulnerable to projected changes in climate (Tietjen et al., 2009; Maestre et al., 2012; Gremer et al., 2015). IPCC earth system model (ESM) projections and observation-based datasets indicate these regions will likely experience more intense warming and droughts, increases in extreme rainfall events, and a greater contrast between wet and dry seasons in the future (IPCC, 2013; Donat et al., 2016; Sippel et al., 2017; Huang et al., 2017).

To simulate the impact of climate change on semi-arid ecosystem functioning, it is essential that the land surface model (LSM) component of ESMs accurately represent semi-arid water flux and storage budgets (and all associated processes). In the last two to three decades, LSM groups have progressively updated their hydrology schemes from the more simplistic "bucket" type models included in earlier versions (Manabe, 1969). The resulting schemes typically include more physically-based

representations of vertical diffusion of water in unsaturated soils (Clark et al., 2015). In addition to increasing the complexity of soil hydrology, several studies have attempted to address the issue that models tend to miscalculate partitioning of evapotranspiration (ET) into transpiration (T) and bare soil evaporation (E), with models systematically underestimating T/ET ratios (Wei et al., 2017; Chang et al., 2018). One such mechanism that models have introduced is an evaporation resistance term that reduces the rate of water evaporation from bare soil surfaces (Swenson and Lawrence, 2014; Decker et al., 2017). The development of these more mechanistic soil hydrology schemes should mean that LSMs better capture high temporal frequency to seasonal and long-term temporal variability of water stores and fluxes. However, it is not always apparent that increasing model complexity provides more accurate representations of reality (as encapsulated by observations of different variables at multiple spatio-temporal scales). Further, increasing model complexity comes at a cost of increased computational resources and unknown parameters. Therefore, it is imperative that we test models of increasing complexity against multiple types of observations at a variety of sites representing different ecosystem types.

New generation LSM water flux and storage estimates have been extensively tested at multiple scales from the site level to the globe (Abramowitz et al., 2008; Dirmeyer et al., 2011; Guimberteau et al., 2014; Mueller et al., 2014; Best et al., 2015; Ukkola et al., 2016b; Raoult et al., 2018; Scanlon et al., 2018, 2019). Model-data biases are observed across all biomes; however, a key finding common to these studies is that models do not capture seasonal to inter-annual water stores and fluxes well during dry periods and/or at drier sites (Mueller et al., 2014; Swenson et al., 2014; De Kauwe et al., 2015; Best et al., 2015; Ukkola et al., 2016a; Humphrey et al., 2018; Scanlon et al., 2019). Mueller et al. (2014) showed that CMIP5 models overestimated multiyear mean daily ET in many regions, with the strongest bias in dryland regions (particularly western North America). Likewise, Grippa et al. (2011) and Scanlon et al. (2019) demonstrated that LSMs underestimate seasonal amplitude of total water storage in semi-arid (and tropical) regions. However, compared to more mesic ecosystems, semi-arid ecosystem LSM water flux and storage simulations have rarely been tested extensively against *in situ* observations, apart from a few exceptions (Hogue et al., 2005; Abramowitz et al., 2008; Whitley et al., 2016; Grippa et al., 2017). Whitley et al. (2016) compared carbon and water flux simulations from six LSMs at five OzFlux savanna sites. Their study highlighted two key deficiencies in modeling water fluxes: i) modeled C4 grass T is too low; and ii) models with shallow rooting depths typically underestimate woody plant dry season ET. As part of a model inter-comparison for West Africa (the AMMA LSM Intercomparison Project – ALMIP), LSM water storage, fluxes, runoff, and land surface temperature were evaluated against *in situ* and remote sensing data in the Malian Gourma region of the central Sahel (Boone et al., 2009; De Kauwe et al., 2013; Lohou et al., 2014; Grippa et al., 2011; Grippa et al., 2017). These studies highlight that temporal characteristics of water storage and fluxes in this monsoon-driven semi-arid region are captured fairly well by models; however, the studies also point to various model issues, including: difficulties in simulating bare soil evaporation response to rainfall events (Lohou et al., 2014); underestimation of dry season ET (Grippa et al., 2011); the need for greater water and energy exchange sensitivity to different vegetation types and soil characteristics (De Kauwe et al., 2013; Lohou et al., 2014; Grippa et at., 2017); and overestimation of surface runoff (Grippa et al., 2017). How models prescribe or predict leaf area index (LAI) has also been highlighted as a driver of hydrological model-data differences (Ha et al., 2015; Grippa et al., 2017).

The aim of this study was to contribute a new LSM hydrology model evaluation in a semi-arid region not previously investigated: the monsoon-driven semi-arid southwestern United States (hereafter, the SW US). The density and diversity of research sites in the SW US provides a rare opportunity to test an LSM across a range of semi-arid ecosystems. The semi-arid SW US has also been identified as one of the key regions of global land-atmosphere coupling (Koster et al., 2004) and the most persistent climate change hotspot in the US (Diffenbaugh et al., 2008; Allen et al., 2016). Expected future soil moisture

deficits in this region will result in strong atmospheric feedbacks, with consequent high temperature increases (Senerivatne et al., 2013) and a potential weakening of the terrestrial biosphere C sink (Berg et al., 2016; Green et al., 2019). Several studies based on model predictions, instrumental records, and paleoclimatic data analyses have suggested that over the coming century the risk of more severe, multi-decadal drought in the SW US will increase considerably (Ault et al., 2014, 2016; Cook et al., 2015). In fact, models suggest that a transition to drier conditions is already underway (Seager et al., 2007; Archer and Predick,

2008; Seager and Vecchi, 2010). Investigating how well LSMs capture hydrological stores and fluxes in this region therefore provides a crucial test for how well models can produce accurate global climate change projections.

    Here, we tested the ability of the ORCHIDEE (ORganizing Carbon and Hydrology in Dynamic EcosystEms) LSM to simulate multiple water flux and storage related variables at six SW US semi-arid Ameriflux eddy covariance sites spanning forest, shrub- and grass-dominated ecosystems (Biederman et al., 2017). We tested two versions of the ORCHIDEE LSM with

115 hydrological schemes of differing complexity: 1) a simple 2-layer conceptual bucket scheme (hereafter, 2LAY) with constant water holding capacity (de Rosnay and Polcher 1998); and 2) a 11-layer mechanistic scheme (hereafter, 11LAY) based on the Richards equation with hydraulic parameters based on soil texture (de Rosnay et al., 2002). Besides the change in the soil hydrology between the 2LAY and 11LAY versions, several other hydrology-related processes have also been modified due to increases in the complexity of the model. These modifications are described further in Section 2.2 and summarized in Table

2. The 2LAY scheme was used in the previous CMIP5 runs, whereas the 11LAY scheme is the default scheme in the current version of ORCHIDEE that is used in the ongoing Coupled Model Intercomparison Project (CMIP6) simulations (Ducharne et al., in prep).

    Our analyses were organized as follows: First, we evaluated how changing from the conceptual 2LAY bucket model to the physically-based 11LAY soil hydrology scheme – and all associated modifications – has influenced the high temporal

frequency and seasonal variability of semi-arid ecosystem soil moisture, ET (and its component fluxes), runoff, drainage, and snow mass/melt. Although there have been many previous studies comparing simple bucket schemes versus mechanistic multi-layer hydrology, we include such a comparison in the first part of our analysis for the following reasons: a) the simple bucket schemes were the default hydrology in some CMIP5 model simulations and these simulations are still being widely used to understand ecosystem responses to changes in climate; b) variations on the simple bucket schemes are still implemented by

design in various types of hydrological models (Bierkens et al., 2015); c) there has not yet been extensive comparisons of these two types of hydrology model for semi-arid regions, and especially not for the SW US; and d) so that the 2LAY can serve as a benchmark for the 11LAY scheme. Second, we evaluated the temporal dynamics of the 11LAY model against observations at three specific soil depths (shallow: ≤ 5cm; mid: 15-20cm; deep: ≥ 30cm) to assess whether the physically-based discretized

scheme accurately captures moisture transport down the soil profile. Note that when evaluating the 11LAY model soil moisture against observations, our primary focus was on the temporal dynamics – rather than the absolute magnitude – given the difficulty of comparing absolute values of volumetric water content between the models and the data (see Section 2.3.2 for more detail). Therefore, in the model-data comparison, we scale the observations to the 11LAY model simulations via linear CDF matching. Finally, having evaluated the standard (default) 11LAY model against *in situ* semi-arid water stores and fluxes, a novel component of our study was to investigate whether some of the site-scale semi-arid LSM hydrology model discrepancies outlined above (e.g. underestimation of C4 grass T, weak dry season ET and therefore low T/ET ratios, ET issues related to incorrect representation of leaf area, and overestimation of surface runoff) are improved with recent ORCHIDEE hydrology model developments. Where the model does not capture observed patterns, we investigated which model processes or mechanisms in the 11LAY scheme might be responsible for remaining model-data discrepancies. In particular, we assessed the impact of a) decreasing the bare soil fraction (thus, increasing leaf area); and b) including the optional bare soil resistance term into the 11LAY scheme (Ducharne et al., in prep.). Given the sparsely-vegetated nature of the low-elevation semi-arid grass- and shrub-dominated sites in our study we hypothesized that inclusion of this term may counter any dry season ET underestimate. Throughout, we explored if there are any discernible differences across sites due to elevation and vegetation composition.

Section 2 describes the sites, data, model and methods used in this study; Section 3 details the results of the two-part model evaluation (as outlined above); and Section 4 discusses how future studies may resolve remaining model issues in order to improve LSM hydrology modeling in semi-arid regions.

## 2 Methods and Data

### 2.1 Southwestern US study sites

We used six semi-arid sites in the SW US that spanned a range of vegetation types and elevations (Biederman et al. 2017). The entire SW US is within the North American Monsoon region; therefore, these sites typically experience monsoon rainfall during July to October, preceded by a hot, dry period in May and June. Table 1 describes the dominant vegetation, species and soil texture characteristics at each site, together with the observation period. The four grass- and shrub-dominated sites (US-SRG, US-SRM, US-Whs and US-Wkg) are located at low-elevation (<1600m) in southern Arizona with mean annual temperatures between 16 and 18°C (Biederman et al., 2017). These four sites are split into pairs of grass- and shrub-dominated systems: US-SRG (C4 grassland site) and US-SRM (mesquite-dominated site) are located at the Santa Rita Experimental Range ~60km south of Tucson, AZ, whilst US-Whs (creosote shrub-dominated site) and US-Wkg (C4 grassland site) are located at the Walnut Gulch Experimental Watershed ~120km to the southeast of Tucson, AZ. Moisture availability at these low elevation sites is predominantly driven by summer monsoon precipitation; however, winter and spring rains also contribute to the bi-modal growing seasons at these sites (Scott et al., 2015; Biederman et al., 2017). The US-Fuf (Flagstaff Unmanaged Forest) and US-Vcp (Valles Caldera Ponderosa) sites are at higher elevations (2215m and 2501m). Both high elevation sites

experience cooler mean annual temperatures of 7.1 and 5.7°C, respectively, and are dominated by ponderosa pine (Anderson-Teixiera et al., 2010; Dore et al., 2012). The high elevation forested sites have two annual growing seasons with available moisture coming both from heavy winter snowfall (and subsequent spring snow melt) and summer monsoon storms. US-Fuf is located near the town of Flagstaff in northern AZ, whilst US-Vcp is located in the Valles Caldera National Preserve in the Jemez Mountains in north-central New Mexico. Groundwater depths across all sites are typically 10s to 100s metres. Flux tower instruments at all six sites collect half-hourly measurements of meteorological forcing data and eddy covariance measurements of net surface energy and carbon exchanges (see Section 2.3.1).

## 2.2 ORCHIDEE Land Surface Model

### 2.2.1 General model description

The ORCHIDEE LSM forms the terrestrial component of the French IPSL ESM (Dufresne et al., 2013), which contributes climate projections to IPCC Assessment Reports. ORCHIDEE has undergone significant modification since the 'AR5' version (Krinner et al., 2005), which was used to run the CMIP5 (Coupled Model Inter-comparison Project) simulations included in the IPCC 5th Assessment Report (IPCC, 2013). The model code is written in Fortran 90. Here, we use ORCHIDEE v2.0 that is used in the ongoing CMIP6 simulations. The hydrology scheme in ORCHIDEE v2.0 is described in detail in Ducharne et al. (in prep.). ORCHIDEE simulates fluxes of carbon, water and energy between the atmosphere and land surface (and within the sub-surface) on a half-hourly time step. In uncoupled mode, the model is forced with climatological fields derived either from climate reanalyses or site-based meteorological forcing data. The required climate fields are: 2m air temperature, rainfall and snowfall, incoming long and shortwave radiation, wind speed, surface air pressure, and specific humidity.

Evapotranspiration, ET, in the model is calculated as the sum of four components: 1) evaporation from bare soil, E; 2) evaporation from water intercepted by the canopy; 3) transpiration, T, (controlled by stomatal conductance); and 4) snow sublimation (Guimberteau et al., 2012b). There are two soil hydrology models implemented in ORCHIDEE: one based on a 2-layer (2LAY) conceptual model, the other on a physically based representation of moisture redistribution across 11-layers (11LAY). In this study, the soil depth for both schemes was set to 2m based on previous studies that tested the implementation of the soil hydrology schemes (de Rosnay and Polcher 1998; de Rosnay et al., 2000; de Rosnay et al., 2002). Further modifications to the model have been made since the implementation of the 11LAY scheme to augment the increased complexity:  In the 2LAY scheme, runoff occurred when the soil reached saturation; whereas in the 11LAY scheme, surface infiltration, runoff and drainage are treated more mechanistically based on soil hydraulic conductivity (see Section 2.2.2). In the 2LAY scheme, there was an implicit resistance to bare soil evaporation based on the depth of the dry soil for the bare soil plant functional type (PFT). In the 11LAY scheme, there is an optional bare soil evaporation resistance term based on the relative soil water content of the first four soil layers, based on the formulation of Sellers et al. (1992) – (see Section 2.2.3). Both resistance terms aim to describe the resistance to evaporation exerted by a dry mulch soil layer. Similarly, the calculation

of moisture limitation on stomatal conductance has changed. In the 2LAY version, moisture limitation depended on the dry soil depth of the upper layer; whereas in the 11LAY version, the limitation is based on plant water availability for root water

uptake throughout the soil column. Finally, in the 2LAY scheme there is no E from the vegetated portion of the grid cell (only T); whereas, in the 11LAY scheme both E and T occur (see Section 2.2.1).The main differences between the two ORCHIDEE configurations used in this study are described in the sections below and are summarised in Table 2.

In ORCHIDEE, a prognostic leaf area is calculated based on phenology schemes originally described in Botta et al. (2000) and further detailed in MacBean et al. (2015 – Appendix A). The albedo is calculated based on the average of the defined

albedo coefficients for vegetation (one coefficient per PFT), soil (one value for each grid cell, referred to as background albedo) and snow weighted by their fractional cover. Snow albedo is also parameterized according to its age, which varies according to the underlying PFT. The albedo coefficients for each PFT and background albedo have recently been optimized within a Bayesian inversion system using the visible and near infrared MODIS white sky albedo product at 0.5x0.5° resolution for years 2000-2010 (Bastrikov et al., in prep). The prior background (bare soil) albedo values were retrieved from MODIS data

using the EU Joint Research Center Two Stream Inversion Package (JRC-TIP).

As in most LSMs, all vegetation is grouped into broad plant functional types (PFTs) based on physiology, phenology, and, for trees, the biome in which they are located. In ORCHIDEE, by default, there are 12 vegetated PFTs plus a bare soil PFT. The 13 PFT fractions are defined for each grid cell (or for a given site, as in this study) in the initial model set-up and sum to 1.0 (unless there is also a "no bio" fraction for bare rock, ice, and urban areas). Independent water budgets are calculated for each

215 "soil tile", which represent separate water columns within a grid cell. In the 2LAY scheme, soil tiles directly correspond to PFTs; therefore, a separate water budget is calculated for each PFT within the grid cell. In the 11-layer scheme there are three soil tiles: one with all tree PFTs sharing the same soil water column, one soil column with all the grass and crop PFTs, and a third for the bare soil PFT. Therefore, three separate water budgets are calculated: one for the forested soil tile, one for the grass and crop soil tile, and one for the bare soil PFT tile (Ducharne et al., in prep. See sections 2.2.2 to 2.2.5 for details on the

220 hydrology calculations). In the 2-layer scheme there is no E from the vegetated tiles (only transpiration). In the 11-layer scheme, both T and E occur in the vegetated (forest and grass/crop) soil tiles. T occurs for each PFT in the "effective" vegetated sub-fraction of each soil tile, which increases as LAI increases, whereas E occurs at low LAI (e.g. during winter) over the effective bare soil sub-fraction of each soil tile. Note that the bare soil sub-fraction of each vegetated soil tile is separate from the bare soil PFT tile itself. The effective vegetated sub-fraction is calculated using the following equation that describes

attenuation of light penetration through a canopy $f_v^j = f^j \left(1 - e^{(-k_{ext} LAI_j)}\right)$, where $f^j$ is the fraction of the grid cell covered by PFT $j$ (i.e. the unattenuated case), $f_v^j$ is the fraction of the effective sub-fraction of the grid cell covered by PFT $j$, and $k_{ext}$ is the extinction coefficient and is set to 1.0. The effective bare soil sub-fraction of each vegetated soil tile, $f_b^j$, is equal to $1 - f_v^j$. The total grid cell water budget is calculated by vegetation fraction weighted averaging across all soil tiles (Guimberteau et al., 2014; Ducharne et al., in prep.). Soil texture classes and related parameters are prescribed based on the

percentage of sand, clay and loam.

## 2.2.2 Soil Hydrology

*2-layer conceptual soil hydrology model*

In the 'AR5' version of ORCHIDEE used in the CMIP5 experiments, the soil hydrology scheme consisted of a conceptual 2-layer (2LAY) so-called "bucket" model based on Choisnel et al. (1995). The depth of the upper layer is variable up to 10 cm and changes with time depending on the balance between throughfall and snowmelt inputs, and outputs via three pathways: i) bare soil evaporation, limited by a soil resistance increasing with the dryness of the topmost soil layer; ii) root water extraction for transpiration, withdrawn from both layers proportionally to the root density profile; and iii) downward water flow (drainage) to the lower layer. If all moisture is evaporated or transpired, or if the entire soil saturates, the top layer can disappear entirely. Three empirical parameters govern the calculation of the drainage between the two layers, which depends on the water content of the upper layer and takes a non-linear form, so drainage from the upper layer increases considerably when the water content of the upper layer exceeds 75% of the maximum capacity (Ducharne et al., 1998). Transpiration is also withdrawn from the lower layer via water uptake by deep roots. Finally, runoff only occurs when the total soil water content exceeds the maximum field capacity, set to 150 kg.m$^{-2}$ as in Manabe (1969). It is then arbitrarily partitioned into 5% surface runoff to feed the overland flow and 95% drainage to feed the groundwater flow of the routing scheme (Guimberteau et al., 2012b), which is not activated here.

*11-layer mechanistic soil hydrology model*

The 11LAY scheme was initially proposed by de Rosnay et al. (2002) and simulates vertical flow and retention of water in unsaturated soils based on a physical description of moisture diffusion (Richards, 1931). The scheme implemented in ORCHIDEE relies on the one-dimensional Richards equation, combining the mass and momentum conservation equations, but is in its saturation form that uses volumetric soil water content $\theta$ (m$^3$m$^{-3}$) as a state variable instead of pressure head (Ducharne et al., in prep). The two main hydraulic parameters (hydraulic conductivity and diffusivity) depend on volumetric soil moisture content defined by the Mualem–van Genuchten model (Mualem, 1976; van Genuchten, 1980). The Richards equation is solved numerically using a finite-difference method, which requires the vertical discretization of the 2m soil column. As described by de Rosnay et al. (2002), 11 layers are defined: the top layer is ~0.1mm thick and the thickness of each layer increases geometrically with depth. The fine vertical resolution near the surface aims at capturing strong vertical soil moisture gradients in response to high temporal frequency (sub-diurnal to few days) changes in precipitation or ET. De Rosnay et al. (2000) tested a number of different vertical soil discretization and decided that 11 layers was a good compromise between computational cost and accuracy in simulating vertical hydraulic gradients. The mechanistic representation of redistribution of moisture within the soil column also permits capillary rise, and a more mechanistic representation of surface runoff. The calculated soil hydraulic conductivity determines how much precipitation is partitioned between soil infiltration and runoff (d'Orgeval et al., 2008). Drainage is computed as free gravitational flow at the bottom of the soil (Guimberteau et

al., 2014). The USDA soil texture classification, provided at 1/12-degree resolution by Reynolds et al. (2000), is combined with the look-up pedotransfer function tables of Carsel and Parrish (1988) to derive the required soil hydrodynamic properties (saturated hydraulic conductivity Ks, porosity, Van Genuchten parameters, residual moisture), while field capacity and wilting point are deduced from the soil hydrodynamic properties listed above and the Van Genuchten equation for matric potential, by assuming they correspond to potentials of -3.3m and -150m respectively (Ducharne et al., in prep). Ks increases exponentially with depth near the surface to account for increased soil porosity due to bioturbation by roots, and decreases exponentially with depth below 30cm to account for soil compaction (Ducharne et al., in prep).

The 11LAY soil hydrology scheme has been implemented in the ORCHIDEE trunk since 2010, albeit with various modifications since that time, as described above and in the following sections. The most up-to-date version of the model is described in Ducharne et al. (in prep). Similar versions of the 11LAY scheme have been tested against a variety of hydrology-related observations in the Amazon Basin (Guimberteau et al., 2012a; Guimberteau et al., 2014), for predicting future changes in extreme runoff events (Guimberteau et al., 2013) and against a water storage and energy flux estimates as part of ALMIP in West Africa (as detailed in Section 1 – d'Orgeval et al., 2008; Boone et al., 2009; Grippa et al., 2011; Grippa et al., 2017).

### 2.2.3 Bare soil evaporation and additional resistance term

The computation of bare soil evaporation, E, in both versions is implicitly based on a supply and demand scheme. E occurs from the bare soil column as well as the bare soil fraction of the other soil tiles (see section 2.2.1) In the 2LAY version, E decreases when the upper layer gets drier, owing to a resistance term that depends on the height of the dry soil in the bare soil PFT column (Ducoudré et al., 1993). In the 11LAY version, E, proceeds at the potential rate $E_{pot}$ unless the water supply via upward diffusion from the water column is limiting, in which case E is reduced to correspond to the situation in which the soil moisture of the upper 4 layers is at wilting point. However, since ORCHIDEE v2.0 (Ducharne et al., in prep.), E can also be reduced by including an optional bare soil evaporation resistance term, $r_{soil}$, which depends on the relative water content and is based on a parameterization fitted at the FIFE grassland experimental site at Konza Prairie Field Station in Kansas (Sellers et al., 1992):

$$r_{soil} = \exp(8.206 - 4.255\, W_1) \tag{1}$$

where $W_1$ is the relative soil water content of the first four layers (2.2cm – Table S1). $W_1$ is calculated by dividing the mean soil moisture across these layers by the saturated water content. The calculation for E then becomes:

$$E = \min(E_{pot}/(1 + r_{soil}/r_a), Q) \tag{2}$$

where $E_{pot}$ is the potential evaporation, $r_a$ the aerodynamic resistance, $Q$ the upward water supply from capillary diffusion through the soil, and $r_{soil}$ the soil resistance to this upward exfiltration. In all simulations, the calculation of $r_a$ includes a dynamic roughness height with variable LAI, based on a parameterization by Su et al. (2001). By default, in the 11LAY version there is no resistance ($r_{soil} = 0$). Note that there is no representation of below canopy E in this version of ORCHIDEE given

there is no multi-layer energy budget for the canopy. Note also that the same roughness is used for both the effective bare ground and vegetated fractions.

### 2.2.4 Empirical plant water stress function, $\beta$

The soil moisture control on transpiration is defined by an empirical water stress function, $\beta$. Whichever the soil hydrology model, $\beta$ depends on soil moisture and on the root density profile $R(z) = \exp(-c_j z)$, where $z$ is the soil depth and $c_j$ (in m$^{-1}$) is the root density decay factor for PFT $j$. In both model versions for a 2m soil profile, $c_j$ is set to 4.0 for grasses, 1.0 for temperate needleleaved trees and 0.8 for temperate broadleaved trees. In 11 LAY, a related variable is $n_{root}(i)$, quantifying the mean relative root density $R(z)$ of each soil layer $i$, so that $\sum n_{root}(i) = 1$.

In the 2LAY version, $\beta$ is calculated as an exponential function of the root decay factor $c_j$ and the dry soil height of the topmost soil layer ($h_t^d$):

$$\beta = exp^{(-c_j, h_t^d)} \tag{3}$$

In the 11LAY, $\beta$ is rather based on the available moisture across the entire soil moisture profile and is calculated for each PFT $j$ and soil layer $i$, and then summed across all soil layers (starting at the 2$^{nd}$ layer given no water stress in the 1$^{st}$ layer – a

conservative condition that prevents transpiration, T, from inducing a negative soil moisture from this very thin soil layer):

$$\beta(j) = \sum_{i=2}^{11} n_{root}(i) \max\left(0, \min\left(1, \max\left(0, \frac{(W_{i,v} - W_{wpt})}{(W_\% - W_{wpt})}\right)\right)\right) \tag{4}$$

where $W_i$ is the soil moisture for that layer and soil tile in kgm$^{-2}$, $W_{wpt}$ is the wilting point soil moisture, and $W_\%$ is the threshold above which T is maximum – i.e. above this threshold T is not limited by $\beta$. $W_\%$ is defined by:

$$W_\% = W_{wpt} + p_\%(W_{fc} - W_{wpt}) \tag{5}$$

Where $W_{fc}$ is the field capacity and $p_\%$ defines the threshold above which T is maximum. $p_\%$ is set to 0.8 and is constant for all PFTs. This empirical water stress function equation means that, in the 11LAY, $\beta$ varies linearly between 0 at the wilting point to 1 at $W_\%$, which is smaller or equal to the field capacity. LSMs typically apply $\beta$ to limit photosynthesis ($A$) via the maximum carboxylation capacity parameter $V_{cmax}$, or to the stomatal conductance, $g_s$, via the g0 or g1 parameters of the $A/gs$ relationship, or both (De Kauwe et al., 2013; 2015). In ORCHIDEE there is the option of applying $\beta$ to limit either $V_{cmax}$ or $g_s$, or both. In

the default configuration used in CMIP6, $\beta$ is applied to both (based on results from Keenan et al., 2010; Zhou et al., 2013; Zhou et al., 2014); therefore, this is the configuration we used in this study.

### 2.2.5 Snow scheme

ORCHIDEE contains a multi-layer intermediate complexity snow scheme that is described in detail in Wang et al. (2013). The

new scheme was introduced to overcome limitations of a single layer snow configuration. In a single layer scheme, the

temperature and vertical density gradients through the snowpack, which affect the sensible, latent and radiative energy fluxes, are not calculated. The single layer snow scheme does not describe the insulating effect of the snow pack, nor the links between snow density and changes in snow albedo (due to aging) in a physically mechanistic way. In the new explicit snow scheme, there are three layers that each have a specific thickness, density, temperature and liquid water and heat content. These variables are updated at each time step based on the snowfall and incoming surface energy fluxes, which are calculated from the surface energy balance equation. The model also accounts for sublimation, snow settling, water percolation and refreezing. Snow mass cannot exceed a threshold of 3000 $kg.m^{-2}$. Snow age is also calculated and is used to modify the snow albedo. Default snow albedo coefficients have been optimized using MODIS white sky albedo data per the method described in Section 2.2.1. Snow fraction is calculated at each time step according to snow mass and density following the parametrization proposed by Niu and Yang (2007).

## 2.3 Data

### 2.3.1 Site-level meteorological and eddy covariance data and processing

Meteorological forcing and eddy covariance flux data for each site were downloaded from the AmeriFlux data portal (http://ameriflux.lbl.gov). Meteorological forcing data included 2m air temperature and surface pressure, precipitation, incoming long and shortwave radiation, wind speed, and specific humidity. To run the ORCHIDEE model, we partitioned the *in situ* precipitation into rainfall and snowfall using a temperature threshold of 0°C. The meteorological forcing data were gap-filled following the approach of Vuichard and Papale (2015), which uses downscaled and corrected ERA-Interim data to fill gaps in the site-level data. Eddy covariance flux data were processed to provide ET from estimates of latent energy fluxes. ET gaps were filled using a modified look-up table approach based on Falge et al. (2001), with ET predicted from meteorological conditions within a 5-day moving window. Previous comparisons of annual sums of measured ET with site-level water balance measurements at a few of these sites show an average agreement within 3% of each other, but could differ by -10 to +17% in any given year (Scott and Biederman, 2019). Estimates of T/ET ratios were derived from Zhou et al. (2016) for the forested sites, and both Zhou et al. (2016) and Scott and Biederman (2017) for the more water-limited low elevation grass- and shrub-dominated sites. Zhou et al. (2016) (hereafter Z16) used eddy covariance tower gross primary productivity (GPP), ET and vapor pressure deficit (VPD) data to estimate T/ET ratios based on the ratio of the actual or apparent underlying water use efficiency (uWUE$_a$) to the potential uWUE (uWUE$_p$). uWUE$_a$ is calculated based on a linear regression between ET and GPP multiplied by VPD to the power 0.5 (GPPxVPD$^{0.5}$) at observation timescales for a given site, whereas uWUE$_p$ was calculated based on a quantile regression between ET and GPPxVPD$^{0.5}$ using all the half-hourly data for a given site. Scott and Biederman (2017) (hereafter SB17) developed a new method to estimate average monthly T/ET from eddy covariance data that was more specifically designed for the most water-limited sites. The SB17 method is based on a linear regression between monthly GPP

and ET across all site years. One of the main differences between the Z16 and SB17 method is that the regression between GPP and ET is not forced through the origin in SB17 because at water-limited sites it is often the case that ET ≠ 0 when GPP = zero (Biederman et al., 2016). The Z16 method also assumes the uWUE$_p$ is when T/ET = 1, which rarely occurs in water-limited environments (Scott and Biederman, 2017). In this study, T/ET ratio estimates are omitted in certain winter months when very low GPP and limited variability in GPP results in poor regression relationships.

### 2.3.2 Soil moisture data and processing

Daily mean volumetric soil moisture content (SWC, $\theta$, m$^3$m$^{-3}$) measurements at several depths were obtained directly from the site PIs. For each site, Table 3 details the depths at which soil moisture was measured. Soil moisture measurement uncertainty is highly site and instrument specific, but tests have shown that average errors are generally below 0.04 m$^3$ m$^{-3}$ if site specific calibrations are made. Given the maximum depth of the soil moisture measurements is 75cm (and is much shallower at some sites) we cannot use these measurements to estimate a total 2m soil column volumetric SWC. Instead, we only used these measurements to evaluate the 11LAY model (and the 2LAY upper layer soil moisture – calculated for 0-10cm) because, unlike the 2LAY model, with the 11LAY version of the model we have model estimates of soil moisture at discrete soil depths. However, several factors mean that we cannot directly compare absolute values of measured versus modelled soil SWC, even though the 11LAY has discrete depths. First, site-specific values for soil saturated and residual water content were generally not available to parameterize the model (see Section 2.4); instead, these soil hydrology parameters are either fixed (in the 2LAY) or derived from prescribed soil texture properties (in the 11LAY – see Section 2.2.2). Therefore, we may expect a bias between the modelled and observed daily mean volumetric SWC. Second, while the soil moisture measurements are made with probes at specific depths, it is not precisely known over which depth ranges they are measuring SWC. Therefore, with the exception of Fig. 1 in which we examine changes in total water content between the two model versions, for the remaining analyses we do not focus on absolute soil moisture values in the model–data comparison. Instead, we focus solely on comparison between the modelled and observed soil moisture temporal dynamics. To achieve this, we removed any model-data bias using a linear cumulative density function (CDF) matching function to re-scale and match the mean and standard deviation of soil moisture simulations to that of the observations for each layer where soil moisture is measured using the following equation:

$$\theta_{Mod,CDF} = \frac{\sigma_{\theta,Obs}(\theta_{Mod}-\bar{\theta}_{Mod})}{\sigma_{\theta,Mod}} + \bar{\theta}_{Obs} \tag{6}$$

Raoult et al. (2018) found that linear CDF matching performed nearly as well as full CDF matching in capturing the main features of the soil moisture distributions; therefore, for this study we chose to simply use a linear CDF re-scaling function. Note that while we do compare the re-scaled 2LAY upper layer soil moisture (top 10cm) and 11LAY simulations at certain depths to the observations (see Section 3.1), we cannot compare the total column soil moisture given our observations do not go down to the same depth as the model (2m). Also note that because of the reasons given above, we chose to focus most of

the model-data comparison on investigating how well the (re-scaled) 11LAY model captures the observed temporal dynamics at specific soil depths (see Section 3.2).

## 2.4 Simulation set-up and post-processing

All simulations were run for the period of available site data (including meteorological forcing and eddy covariance flux data
– see Section 2.3.1 and Table 1). Table 1 also lists: i) the main species for each site and the fractional cover of each model PFT that corresponds to those species; iii) the maximum LAI for each PFT; and iii) the percent of each model soil texture class that corresponds to descriptions of soil characteristics for each site – all of which were derived from associated site literature detailed in the references in Table 1. The PFT fractional cover and the fraction of each soil texture class are defined in ORCHIDEE by the user. The maximum LAI has a default setting in ORCHIDEE that has not been used here; instead, values
based on the site literature were prescribed in the model (Table 1). Note that ORCHIDEE does not contain a PFT that specifically corresponds to shrub vegetation; therefore, the shrub cover fraction was prescribed to the forested PFTs (see Table 1). Due to the lack of available data on site-specific soil hydraulic parameters across the sites studied, we chose to use the default model values that were derived based on pedotransfer functions linking hydraulic parameters to prescribed soil texture properties (see Section 2.2.2). Using the default model parameters also allows us to test the default behavior of the model.

At each site we ran five versions of the model: 1) 2LAY soil hydrology; 2) 11LAY soil hydrology with $r_{soil}$ flag not set (default model configuration); 3) 11LAY soil hydrology with $r_{soil}$ flag not set and with reduced bare soil fraction (increased C4 grass cover); 4) 11LAY soil hydrology with the $r_{soil}$ flag set (therefore, Eqn. 2 activated); and 5) 11LAY soil hydrology with the $r_{soil}$ flag set and with reduced bare soil fraction. Tests 3 and 5 (reduced bare soil fraction) are designed to account for the fact that grass cover is highly dynamic at intra-annual timescales at the low-elevation sites and therefore during certain seasons (e.g.
the monsoon) the grass cover will likely be higher than was prescribed in the model based on average fractional cover values given in the site literature. The C4 grass cover was therefore increased to the maximum observed C4 grass cover under the most productive conditions (100% cover for the Santa Rita sites, and 80% cover for the Walnut Gulch sites). A 400-year spinup was performed by cycling over the gap-filled forcing data for each site (see Table 1 for period of available site data) to ensure the water stores were at equilibrium. Following the spinup, transient simulations were run using the forcing data from each
site. Daily outputs of all hydrological variables (soil moisture, ET and its component fluxes, snow pack, snow melt), the empirical water stress function, $\beta$, LAI, and soil temperature were saved for all years and summed or averaged to derive monthly values, where needed. For certain figures we show the 2009 daily time series because that was the only year for which data from all sites overlapped and a complete year of daily soil moisture observations was available. To evaluate the two model configurations, we calculated the Pearson correlation coefficient between the simulated and observed daily time series for both
the upper layer soil moisture (with the model re-scaled according to the linear CDF matching method given in Section 2.3.2) and ET. We also calculated the RMSE, mean absolute bias, and a measure of the relative variability, $\alpha$, between the modelled

and observed daily ET. The latter is calculated as the ratio of model to observed standard deviations ($\alpha = \frac{\sigma_m}{\sigma_o}$) based on Gupta et al. (2009). All model post-processing and plotting was performed using the Python programming language (v2.7.15) (Python Software Foundation – available at http://www.python.org), the NumPy (v1.16.1) (Harris et al., 2020) numerical analysis package, and Matplotlib (v2.0.2) (Hunter et al., 2007) and Seaborn (v0.9.0) (Waskom et al., 2017) plotting and data visualization libraries.

## 3 Results

### 3.1 Differences between the 2LAY and 11LAY model versions for main hydrological stores and fluxes

Increasing the soil hydrology model complexity between the 2LAY and 11LAY model versions does not result in a uniform increase or decrease across sites in either the simulated upper layer (top 10cm) and total column (2m) soil moisture (kgm$^{-2}$) (Fig. 1 2$^{nd}$ and 3$^{rd}$ panel; also see Fig. S1 for complete daily time series for each site). The largest change between the 2LAY and 11LAY versions in the upper layer soil moisture were seen at the high-elevation ponderosa forest sites (US-Fuf and US-Vcp – Fig. 1 and Figs. S1 a and b). In the 2LAY simulations, the upper layer soil moisture is similar across all sites; whereas, in the 11LAY simulations the difference between the high elevation forest sites and low elevation grass and shrub sites has increased. At US-Fuf, both the upper layer and total column soil moisture increase in the 11LAY simulations compared to the 2LAY, which corresponds to an increase in mean daily ET (Fig. 1 top panel) away from the observed mean, and a decrease in total runoff (surface runoff plus drainage – Fig. 1 bottom panel). In contrast, at US-Vcp while there is an increase in the upper layer soil moisture there is hardly any change in the total column soil moisture. The higher upper layer soil moisture at US-Vcp causes a slight increase in mean ET (and ET variability) that better matches the observed mean daily ET, and a decrease in total runoff. Note that changes in maximum soil water holding capacity are due to how soil hydrology parameters are defined. In the 2LAY, a maximum capacity is set to 150kgm$^{-2}$ across all PFTs; whereas in the 11LAY, the capacity is based on soil texture properties and is therefore different for each site.

At the low-elevation shrub and grass sites (US-SRM, US-SRG, US-Whs, and US-Wkg) the differences between the two model versions for both the upper layer and total column soil moisture are much smaller (Fig. 1). Correspondingly, the changes in mean daily ET and total runoff are also marginal (although the mean total runoff is lower at Walnut Gulch: US-Wkg and US-Whs). Across all sites both model versions accurately capture the overall mean daily ET (Fig. 1). At Santa Rita (US-SRM and US-SRG), the 11LAY soil moisture is marginally lower than the 2LAY, whereas at the Walnut Gulch sites the 11LAY moisture is higher.

As described above, at all sites there is either no change between the 2LAY and 11LAY simulations (Santa Rita) or a decrease in total runoff (surface runoff plus drainage – Fig. 1 bottom panel). Across all sites, excess water is removed as drainage in the 2LAY simulations, with little to no runoff (Figs. S1 a-f 3$^{rd}$ panel); whereas in the 11LAY simulations excess water flows

mostly as surface runoff, with more limited drainage (Figs. S1 a-f 2nd panel). This is explained by the fact that in the 2LAY scheme, the drainage is always set to 95% of the soil excess water (above saturation) and runoff can appear only when the total 2m soil is saturated. However, the 11LAY scheme also accounts for runoff that exceeds the infiltration capacity, which depends on the hydraulic conductivity function of soil moisture (Horton runoff). This means that when the soil is dry, the conductivity is low and more runoff will be generated. In the 11LAY simulations, the temporal variability in total runoff (as represented by the error bars in Fig. 1) has also decreased. As just described, in the 11LAY the total runoff mostly corresponds to surface runoff (Figs. S1 a-f). The lower drainage flux (and higher surface runoff) in the 11LAY simulations corresponds well to the calculated water balance at US-SRM (Scott and Biederman, 2019). The 11LAY limited drainage is also likely to be the case at US-Fuf given that nearly all precipitation at the site is partitioned to ET (Dore et al., 2012). In general, all these semi-arid sites have very little precipitation that is not accounted for by ET at the annual scale (Biederman et al., 2017 Table S1).

Across all sites, the magnitude of temporal variability of the total column soil moisture (represented by the error bars in Fig. 1 3rd panel) only increases slightly between the 2LAY and 11LAY model versions. In the upper layer (top 10cm), the soil moisture temporal variability again only increases marginally between the 2LAY and 11LAY for the high-elevation forest sites (Fig. 1 2nd panel error bars); however, the magnitude of variability decreases considerably in the 11LAY model for the low-elevation shrub and grass sites (also see Fig. S2). At all sites the 2LAY upper layer soil moisture simulations fluctuate considerably between field capacity and zero throughout the year, including during dry periods with no rain. These fluctuations are due to the fact that in the 2-layer bucket scheme the top layer can disappear entirely (see section 2.2.2). In the 11LAY however, the temporal dynamics of the upper layer moisture simulations correspond more directly to the timing of rainfall events (see Fig. 2 bottom panel for an example at three sites in 2009 and Fig. S2 for the complete time series for each site). This results in a much better fit of the 11LAY model to the temporal variability seen in the observations (Figs. 2 and S2). This improvement in upper layer soil moisture temporal dynamics is also indicated by the strong increase in correlation at all sites between the re-scaled modelled and observed 11LAY upper layer soil moisture compared to the 2LAY (increases in R ranged from 0.1 to 0.48 – Table 4). Note that not only is the upper high frequency temporal variability therefore arguably more realistic in the 11LAY version, the finer scale discretization of the uppermost soil layer in this version will also allow a much easier comparison with satellite-derived soil moisture products that can only "sense" the upper few cm of the soil (Raoult et al., 2018).

A major and important consequence of the changes in the upper layer soil moisture temporal dynamics is a considerable improvement across all sites in the 11LAY simulated daily ET (Fig. 2 2nd panel, which shows 2009 for three sites; Figs. S2 a-f shows the complete time series for all sites). Across all sites, the 11 LAY RMSE between daily modelled and observed ET has decreased in comparison to the 2LAY and the correlation has increased by a fraction of 0.3 to 0.4 (Table 4). With the exception of US-Vcp, the mean absolute daily ET model-data bias has increased slightly between the 2LAY and 11LAY versions (Table 4), which is due to the fact that the 2LAY version both underestimates and overestimates ET in the spring and summer respectively, resulting in a smaller mean absolute bias (Fig. S3). However, the 11LAY model only slightly

underestimates mean daily ET at most sites, except at US-Fuf. In both model versions, the biases correspond to less than 10% of the mean daily ET across all low elevation sites. At the high elevation sites, the 11LAY bias corresponds to ~20% of the mean daily ET – an increase (decrease) compared to the 2LAY at US-Fuf (US-Vcp). The ratio of modelled to observed standard deviation in ET, $\alpha$, is also provided as a measure of relative variability in the simulated and observed values (Table 4). With the exception of US-Fuf, $\alpha$ values tend closer to 1.0 in the 11LAY simulations compared to the 2LAY – highlighting again that the 11LAYversion does a better job of capturing the daily variability. The higher ET model-data bias and $\alpha$ at US-Fuf is mostly due to model discrepancies in spring (Fig. S2a), which we discuss further in Section 3.3. As previously discussed, the increase in 11LAY model upper layer moisture content at the high-elevation forest sites (Fig. 1 2nd panel and Fig. 2 bottom panel) have resulted in an increase in E and T at those sites, which in turn results in a lower ET RMSE between the model and the observations (Table 4, and see Figs. 2 and S2 2nd panel) if not a decrease in the mean ET bias for US-Fuf (Table 4 and Fig. 1). At the low-elevation shrub and grass sites, the improvement in ET is also related to changes between the two versions in the calculation of the empirical water stress function, $\beta$ (Figs. 2 and S2 5th panel), which acts to limit both photosynthesis and stomatal conductance (therefore, T) during periods of moisture stress (Section 2.2.4). With the new calculation in the 11LAY version (see Section 2.2.4), we see a stronger, more rapid decrease in $\beta$ (increased stress) during warm, dry periods that correspond to strong reductions in T (light brown shaded zones in Fig. 2). Aside from T and E, the other ET components (interception and sublimation) did not change much between the two hydrology schemes (results not shown); therefore, these terms are not contributing to improvements between the 2LAY and 11LAY versions.

The improvement in daily ET temporal dynamics results in an 11LAY mean monthly ET that is also well captured by the model throughout the year, including both the warm, dry May-June period followed by monsoon summer rains, particularly for low-elevation grass and shrub sites (Fig. 3 and Fig. S3). As previously discussed, the improved, higher monthly ET in the 11LAY version during the period of maximum productivity (i.e. the spring and summer for the high-elevation sites, and the summer monsoon for the low-elevation sites – Fig. 3) is likely due to the increase in plant available water (Fig. 1 – 2nd and 3rd panels and Fig. S1). Despite the improvement in the 11LAY temporal variability at the high-elevation forest sites, there is still a bias in the mean monthly ET magnitude between the 11LAY model and observations: At US-Fuf there is a distinct overestimation of ET during the spring (Fig. S3a), whereas at US-Vcp there is a noticeable underestimation of ET during the spring and monsoon periods (Fig. S3b). We will return to these remaining 11LAY ET model-data discrepancies in Section 3.3 after having evaluated the 11LAY soil moisture against observations at different depths.

### 3.2 Comparison of 11LAY soil moisture against observations at different depths

Fig. 4 compares model versus observed daily volumetric soil water content time-series for 2009 at three different depths (see Fig. S4 for the full time series at each site). The complete model time series were re-scaled via linear CDF matching to remove model–observation biases (see Section 2.3.2); however, the linear CDF matching preserves the mean and standard deviation

of the temporal variability. As seen in Section 3.1 and Fig. 2 (bottom panels showing upper 10cm soil moisture), in Fig. 4 the high frequency temporal variability of the 11LAY soil moisture in the uppermost layer almost perfectly matches the observed,

particularly at the low-elevation shrub- and grass-dominated sites (US-SRM, US-SRG, US-Whs, US-Wkg). At most of the low-elevation sites the soil moisture drying rates in the upper 20cm of soil are well captured by the model, with the small exception of the Santa Rita sites between January to March in which the model appears to dry down at a faster rate than observed (Fig. 4 US-SRM and US-SRG top and middle panels).

In contrast, the temporal mismatch between the observations and the model in the uppermost layer is higher at the forest sites.

The US-Fuf and US-Vcp 11LAY simulations appear to compare reasonably well with observations in the upper 2cm of the soil from June through to the end of November (end of September in the case of US-Vcp) (Fig. 4). However, in some years the model appears to overestimate the SWC at both sites during the winter months (positive model-data bias), and underestimate the observed SWC during the spring months (negative model-data bias), particularly at US-Fuf. Although US-Fuf and US-Vcp are semi-arid sites, their high-elevation means that during winter precipitation falls as snow; therefore, these

apparent model biases may be related to: i) the ORCHIDEE snow scheme; ii) incorrect snowfall meteorological forcing; and/or iii) incorrect soil moisture measurements under a snow pack. During the early winter period the model soil moisture increases rapidly as the snow pack melts and is replenished by new snowfall, whereas the observed soil moisture response is often slower (Fig. 5a and b light blue shaded zones). This often coincides with periods when the soil temperature in the model is below 0°C (Fig. 5 bottom panel), suggesting that in the field soil freezing may be negatively biasing the soil moisture measurements. An

alternative explanation is that ORCHIDEE overestimates snow cover (and therefore snow melt and soil moisture) at the forest sites because it assumes that snow is evenly distributed across the grid cell, whereas in reality the snow mass/depth is lower under the forest canopy than in the clearings.

At US-Fuf, it appears that the model melts snow quite rapidly after the main period of snowfall (Fig. 5a light green shaded zones). Once all the snow has melted, the model soil moisture also declines; however, the observed soil moisture often remains

high throughout the spring – causing a negative model-data bias (Fig. 5a). Unlike US-Fuf, a similar negative model-data bias at US-Vcp often coincides with periods when snow is still falling, although the amount is typically lower (Fig. 5b light green shaded zones); however, the model does not always simulate a high snow mass during these periods. These periods coincide with rising surface temperature above 0°C. Although snow cover, mass, or depth data have not been collected at these sites, snow typically remains on the ground until late spring after winters with heavy snowfall, suggesting the continued existence

of a snow pack and slower snow melt that replenishes soil moisture until late spring when all the snow melts. Therefore, the lack of a simulated snow pack into late spring could explain the negative model-data soil moisture bias. To test the hypothesis that the model melts or sublimates snow too rapidly, thereby limiting the duration of the snowpack and also allowing surface temperatures to rise, we altered the model to artificially increase snow albedo and decrease the amount of sublimation; however, these tests had little impact on the rate of snow melt or the duration of snow cover (results not shown). Aside from

model structural or parametric error, it is possible that there is an error in the meteorological forcing data. Rain gauges may underestimate the actual snowfall amount during the periods when it is snowing (Rasmussen et al., 2012; Chubb et al., 2015).

If the snowfall is actually higher than is measured, it may in reality lead to a longer lasting snowpack than is estimated by the model. To test this hypothesis, we artificially increased the meteorological forcing snowfall amount by a factor of ten and re-ran the simulations. Although this artificial increase is likely exaggerated, the result was an improvement in the modelled springtime soil moisture estimates at US-Fuf (Fig. S5). However, the same test increased the positive model-data bias in the early winter at US-Fuf, and degraded the model simulations at US-Vcp. This preliminary test suggests that inaccurate snowfall forcing estimates may play a role in causing any negative model-data bias spring soil SWC but more investigation is needed to accurately diagnose the cause of the springtime negative model-data bias.

Overall, there is a decrease in the model ability to capture both high frequency and seasonal variability with increasing soil depth. At all sites the temporal dynamics of the deepest observations are not well represented in the model (Fig. 4 bottom panels for each site). At the high-elevation forest sites (US-Fuf and US-Vcp), the model does not capture the response of observed soil moisture in the deepest layer to summer storm events. In contrast, at the low-elevation shrub and grass sites the 11LAY SWC is far too dynamic in the deepest layer. The smoother model temporal profile at depth at the forest sites compared to the sites with higher grass fraction is likely related to impact of rooting depth on exponential changes in Ks towards the surface (see Section 2.2.2). As the forests have deeper roots, the increase in Ks starts from a lower depth in the soil profile than the more grass-dominated sites, which in turn allows for a quicker infiltration of moisture to deeper layers. The higher Ks at depth also allows for a higher drainage and therefore decreased soil moisture temporal variability. However, this description of the model behaviour does not explain the model-data discrepancies. The poor model-data fit at lower depths may be related to the discretization of the soil column with a geometric increase of internode distance. Therefore, the soil layer thicknesses increase substantially beyond ~2-4cm (7th and 8th soil layers – Table S1). For the deeper soil moisture observations, it is therefore harder to match the depth of the observations with a specific soil layer. Alternatively, it is possible that the model description of a vertical root density profile, which is used to calculate changes in Ks with depth, is too simplistic for semi-arid vegetation that typically have extensive lateral root systems that are better adapted for water-limited environments. It is also possible that assigning semi-arid tree and shrub types to temperate PFTs, as we have done in this study in the absence of semi-arid specific PFTs, has resulted in a root density decay factor that is too shallow. In contrast to temperate forests, semi-arid trees and shrubs also often have deep taproots for accessing groundwater. Finally, changes in soil texture that may occur in reality with depth in the soil could alter hydraulic conductivity parameters; in the model however, hydraulic conductivity only changes (exponentially) with depth owing to soil compaction (see Section 2.2.2). In addition, semi-arid region soils often have a higher concentration of rock and gravel (Grippa et al., 2017) – neither of which are represented in the ORCHIDEE soil texture classes.

## 3.3 Remaining discrepancies in ET and its component fluxes

Despite the improvement in seasonal ET temporal dynamics in the 11LAY model, particularly the timing of the reduction during the dry season, key model-data discrepancies in ET remain during spring (March-April) and monsoon (July-September) periods: i) At US-Fuf, the 11LAY ET is overestimated during the spring and early summer (Fig. S3a); ii) At US-Vcp, the model underestimates ET for much of the growing season, likely due to low LAI values in the earlier and later years of the simulation (Fig. S3b); iii) at US-SRM the 11LAY model overestimates springtime ET (in contrast to other low-elevation monsoon sites) (Fig. S3c); and iv) the 11LAY model still slightly underestimates peak monsoon ET at the low-elevation shrub sites (US-SRM and US-Whs– Figs. S3 c-d) as seen in a previous semi-arid model evaluation study (Grippa et al., 2011).

The model overestimate in spring ET at US-Fuf could be related to the snowfall issues that are causing the model to underestimate spring soil moisture during the same period (Figs. 4 and 5 and see Section 3.2). The lack of a persistent snow pack in the model during this period can explain the positive bias in spring ET because in reality the presence of snow would suppress bare soil evaporation. As discussed in Section 3.2, to accurately diagnose this issue we would need further information on snow mass or depth. Further support for the suggestion that modelled spring E is overestimated comes from comparing the model with estimated T/ET ratios (Fig. 6). Although both E and T increase in the US-Fuf (and US-Vcp) 11LAY simulations (compared to the 2LAY – Fig. S3a and b) due to the increase in upper layer soil moisture (as previously described in Section 3.1 and Figs. 2 and S2a and b), the stronger increase in 11LAY E compared to T resulted in lower 11LAY T/ET ratios across all seasons (Fig. S3a and b). While the model captures the bimodal seasonality at the forested sites as seen in the Z16 data-derived estimates (Fig. 6), the magnitude of model T/ET ratios appear to be too low in all seasons given the 100% tree cover at these sites with a maximum LAI of ~2.4. Whilst low spring 11LAY T/ET ratios at US-Fuf may be due to overestimated E as a result of higher soil moisture and underestimated snow cover, the generally low bias in T/ET ratios across all seasons at both US-Fuf and US-Vcp may also point to the issue that no bare soil evaporation resistance term is included in the default 11LAY version. This may explain why the model T/ET ratios do not increase as rapidly as estimated values at the start of the monsoon (Fig. 6). However, discrepancies in the timing of T/ET ratio peak and troughs between the model and data-derived estimates at the forested sites could also be due to the fact evergreen PFTs have no associated phenology modules in ORCHIDEE; instead, changes in LAI are only subject to leaf turnover as a result of leaf longevity, which may be an oversimplification.

At US-SRM, the modelled spring T/ET ratio overestimates the Z16 estimate and underestimates the SB17 estimate (Fig. 6). The current state of the art is that different methods for estimating T/ET typically compare well in terms of seasonality but differ in absolute magnitude; therefore, the uncertainty in data-derived estimates of T/ET magnitude during the spring at US-SRM makes it difficult to glean any information on whether T or E (or both) are responsible for the 11LAY overestimate of modelled springtime ET (Fig. S3c). If the SB17 method is more accurate, then it is probable that modelled spring E at this site is too high (T/ET underestimated), again potentially due to the lack of the bare soil evaporation resistance term in the default 11LAY configuration. However, if the Z16 estimate is accurate, then it is likely that spring T is overestimated at US-SRM,

potentially due to an overestimate in LAI. The model-data bias in spring mean monthly ET appears to correlate well with modelled spring mean LAI at US-SRM (Fig. S6). If model LAI at US-SRM is too high during the spring, it is impossible to determine whether the shrub or grass LAI are inaccurate without independent, accurate estimates of seasonal leaf area for each vegetation type, which are not available at present; however, in the field the spring C4 grass LAI is typically half that of its

monsoon peak – a pattern not seen in the model (Fig. S6).

During the monsoon at the low elevation grass- and shrub-dominated sites, both data-derived estimates of T/ET agree on the seasonality and, while different magnitudes, both are higher than the model T/ET values (Fig. 6). Given this agreement, both sets of estimated values can help to diagnose why the 11LAY model also underestimates monsoon peak ET at the low-elevation shrub sites (US-SRM and US-Whs– Figs. S3 c-d). The underestimate in modelled monsoon T/ET ratios across all grassland

and shrubland sites could be either because T is too low or E is too high. At the shrubland sites (US-SRM and US-Whs), both monsoon ET and T/ET are underestimated; therefore, for these sites it is plausible that the dominant cause is a lack of transpiring leaf area. As was the case for spring ET at US-SRM, monsoon model-data ET biases are better correlated with LAI at shrubland sites compared to grassland sites (Fig. S8). In contrast, at the grassland sites (US-SRG and US-Wkg) monsoon ET is well approximated by the 11LAY model; thus, the underestimate in T/ET ratios suggests that both the transpiration is

too low and the bare soil evaporation too high. Furthermore, although the 11LAY does capture the decrease in ET during the hot, dry period of May to June at the grass and shrub sites (which is a significant improvement compared to the 2LAY – see Section 3.1), the 11LAY T/ET ratios are slightly out of phase with the estimated values. Both data-derived estimates agree that T/ET ratios at all grass and shrub sites decline in June during the hottest, driest month (as expected); however, the model T/ET ratios reach a minimum one month later in July (Fig. 6). This one-month lag in model T/ET ratios is apparent despite the fact

that the ET minimum is accurately captured by the model (Figs. 3b and S3c-f). The modelled T/ET ratios also do not increase as rapidly as both estimates during the wet monsoon period (July – September), which can be explained by the fact that the model E at the start of the monsoon increases much more rapidly than modelled T. Taken together, these results suggest that LAI is not increasing rapidly enough after the start of monsoon rains (see Fig. S7), resulting in negatively biased T/ET ratios in July. Meanwhile the increase in available moisture from monsoon rains, potentially coupled with a lack of bare soil

evaporation resistance in the default 11LAY version, is causing a positively biased model E that compensates for the lower T. These compensating errors result in accurate ET simulations. The underestimate in modelled leaf area during the monsoon could either be: i) incorrect timing of leaf growth for either grasses or shrubs and an underestimate of peak LAI; and/or ii) due to the fact the static vegetation fractions prescribed in the model do not allow for an increase in vegetation cover during the wet season (i.e. the model lacks the ability to grow grass in interstitial bare soil areas).

We attempted to explore both the hypotheses that could explain discrepancies in model ET and T/ET ratios (incorrect T due to lack of transpiring leaf area at low elevation grass and shrub sites, or overestimated E across all sites) with two further tests. These final tests and their results are described in the following section.

**3.4 Testing decreased bare soil cover and the addition of the 11LAY bare soil resistance term**

To further investigate the possibility that summer ET and T/ET ratios are underestimated at low-elevation sites because of a lack of transpiring leaf area, we reduced the bare soil fraction and increased C4 grass fraction to the maximum observed C4 grass cover under the most productive conditions. This decrease in bare soil fraction increased ET and T/ET ratios during the monsoon period at all low elevation grass- and shrub-dominated sites and also increased ET during spring at the Santa Rita sites (Fig S9; mean across low elevation sites in Fig. 7). However, although the T/ET ratios reduced the negative model biases

in the summer monsoon period when compared to the data-derived estimates, the model now overestimated ET in all seasons (Figs. 7 and S9). Furthermore, the spring ET model-data bias at US-SRM was further exacerbated by the decrease in bare soil fraction (Fig. S9) and the mean estimated T/ET ratios across all low elevation grass and shrub sites were a closer match to the original 11LAY version (Fig. 7). And finally, while the decrease of the bare soil fraction (increase in C4 grasses) may have partially accounted for the negative bias in T/ET ratios at the start of the monsoon, the changes did not correct the phase

discrepancy between the estimated and modelled T/ET seasonal trajectories: the estimated T/ET still declined to a minimum in June (as expected during the hot, dry period), whereas the model declined one month later. Putting the latter points together, this new test gives further weight to the suggestion put forward in Section 3.3 that the model is not capturing the correct *increase* in leaf area at the start of the monsoon – i.e. the problem is not just that there is a lack in the overall amount of transpiring leaf area (or a too high bare soil fraction) – due to issues with the model phenology for individual PFTs and/or its

ability to capture dynamic changes in seasonal vegetation cover.

    As described in Section 3.3, the remaining model ET issues (and its component fluxes) in both high-elevation forest sites and low-elevation shrub- and grass-dominated sites could also be due to the fact the model simulates too much bare soil evaporation. The 11LAY version has an optional bare soil evaporation resistance term that is not activated in the default version; therefore, the 11LAY simulations presented thus far have not included any such resistance term. Therefore, we tested

the inclusion of the bare soil resistance term at all sites. Although there is no bare soil fraction at the high-elevation forested sites (US-Fuf and US-Vcp), in the 11LAY version E still occurs over the bare soil sub-fraction of the vegetated soil tiles. The bare soil sub-fraction of the vegetated soil tiles increases at low LAI during winter months (see section 2.2.1); therefore, including the bare soil resistance term caused a reduction in E during the winter (lower LAI - Fig. 8 bottom left panel). The reduction in winter E at the forested sites in turn allowed for higher overall soil moisture content (Figs. S10 a and b) and

therefore a greater T (and E) during the spring and summer (Fig. 8 – left column). As a result, T/ET ratios were increased with the addition of the bare soil evaporation term, thus potentially partially resolving the issue of negatively biased T/ET ratio issue seen in the default 11LAY simulations (see Section 3.3). The increase in plant available moisture with the addition of the resistance term also led to a strong increase in LAI at US-Vcp from a mean around 0.5 to a mean around 2.1 (Fig. S10b), which is much closer to the observed LAI for the site. However, the dramatic increase in T resulted in a simulated ET at both

forest sites that strongly overestimated the observations (Fig. 8 and Figs. S10a and b); therefore, overall the addition of the

bare soil evaporation resistance term did not improve the ET model-data fit at these sites. As discussed in Section 3.2, spring ET may also be overestimated at these sites due to the lack of a persistent snowpack.

At all the low-elevation grass and shrub sites the addition of the bare soil resistance term resulted in a strong decrease in soil evaporation during the monsoon season, and a lesser, but non-negligible, decrease to almost zero evaporation during the winter (Fig. 8 – right column). Bare soil evaporation remained much the same during the spring and the hot, dry season months of May and June. As seen for the forest sites, the decline in bare soil evaporation during the monsoon period results in a slightly higher moisture storage (Figs. S10c-f), which in turn fractionally increases T throughout the year (Fig. 8). The net effect is a reduction in ET during summer and winter and an increase in spring and dry season ET (Fig. 8). However, as for the forested sites, this net effect in the simulated ET produces a worse fit to the data. Therefore, the addition of this term does not resolve the ET issues documented in Section 3.3: A further positive bias in spring ET estimates is observed at US-SRM (Fig. S10c), and the underestimate in monsoon ET at US-SRM and US-Whs (Figs. S10c and d) is further exacerbated. Furthermore, the near zero evaporation in the winter months with the introduction of the bare soil resistance term results in an increase in winter T/ET ratios. Therefore, at the low-elevation sites the monthly seasonality of T/ET differs quite considerably from the default 11LAY model runs (Figs. S10c-f) and generally does not follow the seasonal trajectories estimated by either Zhou et al. (2016) or Scott et al. (2017) (Fig. 6).

In a final test, we combined both the decrease in bare soil fraction with the addition of the bare soil resistance term for the low elevation sites. The addition of the bare soil resistance term reduced the positive bias seen with the increase of C4 grass (decrease of the bare soil fraction) (Fig. S11). However, as seen in the bare soil resistance tests with the original vegetation and bare soil fractions, the addition of the resistance term increased spring T due to the higher spring soil moisture – thus exacerbating the positive bias in ET. It is clear that neither of these tests fully deal with remaining ET model-data biases in the 11LAY version – nor do they account for the issues in the model seasonality of T/ET ratios. The ET seasonal temporal dynamics remain much the same in all tests. We point out however that the model fit to ET observations was still greatly improved in the 11LAY version compared to the 2LAY and many of the remaining model-data discrepancies are less significant by comparison. It is therefore possible that some combination of the additional bare soil evaporation resistance term, decreased bare soil fraction, improved semi-arid leaf phenology schemes, and further calibration of hydrology, phenology, stomatal conductance, and water-limitation parameters would be able to resolve most, if not all, of the remaining model-data discrepancies in ET and T/ET estimates at these sites. This is beyond the scope of this study, but the options are discussed more in Section 4.

## 4 Discussion

This study showed that in comparison to a simple bucket model (Manabe, 1969), a discretized soil hydrology scheme based on the Richards equation – and associated model developments – results in considerable improvements in simulated semi-arid site soil moisture temporal dynamics that exhibit a more realistic response to rainfall events (contrary to the model-data

comparison of Lohou et al., 2014). As a result, we see dramatic improvements in high temporal frequency to seasonal ET simulations. Previous studies have also demonstrated that the more mechanistic descriptions of soil hydrology included in the latest LSM versions have resulted in improvements to surface latent and sensible heat fluxes (de Rosnay et al., 2002; Best et al., 2015); yet, few studies have specifically compared these two model versions across a range semi-arid ecosystems, as we have attempted in this study. However, there remain a number of missing hydrological processes that have not yet been incorporated into LSMs, and/or inadequate existing processes, which will clearly have an impact on semi-arid hydrological modeling (Boone et al., 2009; Grippa et al., 2017) and may resolve some of the remaining model-data discrepancies we were not able to address in this study. We highlight these in the sections below.

**Issues with modelling vegetation dynamics in semi-arid ecosystems**

Our analysis has suggested that that biases in low-elevation shrub and grassland site ET might be due to incorrect simulations of seasonal vegetation dynamics; therefore, in order to obtain realistic estimates of ET and its component fluxes, it is important that the model can accurately simulate seasonal changes in leaf area and/or grass versus bare soil fractional cover. The connection between vegetation fractional cover and LAI is a particular issue in sparsely vegetated regions when low LAI effectively means more bare soil is coupled with the atmosphere and E increases. To account for this in ORCHIDEE, the bare soil fraction is slightly increased when LAI is low (see section 2.2.1), which is often the case at these sites; however, there are only limited observations to support this model specification. Similarly, there are not many LAI measurements for grasses and shrubs in these ecosystems; therefore, we have relied on estimating the LAI$_{max}$ parameter from MODIS LAI data. While different satellite LAI products often correspond well to each other in terms of temporal variability, there is often a considerable spread in their absolute LAI values (Garrigues et al., 2008; Fan et al., 2013); therefore, the MODIS LAI peak values may not be accurate for these ecosystems. In any case, the satellite LAI values represent a mix of different vegetation types, and unlike satellite reflectance data it is not possible to linearly unmix the satellite LAI estimates based on fractional cover. More field LAI measurements are needed from different vegetation types (especially annual versus perennial grasses and shrubs) to verify what the likely maximum LAI is for each PFT.

As mentioned in the results, it is also possible that LSMs contain an inaccurate representation of different semi-arid vegetation phenology, including drought-deciduous shrubs and annual versus perennial C4 grasses. The model does yet discern between perennial grasses and annual C4 grasses that only grow during warmest, wettest periods (Smith et al., 1997). It is possible that LSMs need new phenology models that account for annual C4 grass strategies in order to obtain accurate simulations of semi-arid water and carbon fluxes. Finally, it is possible that incorrect seasonal LAI trajectories are also causing the issues in the T/ET seasonality seen at the higher elevation forested sites due to the lack of an evergreen phenology module in ORCHIDEE. Recently, a new evergreen phenology module has been implemented in ORCHIDEE (Chen et al., 2020); however, this scheme was developed for humid tropical forests. Testing it for evergreen trees in semi-arid regions is beyond the scope of this study

but will be investigated in future work. Again, seasonal LAI measurements of different high elevation semi-arid vegetation types would significantly help to improve or further develop semi-arid phenology models.

Alternatively, it may be that other model parameters and processes involved in leaf growth – for example phenology, root zone plant water uptake, water-limitation, and photosynthesis-related parameters – are inaccurate and in need of statistical calibration (e.g. MacBean et al., 2015). Incorrect representations of how we model low temperature and high VPD constraints on stomatal conductance may also play a role. At the high-elevation sites, we assumed the ponderosa pine trees should be modelled as temperate needleleaved evergreen PFT. The default model parameters assigned to this PFT may not be appropriate for modelling this plant functional type in water-limited semi-arid environments. Another likely issue for modelling low elevation sparsely vegetated semi-arid ecosystems with ORCHIDEE is that there is no specific shrub PFT, although a recent ORCHIDEE version includes shrub PFTs for high latitude tundra ecosystems (Druel et al., 2017). In future work we will adapt similar shrub parameterizations for semi-arid environments.

The importance of vegetation cover and seasonal changes in leaf area for modeling hydrological fluxes – particularly T – is not a new observation (e.g. Ha et al., 2015; Grippa et al., 2017). Baldocchi et al. (2010) found that LAI was important at five Mediterranean sites in California and Europe for determining how much carbon is assimilated and how much water is lost. Hogue et al. (2005) also found the Noah LSM was not able to replicate monsoon period LE increases at the Walnut Gulch sites, which they suggested may be related to inaccuracies in the satellite greenness fraction estimates that are used to run the model. Whitley et al. (2016 and 2017) also proposed that any improvements needed for terrestrial biosphere modelling of savanna ecosystems should include modifications to the phenology schemes and the split between fractional cover of trees and grasses.

**ET partitioning (T/ET ratio)**

In agreement with this study, Lian et al. (2018) also show that CMIP5 models vastly underestimate T/ET ratios. They estimated a new global T/ET ratio of $0.62 \pm 0.06$, which is similar to the upscaled estimate of 0.57 +/= 7% of Wei et al. (2017), and suggest that model underestimates could be caused by misrepresentation of vegetation structure impacts on canopy light use, interception loss and root water uptake. Their conclusions lend further weight to our suggestion that further improvements in T/ET ratios may result from more accurate simulations of seasonal phenology and fractional vegetation cover (see Section 3.3). Alternatively, Chang et al., (2018) have suggested that neglecting to account for lateral redistribution of moisture is responsible for model inability to capture T/ET partitioning. Current LSM versions do not simulate extensive shallow root systems that are typical of semi-arid vegetation that is more adapted to water limited conditions. However, they also mention other LSM issues that might be affecting the T/ET ratio, such as the lack of root dynamics, vegetation shading, topographic effects and the representation of bare soil evaporation. In order to properly diagnose if discrepancies in modelled T/ET are caused by inaccurate representation of lateral moisture redistribution, we need to perform a comparison of a spatially distributed model simulation with a high-density network of hydrological observations. Nevertheless, in spatially

heterogeneous mixed shrub-grass ecosystems it seems likely that missing model processes will need to be accounted for before accurate simulations of T/ET ratios can be achieved. One example of this might be the need to include in the model a representation of shrub understory and below canopy E. Diagnosing and addressing discrepancies between modelled and estimated T/ET is important, specifically for dryland ecosystems where increases in vegetation productivity and/or cover in response to rising atmospheric $CO_2$ appears to be driving higher T/ET rates (Lian et al., 2018).

**Bare soil evaporation**

The addition of a term that simulates bare soil evaporation resistance to dry soil served to alleviate discrepancies in T/ET ratios compared to data-derived estimates; however, resulting changes in modelled ET provided a worse fit to the observations. It is possible that the bare soil resistance is only part of the solution, as discussed in Section 3.4. Future studies could also investigate the impact of uncertainty in the use of pedotransfer functions (e.g. Mermoud et al., 2006) in deriving soil hydraulic parameters from soil texture information. The low-elevation sites typically have a very cobbly, rocky soil surface that is not accounted for in ORCHIDEE. Including soil texture variability with different soil horizons could further improve ORCHIDEE's capability to capture the correct E, ET and T/ET ratios. Alternatively, the relatively simple implementation of a bare soil resistance term (Eq. 2 – Section 2.2.3) might need to be adapted to include bare soil evaporation resistance across a litter or biocrust layer. Atthe sparsely vegetated grass and shrub dominated sites in southern Arizona biological soil crusts (biocrusts) composed of assemblages of lichens, bryophytes, cyanobacteria, algae and microbes form across much of the bare soil surface (Belnap et al., 2016). Biocrust layers may significantly alter bare soil evaporation (and other aspects of ecosystem ecology and functioning – Ferrenberg et al., 2017) in sparsely vegetated regions in ways that have not yet been considered in any LSM bare soil evaporation scheme. Therefore, it is possible that in addition to a more mechanistically-based formulation of resistance to bare soil evaporation due to a litter layer (as per Swenson and Lawrence, 2014 or Decker et al., 2017), separate formulations of evaporation through biocrust/mulch layers may need to be developed (e.g. Saux-Picard et al., 2009).

**High-elevation model snowpack and snow melt predictions**

The model also needs to be tested at other high-elevation semi-arid mountainous sites (such as the Sierra Nevada mountains in California) for which spring snowmelt is the predominant (and controlling) annual source of moisture. More specifically, more information on snow cover, depth or mass, particularly under closed forest canopies, would be useful to diagnose potential sources of bias in the snowfall simulations. It is crucial that LSMs accurately capture semi-arid high-elevation snowfall temporal dynamics if we are to have unbiased projections in future moisture availability and productivity for these regions.

**Implications for modelling plant water stress**

Similar to Whitley et al. (2016), the original 2LAY version of the model underpredicted wet monsoon season ET. The peak ET fluxes were generally much better captured in the 11LAY version. However, in contrast to the findings of Whitley et al. (2016), the 2LAY simulations overestimated ET during the hottest, driest period between May and June. Our results demonstrated that a modified empirical beta water stress function (used to downregulate stomatal conductance during periods of limited moisture) that takes into account available soil moisture and root density across the entire soil column (Section 2.2.4) helped to better capture dry season ET dynamics. These results are interesting in light of previous studies showing that LSMs employing empirical beta water stress functions show considerable differences in their simulated response to water stressed periods (Medlyn et al., 2016; De Kauwe et al., 2017). These studies argue for more evidence-based formulations of plant response to drought. De Kauwe et al. (2015) also highlight the need for models to incorporate dynamic root zone soil moisture uptake down profile as the soil dries. It is therefore possible that while the modified beta function used in the 11LAY does help to capture seasonal water stress, as seen across sites in this study, new mechanistic plant hydraulic schemes that can track transport of water through the xylem (e.g. Bonan et al., 2014; Naudts et al., 2015) may be needed when simulating plant response to prolonged drought periods. However, comparing beta functions versus plant hydraulic schemes under severe water stressed periods was not within the scope of this study. When discussing woody plant responses to drought, it is also worth noting that many LSMs to date are also missing any representation of groundwater (Clark et al., 2015). As described in Section 2.1, the water table is typically very deep (10s to 100s metres) at these sites. Previous modeling studies have shown that only rather shallow water tables (~1m) are likely to significantly increase ET in the SW US (e.g. by >=2.4mmd$^{-1}$ in Fig. 4g of Wang et al., 2018). However, the fact LSMs typically do not include adequate descriptions of groundwater (and deeper tap roots) could impact their ability to simulate semi-arid ecosystem water uptake in the dry season given that drought deciduous shrubs are more resilient to droughts due to their ability to access groundwater reserves (e.g. Miller et al., 2010). A new groundwater module is being developed for ORCHIDEE and will be tested in future studies.

**5 Conclusions**

These results strongly suggest that a more complex, process-based hydrology model – in particular, one which contains fine scale discretization of the upper soil moisture layers and associated improvements in bare soil evaporation and plant water stress functions – improves daily to seasonal predictions of the upper layer root-zone soil moisture dynamics and ET (as seen in de Rosnay et al., 2002). In particular, there is a dramatic improvement in the model's ability to capture the decline in ET during the hot, dry May-June period. Associated changes in the calculations of runoff, soil moisture infiltration, and bottom layer drainage also appear to result in more plausible (lower) simulations of total runoff (surface runoff plus drainage) at the forest sites given that across all these semi-arid sites most precipitation is accounted for by ET at the annual scale. Such improvements might counter previous work highlighting that models tend to overestimate runoff (Grippa et al., 2017).

ORCHIDEE CMIP5 simulations used the 2-layer conceptual bucket scheme of Manabe (1969); therefore, ORCHIDEE CMIP5 predictions of semi-arid water availability and consequent impacts on ecosystem functioning and feedbacks to climate were likely inaccurate. Despite the appeal of simplicity and low calculation costs, 2-layer simple bucket hydrology models are likely unsuitable for accurate semi-arid water flux simulations (at least in the semi-arid SW US). The forthcoming ORCHIDEE CMIP6 simulations will likely provide more accurate and reliable results of semi-arid soil moisture availability and evapotranspiration.

Remaining discrepancies in both overestimated and underestimated winter and spring soil moisture at high-elevation semi-arid forested sites might be respectively related to issues with soil moisture data during periods of soil freezing and/or underestimated snowfall forcing data causing a limited duration snowpack, with consequent implications for predictions of water availability in regions that rely on springtime snowmelt. However, biases in soil moisture at both the forested sites do not translate into the same biases in modelled ET, suggesting other factors such as issues in evergreen phenology or the lack of resistance to bare soil evaporation may also play a role.

The addition of an empirical bare soil evaporation resistance term by itself did not improve estimates of ET in these ecosystems, although T/ET ratios were increased, potentially reducing the negative biases in the monsoon season when comparing to data-derived T/ET estimates. The increase in transpiring leaf area (from a reduction in bare soil fraction) at the low elevation forest sites also could account for the same monsoon season T/ET bias. However, issues in the timing of the simulated transition from low to high T/ET ratios at the start of the monsoon remain. Our analysis shows that remaining discrepancies in semi-arid site ET simulations (and its constituent fluxes) might therefore be related to a combination of factors impacting both the amount and timing of transpiring leaf area and resistance to bare soil evaporation. We recommend that future work on improving LSM semi-arid hydrological predictions focuses not only on issues highlighted in previous studies such as dynamic root zone moisture uptake, inclusion of ground water, lateral and vertical redistribution of moisture (e.g. Whitley et al., 2016; 2017; Grippa et al., 2017) but also on: i) multi-variable calibration of vegetation and hydrology-related parameters across all sites; ii) more data to better evaluate the seasonal trajectory of LAI across all sites as well as the vegetation fractional cover and peak LAI magnitude at low elevation sites; iii) more data to test modelled snow mass or depth at high elevation sites; and iv) testing of a more mechanistic description of resistance to bare soil evaporation.

**Code availability**

The ORCHIDEE v2.0 model code and documentation are publicly available via the ORCHIDEE wiki page (http://forge.ipsl.jussieu.fr/orchidee/browser/tags/ORCHIDEE_2_0/ORCHIDEE/) under the CeCILL license (http://www.cecill.info/index.en.html). The ORCHIDEE model code is written in Fortran 90 and is maintained and developed under an SVN version control system at the Institute Pierre Simon Laplace (IPSL) in France. Simulation post-processing and

plotting scripts were performed in Python and are provided on NM's GitHub repository: https://github.com/nmacbean/SW-US-Hydro-Model-Eval-HESS.

## Data availability

Meteorological forcing and evapotranspiration data for each are available from the Ameriflux site: https://ameriflux.lbl.gov. Soil moisture was obtained directly from site PIs. Vegetation and soil texture characteristics were derived from the published literature, as specified in Table 1, and from site PIs. Model simulations are provided on NM's GitHub repository: https://github.com/nmacbean/SW-US-Hydro-Model-Eval-HESS.

## Author contribution

NM, RLS, JAB and DJPM designed the overall study. NM carried out the model simulations, post-simulation analysis and figure plotting. CO, NV and AD provided detailed inputs on model description/code and recommendations for further tests to diagnose model-data deficiencies. NV provided scripts to gap-fill the meteorological data. JAB gap-filled the ET data. RLS, JAB, TK and ML provided gap-filled soil moisture data and information on site characteristics and typical behaviour of seasonal vegetation cover, LAI, and snowfall. NM wrote the manuscript. All co-authors provided detailed comments,
suggestions, and edits on the first and second drafts of the manuscript.

## Competing Interests

The authors declare that they have no conflict of interest.

## Acknowledgements

Funding for AmeriFlux data resources and data collection at US-SRM, US-SRG, US-Wkg, and US-Whs was provided by the
890 U.S. Department of Energy's Office of Science and the USDA. Data collection at US-Fuf supported by grants from the North American Carbon Program/USDA CREES NRI (2004-3511115057), the U.S. National Science Foundation MRI Program, Science Foundation Arizona (CAA 0-203-08), the Arizona Water Institute, and the Mission Research Program, School of Forestry, Northern Arizona University (McIntire-Stennis/ Arizona Bureau of Forestry). The US-Vcp site funded by U.S. DOE Office of Science through the AmeriFlux Management Project (AMP) at Lawrence Berkeley National Laboratory (Award
#7074628) and Catalina-Jemez Critical Zone Observatory (NSF EAR 1331408). NM was funded by the US National Science Foundation Award Numbers 1065790 (Emerging Frontiers Program) and 1754430 (Division of Environmental Biology Ecosystems Program). We would like to thank the ORCHIDEE team for development and maintenance of the ORCHIDEE

code and for providing the ORCHIDEE version used in this study. Finally, we thank the three anonymous referees for their comprehensive and useful reviews.

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

**Table 1: Site descriptions, period of available site data, and associated ORCHIDEE model parameters, including vegetation plant functional type (PFT), soil texture fractions and maximum LAI used in ORCHIDEE model simulations (also see Table 1 for general site descriptions). Simulation period correspond to the period of available site data. PFT fractional cover and the fraction of each soil texture class are defined in ORCHIDEE by the user. Note that ORCHIDEE does not contain and explicit representation of shrub PFTs; therefore, shrubs were included in the forest PFTs. The maximum LAI has a default setting in ORCHIDEE that has not been used here; instead, values based on the site literature have been prescribed in the model. The USDA soil texture classification (12 classes – see section 2.3.3 for description) is used to define hydraulic parameters in the ll-layer mechanistic hydrology scheme (see Section 2.2.2 and 2.2.3 for a description). For some sites, soil texture fractions are taken from the ancillary Ameriflux BADM (Biological, Ancillary, Disturbance and Metadata) Data Product BIF (BADM Interchange Format) files (see https://ameriflux.lbl.gov/data/aboutdata/badm-data-product/) that are downloaded with the site data. PFT acronyms: BS = Bare soil; TeNE = Temperate Needleleaved Evergreen forest; TeBE = Temperate Broadleaved Evergreen forest; TeBD = Temperate Broadleaved Deciduous forest; C3G = C3 grass; C4G = C4 grass.**

| Site ID | Description | Dominant Species | Soil texture | Period of site data | PFT fractions | Soil texture class fractions | Maximum LAI | Reference |
|---|---|---|---|---|---|---|---|---|
| US-SRM | Shrub encroached C4 grassland / savanna | *Prosopis velutina, Eragrostis lehmanniana* | Deep loamy sands | 2004-2015 | 50% BS; 35% TeBD; 15% C4G | USDA: Loamy sand | 0.85 (TeBD & C4G) | Scott et al. (2015); Ameriflux BADM. |
| US-SRG | C4 grassland | *Eragrostis lehmanniana* | Deep loamy sands | 2008-2015 | 45% BS; 11% TeBD; 44% C4G | USDA: Loamy sand | 1.0 (C4G) | Scott et al. (2015) |
| US-Whs | Shrub-dominated shrubland | *Larrea tridentata, Parthenium incanum, Acacia constricta, Rhus microphylla* | Gravelly sandy loams | 2007-2015 | 57% BS; 40% TeBE; 3% C4G | USDA: Sandy loam | 0.6 (TeBE & C4G) | Scott et al. (2015) |
| US-Wkg | C4 grassland | *Eragrostis lehmanniana, Bouteloua* spp. *Calliandra eriophylla* | Very gravelly, sandy to fine sandy, and clayey loams | 2004-2015 | 60% BS; 3% TeBE; 37% C4G | USDA: Sandy loam | 0.85 (C4G) | Scott et al. (2015); Ameriflux BADM. |
| US-Fuf | Unmanaged ponderosa pine forest | *Pinus ponderosa* | Clay loam | 2005-2010 | 100% TeNE | USDA: Clay loam | 2.4 | Dore et al. (2010; 2012); Ameriflux BADM. |

| US-Vcp | Unmanaged ponderosa pine forest | *Pinus ponderosa* | Silt loam | 2007-2014 | 100% TeNE | USDA: Silt loam | 2.4 | Anderson-Teixeira et al. (2011) |

**Table 2: Summary of differences between 2LAY and 11LAY model versions. All other parameters and processes in the model, including the PFT and soil texture fractions (Table 1), the vegetation and bare soil albedo coefficients (Section 2.2.1), and the multi-layer intermediate complexity snow scheme (Section 2.2.5) are the same in both versions.**

| Model Process | Model Version | |
| --- | --- | --- |
| | **2LAY** | **11LAY** |
| **Soil Moisture** (Section 2.2.2) | 2-layer bucket scheme – upper layer variable to 10cm depth and can disappear | 1D Richards equation describing moisture diffusion in unsaturated soils |
| **Maximum water holding (field) capacity** Section 2.2.2) | Constant (150kgm$^{-2}$) for all soil types | Derived using Van Genuchten (VG) relationships for characteristic matric potentials and vary with soil texture |
| **Runoff/Drainage** (Section 2.2.2) | When soil moisture exceeds field capacity 5% partitioned as surface runoff and 95% as groundwater drainage | Calculated soil hydraulic conductivity determines precipitation partitioning into infiltration and runoff. Drainage in form. Of free gravitational flow at bottom of soil. |
| **Bare soil evaporation resistance** (Section 2.2.3) | Based on depth of dry soil for bare soil PFT. Not optional – included by default | Empirical equation based on relative water content of the 1$^{st}$ four layers. Optional – not included by default |
| **Empirical plant water stress function, $\beta$** (Section 2.2.4) | Based on dry soil depth of upper layer | Based on plant water availability for root water uptake throughout soil column |
| **E and T over vegetated grid cell fraction** (Section 2.2.1) | Only T occurs | Both T and E occur over effective vegetated and effective bare soil fraction, respectively. Calculation of effective fractions based on LAI (Beer-Lambert approach) |

**Table 3: Soil moisture measurement depths (and corresponding model layer in brackets – see Table S1).**

|  | US-SRM | US-SRG | US-Whs | US-Wkg | US-Fuf | US-Vcp |
|---|---|---|---|---|---|---|
| **Soil moisture depths** | 2.5-5cm (5) | 2.5-5cm (5) | 5cm (6) | 5cm (6) | 2cm (4) | 5cm (6) |
|  | 15-20cm (7) | 15-20cm (7) | 15cm (7) | 15cm (7) | 20cm (8) | 20cm (7) |
|  | 60-70cm (9) | 75cm (9) | 30cm (8) | 30cm (8) | 50cm (9) | 50cm (9) |

5   **Table 4: Model evaluation metrics comparing the 2LAY and 11LAY daily upper layer soil moisture (re-scaled via linear CDF matching) and daily ET simulations to observations across the whole timeseries (where data present – see Fig. S2). Metrics include: correlation coefficient (R), root mean squared error (RMSE), mean absolute bias, and a measure of the relative variability, $\alpha$, between the model and the observations. The mean absolute bias = model – observations; therefore, a negative value represents a mean model underestimation of observed ET. $\alpha = \frac{\sigma_m}{\sigma_o}$ (see section 2.4, with 'ideal' values approaching 1).**

| Site | Model Version | Upper Layer (0-10cm) Soil Moisture R | ET R | ET RMSE $(mmd^{-1})$ | ET Mean Bias $(mmd^{-1})$ | ET relative variability, $\alpha$ |
|---|---|---|---|---|---|---|
| **US-Fuf** | 2LAY | 0.30 | 0.36 | 1.04 | -0.08 | 1.08 |
|  | 11LAY | 0.78 | 0.76 | 0.86 | 0.38 | 1.33 |
| **US-Vcp** | 2LAY | 0.27 | 0.26 | 1.39 | -0.54 | 0.79 |
|  | 11LAY | 0.37 | 0.59 | 1.02 | -0.27 | 0.82 |
| **US-SRM** | 2LAY | 0.52 | 0.53 | 0.84 | -0.03 | 0.70 |
|  | 11LAY | 0.85 | 0.84 | 0.53 | -0.07 | 0.87 |
| **US-Whs** | 2LAY | 0.56 | 0.54 | 0.68 | -0.03 | 0.67 |
|  | 11LAY | 0.90 | 0.85 | 0.43 | -0.02 | 0.89 |
| **US-SRG** | 2LAY | 0.48 | 0.52 | 1.02 | 0.01 | 0.70 |
|  | 11LAY | 0.67 | 0.88 | 0.57 | -0.11 | 0.90 |
| **US-Wkg** | 2LAY | 0.46 | 0.62 | 0.63 | 0 | 0.71 |
|  | 11LAY | 0.76 | 0.9 | 0.37 | -0.01 | 1.07 |

**Figure 1: Comparison of the 2LAY versus 11LAY mean daily hydrological stores and fluxes: i) evapotranspiration (ET, mmd$^{-1}$ – top panel); ii) total soil moisture (SM, kgm$^{-2}$) in the upper 10cm of the soil (2$^{nd}$ panel); iii) total column (0-2m) SM (3$^{rd}$ panel); iv) surface runoff (mmd$^{-1}$, 4$^{th}$ panel); v) drainage (mmd$^{-1}$, 5$^{th}$ panel); and vi) total runoff (surface runoff plus drainage – bottom panel). Error bars show the standard deviation for ET and SM, and 95% confidence interval for runoff and drainage. For soil moisture, the absolute values of total water content for the upper layer and total 2m column are shown for both model versions, i.e. the simulations have not been re-scaled to match the temporal dynamics of the observations (as described in Section 2.3.2); therefore, soil moisture observations are not shown. Observations are only shown for ET.**

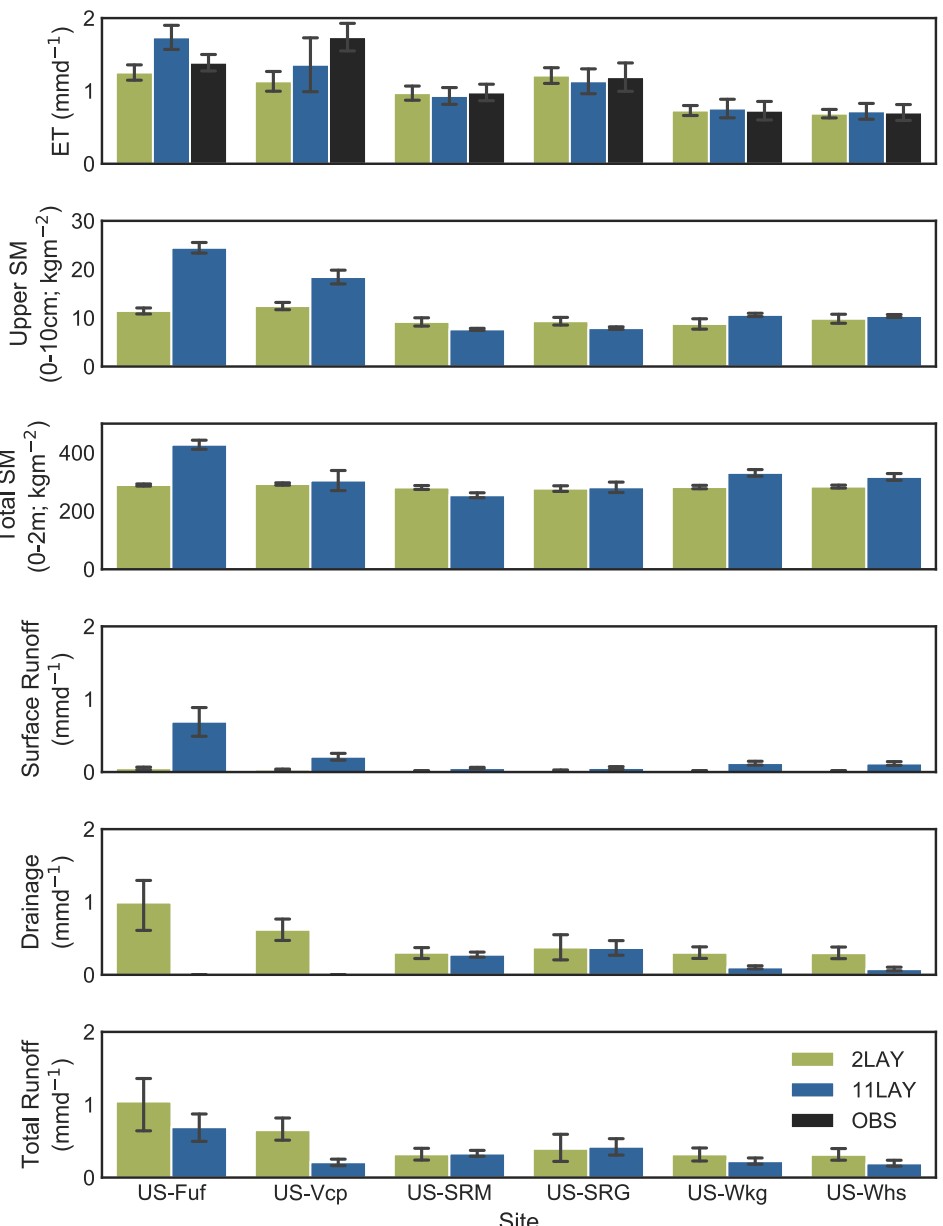

**Figure 2: Comparison of daily time series (for 2009) of upper layer soil moisture, surface water fluxes and related variables between the 2LAY (green curve) and 11LAY (blue curve) simulations. Changes between the two versions are shown for three sites representing the main vegetation types: left column = high-elevation tree-dominated site (US-Fuf); middle column = low-elevation mesquite shrub-dominated site (US-SRM); right column = low-elevation C4 grass site (US-SRG). At each site, top panel: LAI; 2nd panel: ET compared to observations (black curve); 3rd panel: bare soil evaporation; 4th panel: transpiration; 5th panel: empirical water limitation function ($\beta$) that scales photosynthesis and stomatal conductance; bottom panel: model soil moisture (re-scaled via linear CDF matching) expressed as volumetric soil water content (SWC) in the uppermost 10cm of the soil compared to observations (black curve). Precipitation is shown in the grey lines in the bottom panel for each site. (Note: full time series across all years are shown for all site in Figs. S2a-f). Light brown shaded zones show periods of maximum plant water limitation ($\beta$) at Santa Rita and consequent troughs in T and SWC.**

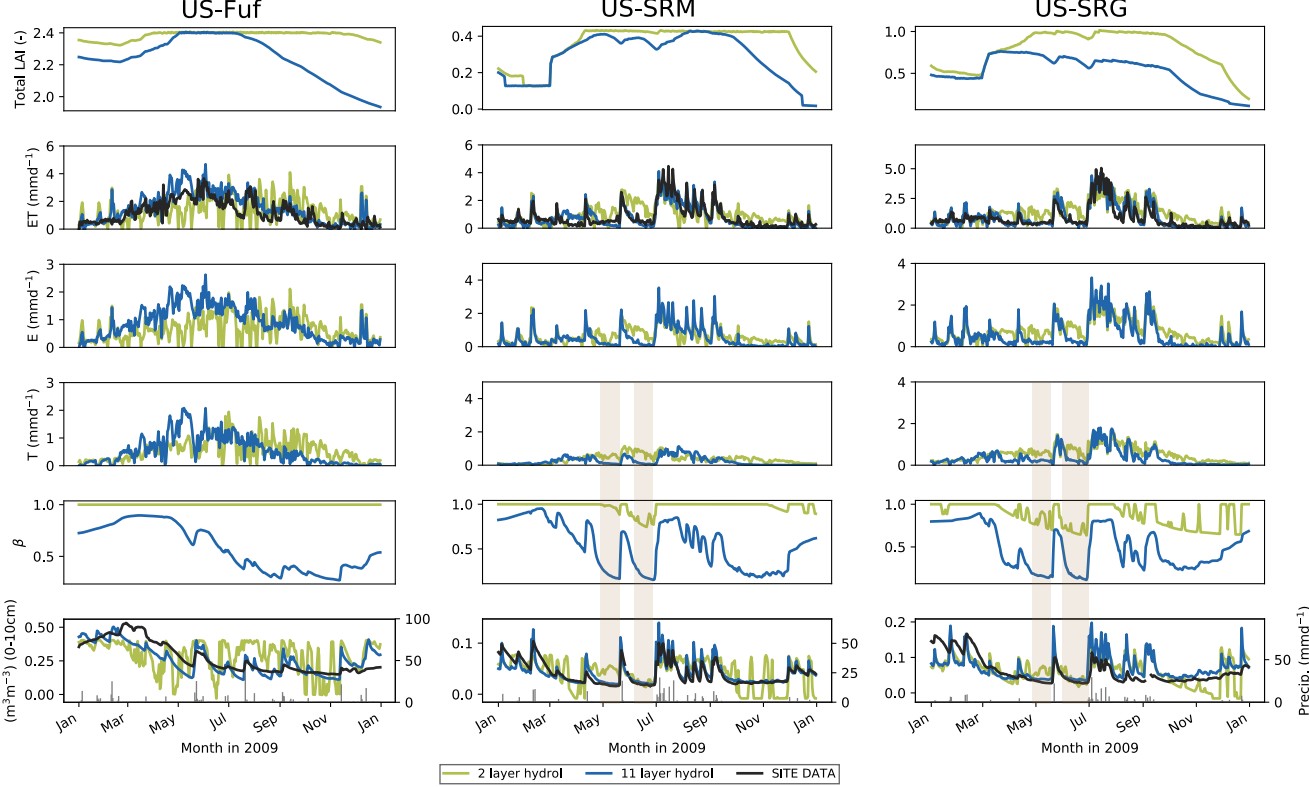

**Figure 3: Evapotranspiration (ET) monthly mean seasonal cycle comparing the 2LAY (green curve) and 11LAY (blue curve) simulations with observations (black curve). Individual site simulations have been averaged over the high-elevation tree dominated sites (left panel) and across all the low-elevation grass- and shrub-dominated sites (right panel). Units are mm per month (mm month$^{-1}$).**

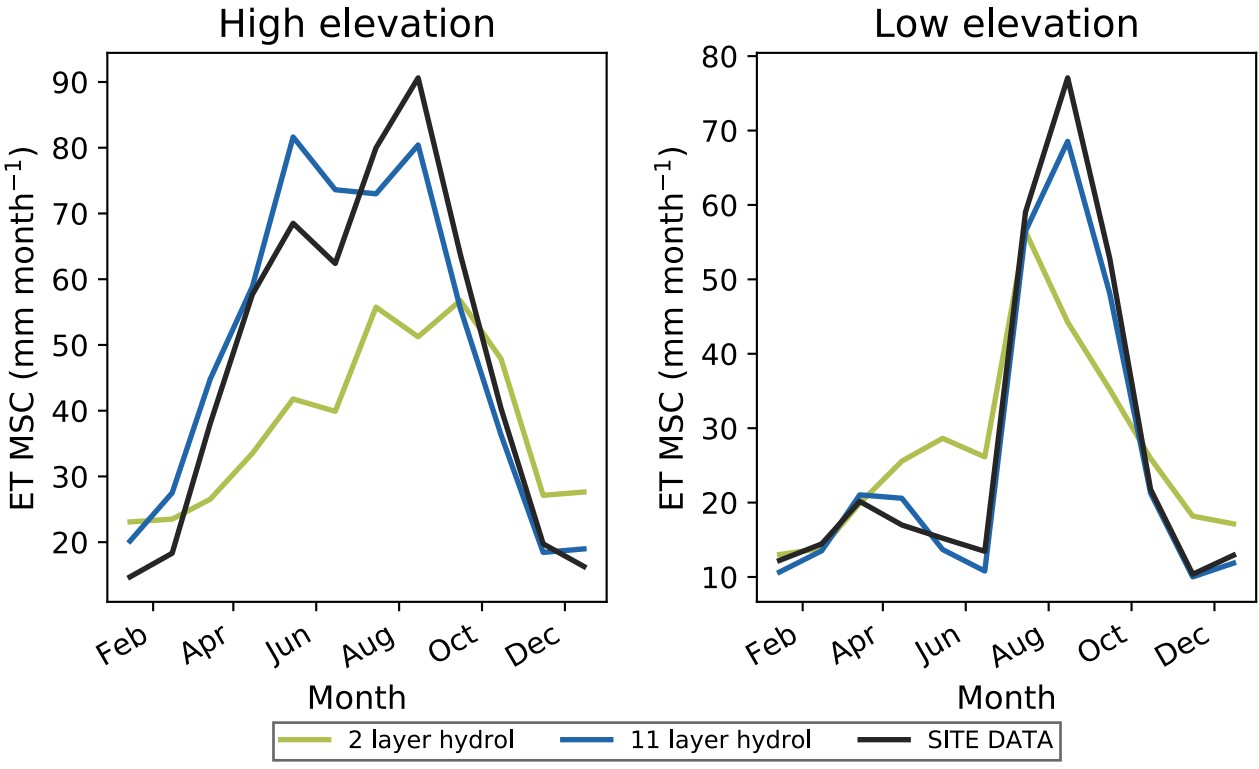

**Figure 4: Daily simulated volumetric soil water content (SWC – m³m⁻³) in 2009 (re-scaled via linear CDF matching) compared to observations at each site for three depths (upper, middle, lower) in the soil profile. The soil depths and their corresponding model layers are given in Table 3. Precipitation is shown in the grey lines in the bottom panel for each site.**

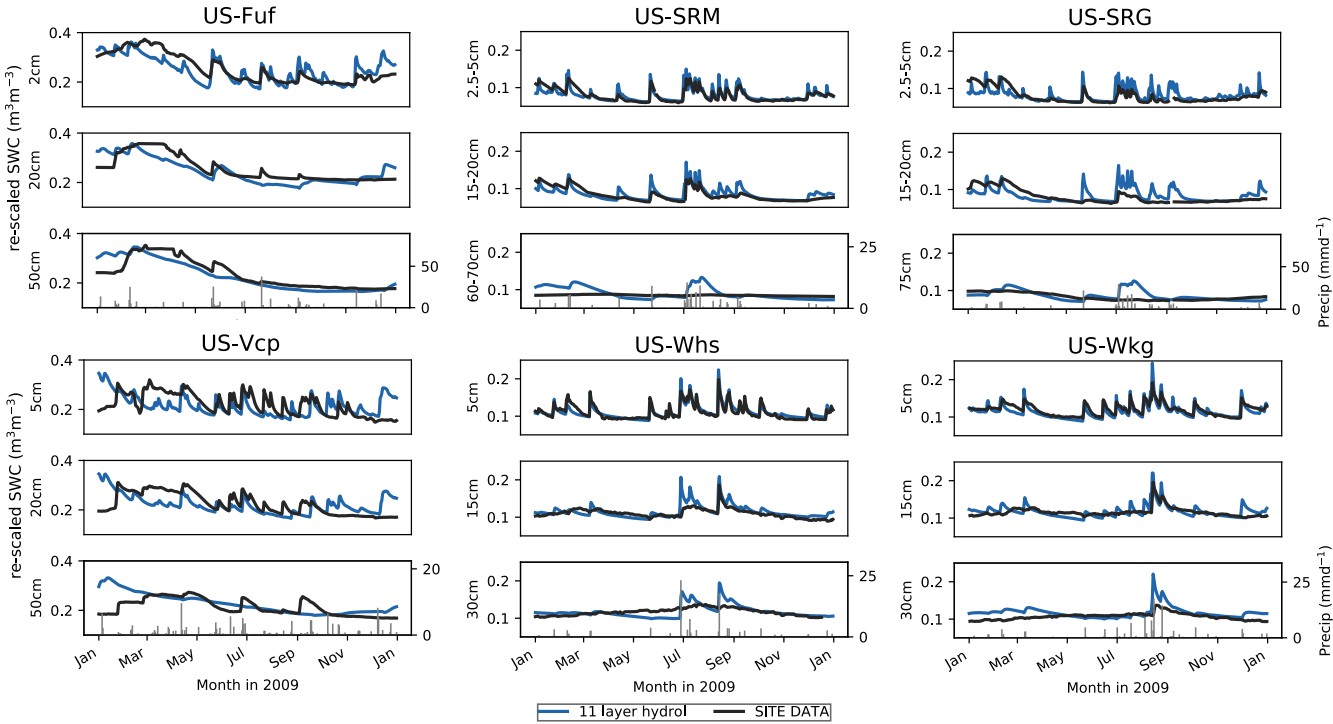

**Figure 5: a) US-Fuf and b) US-Vcp 11LAY (blue curve) daily time series (2007-2010) of model (re-scaled via linear CDF matching) versus observed volumetric soil water content (middle panel SWC – m³m⁻³) (black curve), compared to simulated snow mass (top panel) and soil temperature from the corresponding 2cm soil thermal layer (bottom panel). Snowfall is also shown as grey lines in the SWC time series. In the bottom panel the grey horizontal dashed line shows 0°C threshold. Light blue shaded zones show periods where the model overestimates the observations; light green shaded zones show periods where the model underestimates the observations.**

a)  **US-Fuf**

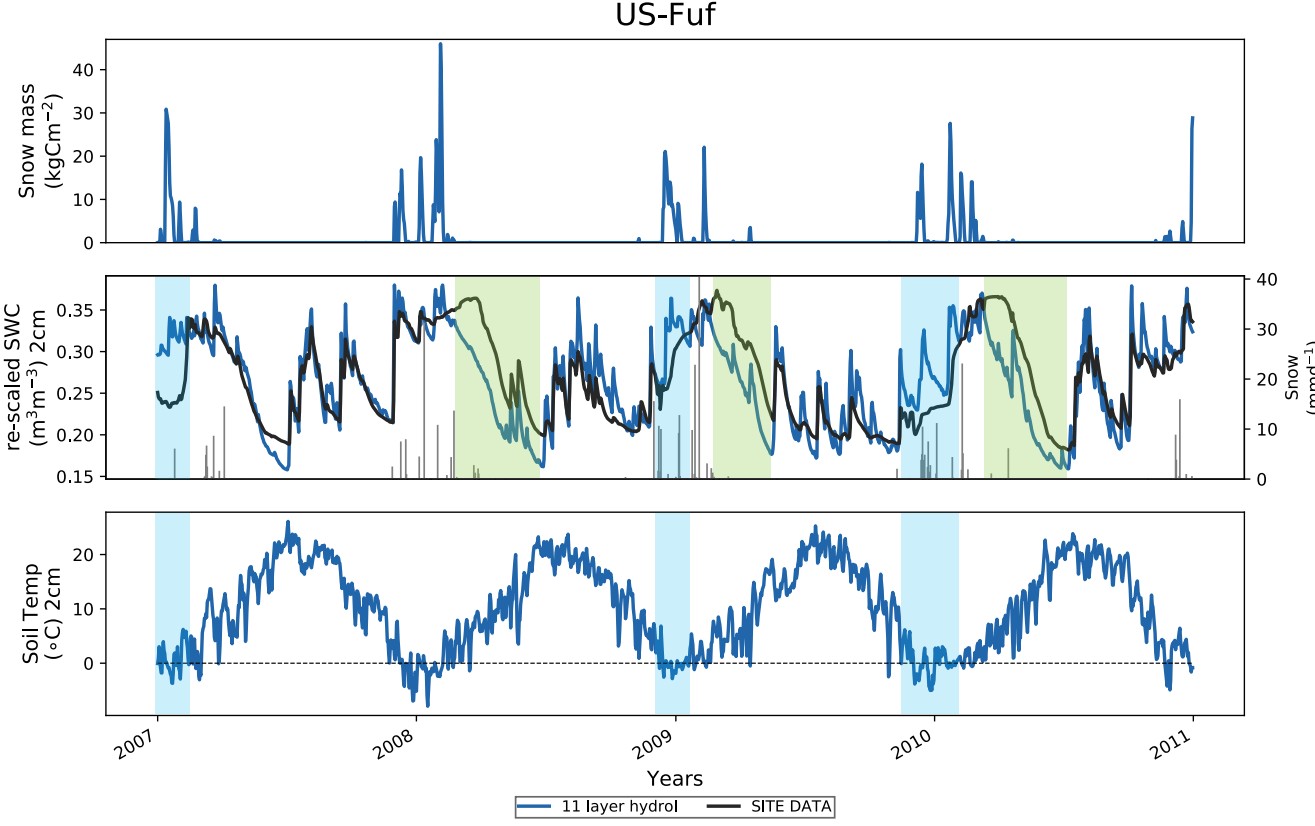

**b) US-Vcp**

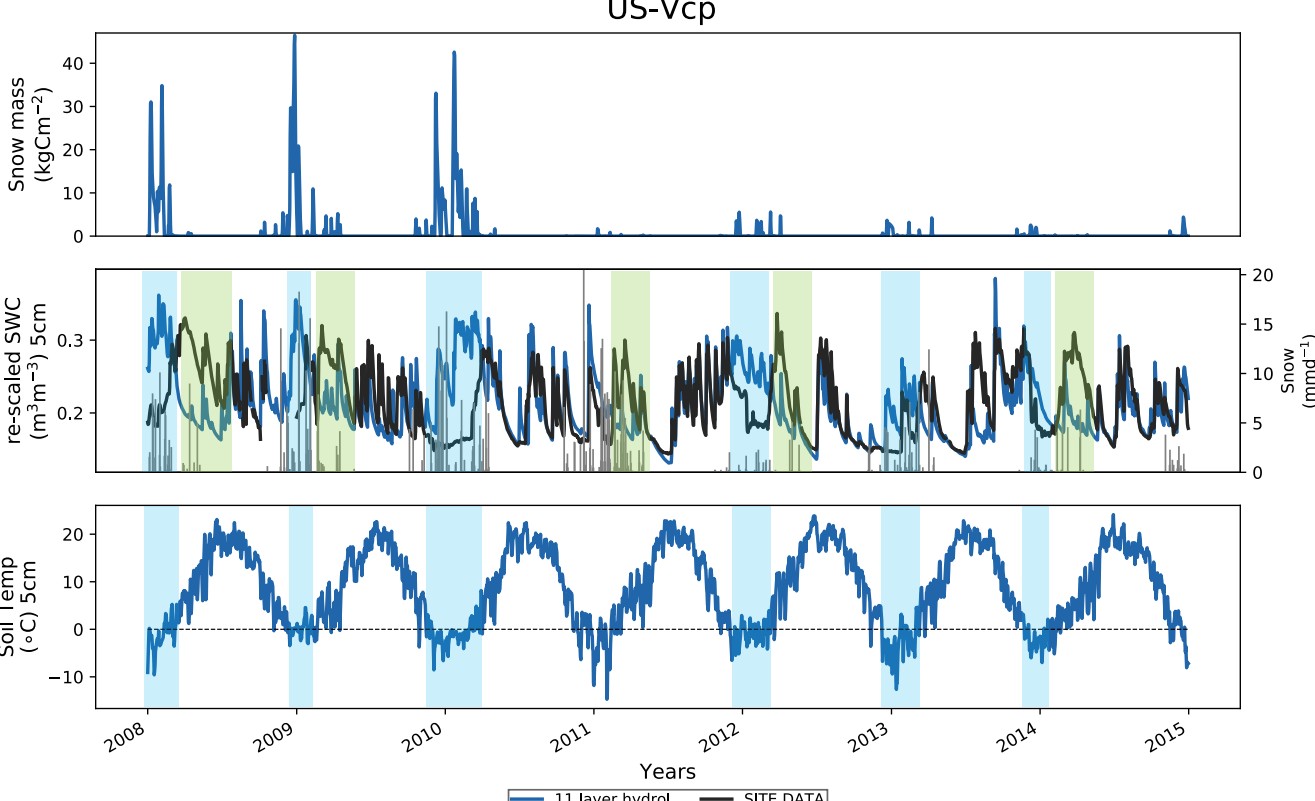

**Figure 6: Comparison of modelled and data-derived estimates of mean monthly T/ET ratios for each site. Forest site (US-Fuf and US-Vcp) T/ET estimates are derived using the method of Zhou et al. (2016 – Z16 – green curve). Monsoon low-elevation grass- and shrub-dominated site T/ET estimated are based on both Zhou et al. (2016) and Scott and Biederman (2017 – SB17 – orange curve). Blue curves show the model ratios at each site. Please see Section 2.3.1 for details on methods for data-derived T/ET estimates.**

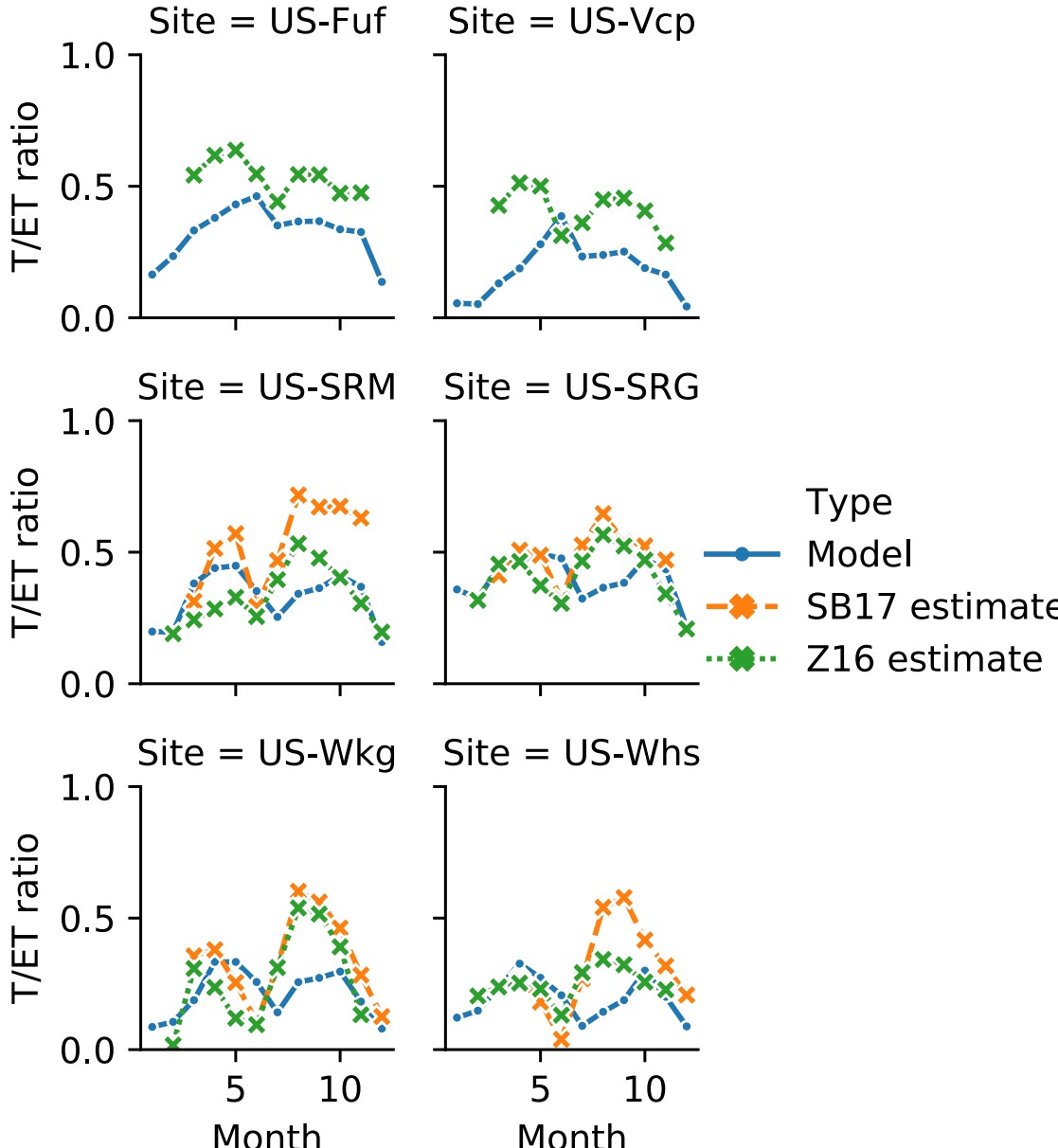

**Figure 7: Monthly mean seasonal cycle for ET, T/ET ratios, T and E averaged across all low-elevation grass- and shrub-dominated sites comparing the default 11LAY simulations (blue curve) with a simulation in which bare soil fraction is decreased (C4 grass cover increased (yellow curve). ET is compared to observations (black dashed curve) and T/ET ratios are compared to the data-derived estimates from Scott and Biederman (2017 – orange dashed curve) and Zhou et al. (2016 – green dashed curve). Units are mm per month (mm month$^{-1}$).**

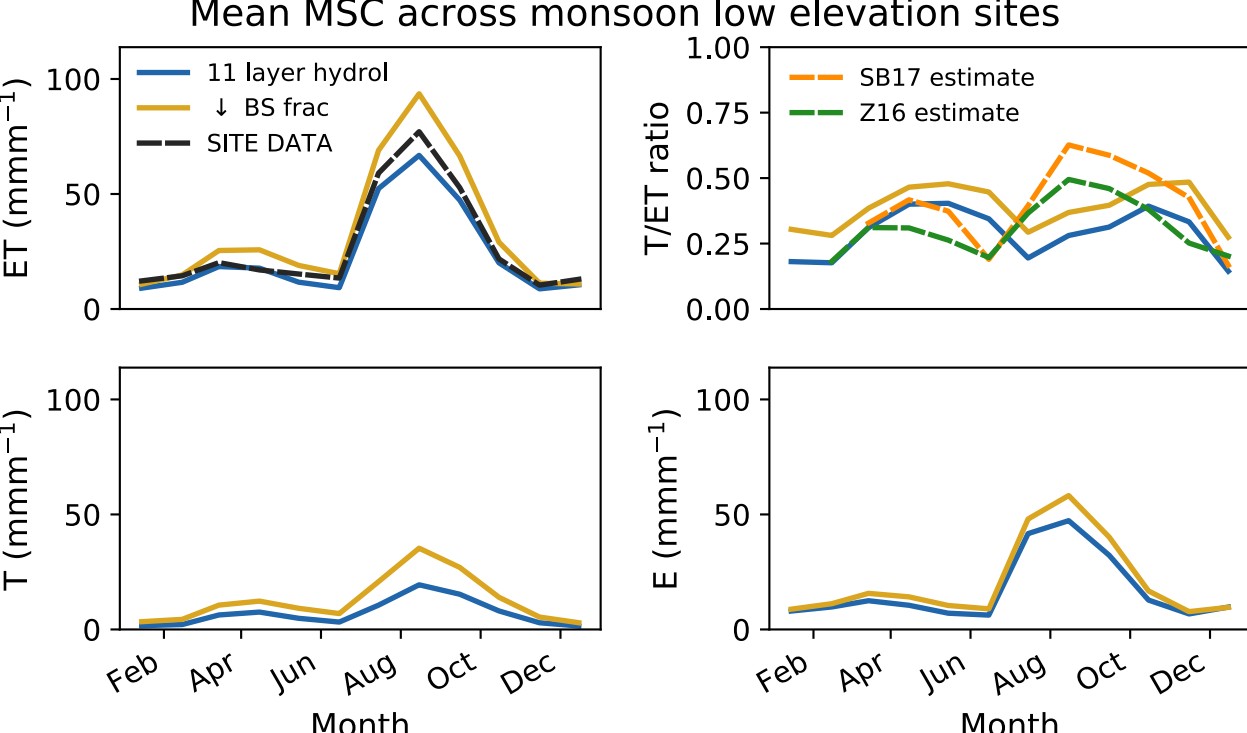

**Figure 8: Monthly mean seasonal cycle for evapotranspiration (ET), transpiration, T, and bare soil evaporation, E, averaged across all high-elevation forest sites (left column) and low-elevation monsoon grass- and shrub-dominated sites (right column) for the default 11LAY simulations (blue curve) compared to a simulation that included an additional bare soil evaporation resistance term (red curve). ET is also compared to observations (black curve). Units are mm per month (mm month$^{-1}$).**

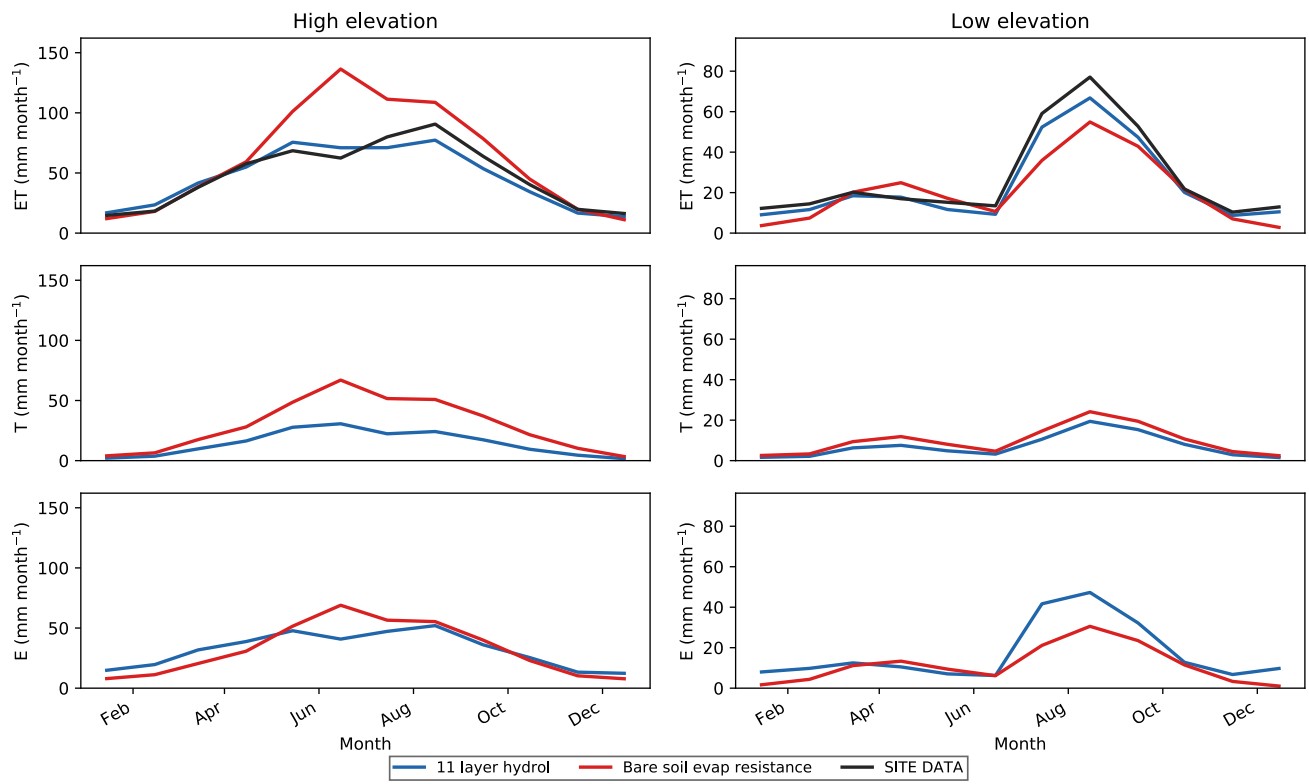

