# Peer review of "Multi-variable, multi-configuration testing of ORCHIDEE land surface model water flux and storage estimates across semi-arid sites in the southwestern US"

_Hydrology and Earth System Sciences, 2019_

## Referee Comment (RC1) · Anonymous Referee #1 · 20 Dec 2019

MacBean and colleagues compare the land surface model ORCHIDEE against six semi-arid flux sites, using the old 2-layer soil hydrology scheme and the new 11-layer scheme of ORCHIDEE.

The study is certainly done correctly and the comparisons are fine. Specific remarks and questions are below.

However, one asks him/herself why one needs another validation of a Richards model in an LSM, showing that it performs better than on old bucket or 2-bucket version?

Specifically the multi-layer soil model of ORCHIDEE was tested quite a number of times already.

But semi-arid ecosystems are interesting because quite a few model assumptions of LSMs get challenged there. Unfortunately the paper does not talk about it nor tries to advance in this direction.

For example, ORCHIDEE uses tiles or fractions to deal with different land cover within one grid cell. To my knowledge, if a grid cell is vegetated then there is only transpiration (T). Evaporation (E) is from a special bare soil fraction only. There is no below-canopy E, which experiences lower wind speed, higher humidity and a litter layer compared to bare soil. This might have changed in the 11-layer version. Would be interesting to know. If the bare soil fraction mimics below-canopy E, then it is just a modelling concept and should be treated like this.

Semi-arid ecosystems are probably the only ecosystems where this model structure is valid for soil evaporation. However, the rest of the model structure with fractions comes to its limits. If there is a shrub-encroached grassland, the shrubs (trees in this study) get all crammed into a small tile, shading each other and competing for soil moisture. Or is there a gap fraction in ORCHIDEE? Does it allow for shrub (tree) roots to forage in the grass tile? The grass in semi-arid ecosystems dies off during the year. This changes the LAI as discussed in the paper. But does the grass fraction stay constant? Should LAI rather stay constant in the grass tile but the tile should shrink, leading to more bare soil fraction? I think that one cannot discuss semi-arid ecosystems without talking about vegetation (dynamics). The $CO_2$ fluxes could be interesting in this respect as well. They are omitted in the current paper.

The paper discusses quite a few shortcomings of ORCHIDEE, or even LSMs in general. But there is no assessment of the importance of each point. They all seem to be similar important. I would have loved to see either prioritisation for model development or at least a guidance to the reader how to evaluate model shortcomings. The model

might already be fine from an atmospheric perspective, or it might lead to a wet bias in spring.

Specific remarks are:

- I would change the tile. "Multi-variable" and "flux and storage" is tautologic. "Multi-configuration" is a bit much for two configurations.

- You should only cite one paper in preparation for CMIP6 and not once Ducharne et al. (in prep.) and once Peylin et al. (in prep.).

- There are three personal communications, which are all from co-authors. Which co-author talked to which co-author?

- The description of "Richards and Darcy's equation" is strange. Darcy is part of Richards. The description is strange at two places (l.110 and l.211ff). I think that Richards equation is known sufficiently so it is only interesting which form is solved, the saturation-based or the head-based form.

- If LAI was identified to be important why is no local LAI data used? I found local LAI data in Scott and Biederman (2017) for some of the sites.

- Why are different T/ET algorithms used for different sites?

- T/ET is seen as a measurement in the manuscript. But it is not. Any validation is missing in the Scott and Biederman (2017) paper, because it is pretty impossible to validate it. So T/ET should be seen only as an estimate. There are quite some algorithms in the literature to calculate T/ET and it is hard to tell why one should be more correct than the other.

- l.255: what is the subscript j on $c_j$?

- l.255ff: R(z) is explained but not $n_{root}$. If $n_{root}$ were explained then one does not have to (confusingly) start the sums from 2 because $n_{root}=0$ in $\nu=1$ and i=1.

- l265ff: Why is the relative water content weighted with $n_{root}$? This formulation is an empirical observation and the beta term is never weighted by root length density (or similar) in the data papers (e.g. Keenan et al. (Biogeosci 2009)).

- l.268: Should W be in kg/m$^3$ instead of kg/m$^2$? Why is W used and not volumetric soil moisture theta?

- l270ff: Why is p% = 0.8? There is quite some literature that it should be around 0.4 (e.g. Granier et al. (AFM 2007)), at least for forests?

- l.276f: The references are missing. And only the Keenan et al. paper actually supports this claim. The Zhou et al. papers do something very different and act only on stomatal conductance.

- l.303f: I wondered if this claim means that you have a near perfect energy balance closure?

- l.315f: why are there no site-specific soil characteristics? They must have been done at some point in the past.

- Fig. 1: Where are the observations?

- Fig. 1: Harmonise scales of ET, Runoff and Drainage, as well as of Upper SM and Total SM so that one can compare the fluxes/stocks. For example, why is Total SM up to 1000? If kg/m$^3$, then Upper SM and Total SM could have the same scale. If kg/m$^2$, they should be scaled according to layer depth.

- Fig. 1: Why is there (almost) no drainage at forested sites with the 11-layer version? Is this realistic? There is only a very small mention for US-Fuf in the text.

- Fig. 2: I think the titles of the y-axes of row 3 and 4 are swapped.

- Fig. 4: please put the 2 cm, 20, cm and 50 cm plots on the same scales.

- Fig. 5b: Data stays low during much of the snowfall period. This can happen if the data is measured inside a forest whereas the model assumes open space. Much of SnowMIP's model intercomparison, at which ORCHIDEE probably participated, focussed on open sites. We might not know well the behaviour of our models at forest sites.

  It looks like that the data is even decreasing at the beginning of the snowfall period. This could point to soil freezing. Some soil moisture sensors measure only liquid water, so low values are measured during frozen soil conditions. So sites also do not include possible ice phases in their transformations from voltage to soil moisture.

  Both processes were not discussed.

- l.384f: This is a "false friend" to me. Evaporation is water vapour but the Richards equation (as used in ORCHIDEE) does not include vapour transport in the soil. So the model has to compensate for this omittance. This is one of the primary reasons why the Richards solvers need very thin layers at the top of the soil. These layers cannot be seen as physical layers because they have to compensate for all the model deficiencies on top of possible litter layers. It is thus doubtful that these first few layers should be compared satellite measurements.

- l.396: Isn't his a contradiction to Whitley et al. (2016). You state in the introduction that Whitley et al. (2016) found that T of the vegetation is mostly too low in

the models. Is 2-layer ORCHIDEE different so that 11-layer ORCHIDEE can decrease T during the warm season?

- l.448ff: There also seems to be a problem with infiltration. At the model attenuates precipitation peaks too much at forest sites, while it is almost not attenuating at the grassland sites. Could you explain that please. There seems to be a difference in the model why water can flow quickly to deep layers in grassland but not in forests. Or is it the bare soil fraction?

- l.454: I was wondering why the model was not tested with more layers, say 100?

---

## Referee Comment (RC2) · Anonymous Referee #2 · 10 Feb 2020

Review of **Multi-variable, multi-configuration testing of ORCHIDEE land surface model water flux and storage estimates across semi-arid sites in the southwestern US** by MacBean et al.

The manuscript by MacBean et al. deals with two different soil schematizations of the ORCHIDEE land surface model. One model set-up consists of a 2-layer soil schematization, whereas the other set-up makes use of an 11-layer soil scheme. In addition, resistance for soil evaporation was varied and bare soil fractions were reduced. The model set-ups were evaluated for several sites in the southwestern US. The authors show that adding a more detailed soil schematization improves the model results, especially regarding total evaporation and high frequency moisture dynamics.

The manuscript is generally well-written, and the figures are clear and of high quality. Most of the statements are supported by the data, and I think the article is interesting, because I agree that the hydrology in LSMs deserves attention. Nevertheless, after reading the article, I have several questions that remain.

One of the first things the authors observe is that the forested sites show differences in transpiration and soil moisture. The soil schemes are different between the model runs, but rooting depths and rooting profiles are hardly mentioned by the authors. However, different rooting depths for both set-ups will have a strong influence on the findings of the authors.  So how are these parameterized and are these different for the different model set-ups?

Similarly, the authors also often refer to the low and high elevation sites, but these also come with different vegetation types (forested vs grass/shrubland). I think the different vegetation types are much more the reason for the differences between the different sites, so I suggest that the authors distinguish more between the different vegetation types instead of the elevation, especially in the figures.

The authors also decided to model the soils with a thickness of 2 m, and mention that for the 11LAY-model drainage occurs as free gravitational flow at the bottom of the soil. This thickness, which is rather arbitrary, will also have a strong influence on the results as presented. The groundwater tables may influence the soil moisture profiles, and I wonder therefore if the authors have some idea on the groundwater tables at these sites. I do not object to this model choice of a 2 meter soil thickness, as you probably have to make an assumption here, but I believe it would be good to reflect on it, especially as the goal of the authors is to get the hydrology right, from which the groundwater is an important aspect and that is now basically assumed to be negligible.

There are also two methods used to derive ratios of transpiration/evaporation (Figure 6), but also here I have several questions. First, I wonder what the difference is between the two methods and if it is a fair comparison. There is also no data in the first months, and no data for US-Vcp, why is that? In addition, at US-Fuf, the data-derived estimates show that almost half of the total evaporation is transpiration, even during winter. At the same time, the site is described as having snow, at a high elevation, and one would therefore expect hardly any transpiration in winter here. This is also what the model actually does, it shows a strong reduction during winter. So how reliable are the estimated observations here?

The authors often argue that snow is not correctly modelled, and I think the statement of the authors on page 14, lines 442-444 is important here. Snow usually falls within a temperature range around 0 degrees Celsius, and the authors mention that the results improved by changing the temperature threshold, but these results are not shown, so please add these results.
In addition, the reasoning of the authors regarding the snow modelling relates to the overestimation of ET at US-Fuf for 11LAY, but this does not happen for 2LAY.  At the same time, US-Vcp also shows an

underestimation and has snow, so it does not seem to be a consistent problem here. Do the two model set-ups use the same snow module and are the parameterizations the same for the different sites? As suggestion, it could also help the authors to look at remotely sensed snow cover products such as MODIS10A. These products are relatively easy and could provide already a quick check if the snow temporal dynamics are captured in the model.

My most important point relates however to the fact that the article misses sometimes a bit focus regarding the goal of the authors, which is comparing a simple two-layer scheme with a more complex scheme in order to improve the hydrology. A couple of times the authors only look at the 11LAY-results, or do not use observations to assess if there are any improvements. For example, the authors only compare 11LAY with the soil moisture measurements (Fig. 4,5, paragraph 3.2). I do understand why, as the authors explain this in paragraph 2.3.2, but I am not sure if there is any point in evaluating 11LAY-results with soil moisture data, if you can not do the same for 2LAY. After reading paragraph 2.2.2 I still think the authors could at least compare also the temporal dynamics in the 2LAY-model, as this is what the authors do anyway with equation 5.
Similarly, a large part of paragraph 3.1 gives a description on the differences between the two model set-ups, and discusses Figure 1. Nevertheless, without any idea on how reality looks like, it is hard to really get an understanding on what is actually better. So I am not sure if this part of the paragraph really adds something, unless the authors add some observations. The authors do have soil moisture data and flux tower data, so I suggest to add these to Figure 1.
One of the main conclusions is also that the high frequency soil moisture dynamics are more realistic for the 11LAY-model. This conclusion is however not supported by the data as shown, there is no figure in the manuscript and supplementary material that actually compares both 11LAY and 2LAY soil moisture values with observations, so you can unfortunately not state that 11LAY is clearly better here. The conclusion that surface runoff is more realistic (P21.L669) came even as a bigger surprise to me, I believe there is no data on surface runoff in the manuscript, or I must have completely missed this.

Concluding, the manuscript is interesting, but the authors should make sure they build a systematic case why one hydrological schematization should be preferred over another. I have sometimes the feeling the authors have a preference for the 11LAY-scheme, but I think it is important to objectively assess the performance of both set-ups. I hope my comments are useful for the authors and look forward to an improved manuscript.

**Minor comments**
P1.L36. Results better → results in a better?
P2.L62. A evaporation → an evaporation
P3.L79 have been rarely been → have rarely been
P4.L115. Define PFT
P6. L187. What do you mean with soil tile? The spatial distribution of different soils within a grid cell?
P6.L189. "all three PFT's" –> It is mentioned before that there are 12, so why three now?
P6.L191. Related parameters) → remove ")"
P7.L210. At al → et al
P7.L217. At al → et al
P8.L227. Seems a bit arbitrary to me, why these numbers?
P8.L229. Has been test → have been tested
P8.L256. The the root density → the root density
P8.L256-257. Why these values? What are they based on?
Eq3. Please define and describe also h_t and d
P8.L267. Is T here transpiration? Please define.

Eq5. Please define your variables

P12.L351. Higher compared to the other sites? It is not higher than the 11LAY-scheme.

P12.L380. I do not see any values going to 0 in Figure S1 for VWC in the upper 2m. Basically 2LAY seems to drain the upper layer faster.

P12.L383-384. I do not think you can conclude 11LAY is better based on the data as shown, there are no observations shown of soil moisture in Fig. 2.

P14.L421. Fig 4 →Fig. 4

P14.L422. So which sites in fig4 do you mean? It's easier to add the names, then the reader knows where to look.

P14.L445-448. Where can I see this? Please make sure you back up your conclusions by showing the evidence.

P15.L460-480. I was a bit confused by the term evaporation E, whereas you also discuss evapotranspiration ET (which are often used interchangeably), but you mean here interception evaporation, correct? For clarity it might be good to add a subscript Ei and talk about interception evaporation.

P15.L467. You mention before that US-Vcp underestimated ET, instead of overestimated.

P16.L480. Are be responsible –> are responsible?

P17.L517. You do not show that T/ET fractions are better with the reduced bare soil fraction.

P17.L523. TeNE-forest?

P17.L529. Spring → spring

P19.L592. ORCHIEE → ORCHIDEE

P21.L669. I am not sure how you can conclude this without runoff data and never evaluating it.

Table3. Please note that RMSE also has a unit

Figure 3. The unit is mmm-1, I believe you mean mm/month, but please make this clearer.

Figure 6. Why not include also the 2LAY-estimates? There are two methods used to estimate the ratios for the high and low elevation sites, is this a fair comparison then? Why is there no data for the first months? Why no data for US-Vcp?

Figure 7. Why would you average over all the sites? This is just removing information, please show all sites individually, there is no point in lumping this together.

Figures S5 and S6. Please add units and a legend. And as these are regressions, why are there no data points shown? I only see a regression line, so I am not sure how to interpret these figures.

Data availability:
Where are the model results shared?

---

## Short Comment (SC1) · 17 Feb 2020

We are grateful to anonymous reviewer 1 for providing us with such a thoughtful and useful review. Here, we briefly respond to some of the major issues raised by reviewer 1 in the hope of some further discussion before the interactive public discussion period ends on 23rd February. We will then provide a detailed response to reviewer 1 along with a revised manuscript.

We absolutely agree with the reviewer that many interesting aspects related to hydrol-

ogy of heterogeneous semi-arid ecosystems were not either a) detailed in the model description and/or b) not elaborated on in the discussion. We will address these points thoroughly in our revised manuscript and in our detailed response to this review that we will submit after the discussion period has closed.

Another point raised was why we compared the 2-layer vs 11-layer model given that has already been examined a lot in the literature. We agree with this point and in fact there was some debate about this amongst ourselves. It is true that most land surface models do now have a more mechanistic Richards' equation-type approach to modeling soil moisture dynamics. It's also the case that it is hard to compare the 2-layer and 11-layer approach given how different the representation of soil hydrology is and that it is very difficult to compare the 2-layer version to observations (much harder than the 11-layer). However, despite these considerations we decided to keep the 2-layer vs 11-layer comparison in an initial first part to the results because of the following reasons: firstly, we are expecting that not all readers are land surface modelers and some of those people might either not be familiar with simple bucket models or they might be users of hydrological or other types of models that still use a simple bucket scheme. For these readers we wanted to show once again that the bucket model really does not represent the temporal dynamics of the soil moisture or ET well; therefore, they should actively not trust ET predictions from any model that uses these types of soil hydrology schemes. Secondly, the ORCHIDEE model CMIP5/IPCC AR5 simulations were based on the 2-layer version of the hydrology model. While this was a long time ago now and the CMIP6 simulations are being released, many people are still using CMIP5 to study various aspects of earth system processes, climate change impacts or to understand model deficiencies. Given the fact that CMIP6 results are $\sim$1 year delayed, we expect that people will continue to use CMIP5 simulations for at least another year. Therefore, we explicitly wanted to mention that the ORCHIDEE CMIP5 ET predictions might not be as accurate as previously thought for semi-arid regions, with consequences for predictions of other variables. The reviewer specifically mentioned CO2 fluxes for example, and with good reason. In fact, this paper is part of a

series of papers that are addressing multiple aspects of modeling semi-arid ecosystem functioning, including CO2 fluxes. We have chosen not to include any aspect related to CO2 fluxes in this paper because the model evaluation has pointed to more significant model deficiencies that we are currently trying to address – so this will be the subject of a forthcoming follow up paper. If the reviewer is interested, this work was presented at AGU last year: https://agu.confex.com/agu/fm19/meetingapp.cgi/Paper/489913 and https://nmacbean.files.wordpress.com/2020/02/nmacbean_agu2019_swusnee_poster.pdf.

With all this being said, we are inclined to keep the comparison between the 2 vs 11 layer, but in our revised manuscript and detailed response to the reviewer we will attempt to outline this reasoning more clearly, including pointing out other fields/models that are still using bucket layer schemes. We will also de-emphasize the first part (2 vs 11 layer comparison) and instead emphasize that the second part (11 layer comparison to observations and remaining model-data discrepancies) is the key part of the paper, particularly in terms of land surface modeling. This will include a more comprehensive discussion of aspects of the 11-layer model that don't address some of the semi-arid issues outlined in this review. In the meantime, we welcome reviewer 1's thoughts on our reasoning behind retaining the 2 vs 11 layer comparison as part of this paper.

One final point we'd like to raise at this point in the review processes related to reviewer 1's specific comment "If LAI was identified to be important why is no local LAI data used? I found local LAI data in Scott and Biederman (2017) for some of the sites." In fact, in Scott and Biederman (2017) it is MODIS LAI (satellite-derived) that is used. Unfortunately, to date we do not have any timeseries of local LAI measurements related to specific vegetation types. This would be extremely useful – as the reviewer points out. The MODIS LAI cover 250m or more and are therefore considered landscape scale estimates – therefore, we cannot use them to validated specific PFT LAI simulations from the model. Furthermore, while the timing of satellite LAI estimates generally agree, the absolute magnitude of different satellite-LAI products varies widely. This is due to differences in the retrieval algorithms used to infer LAI from the raw radiance

data (e.g. D'Odorico et al., 2014; Garrigues et al., 2008; Pickett-Heaps et al., 2014). We were hesitant to use these data for these reasons; however, we will re-think this decision when we revise the manuscript, specifically in terms of whether it might be useful to normalize the satellite (and model) LAI and only consider their temporal dynamics.

Once again, we thank reviewer 1 for their detailed review and we apologize for not leaving much time in the interactive discussion period to reply to this initial informal response to their review. Otherwise, we look forward to providing a detailed comment-by-comment response to their review after the interactive discussion period has closed.

References: D'Odorico, P., Gonsamo, A., Pinty, B., Gobron, N., Coops, N., Mendez, E., and Schaepman, M. E.: Intercomparison of fraction of absorbed photosynthetically active radiation products derived from satellite data over Europe, Remote Sens. Environ., 142, 141–154, doi:10.1016/j.rse.2013.12.005, 2014.

Garrigues, S., Lacaze, R., Baret, F., Morisette, J. T., Weiss, M., Nickeson, J. E., Fernandes, R., Plummer, S., Shabanov, N. V., Myneni, R. B., Knyazikhin, Y., and Yang, W.: Validation and in- tercomparison of global Leaf Area Index products derived from remote sensing data, J. Geophys. Res.-Biogeo., 113, G02028, doi:10.1029/2007JG000635, 2008.

Pickett-Heaps, C. A., Canadell, J. G., Briggs, P. R., Gobron, N., Haverd, V., Paget, M. J., Pinty, B., and Raupach, M. R.: Evaluation of six satellite-derived Fraction of Absorbed Photosynthetic Active Radiation (FAPAR) products across the Australian continent, Remote Sens. Environ., 140, 241–256, doi:10.1016/j.rse.2013.08.037, 2014.

---

## Short Comment (SC2) · 17 Feb 2020

We are grateful to anonymous reviewer 2 for their detailed review and for raising inconsistencies or points to further consider. Here, we briefly respond to some of the major issues raised by reviewer 2 in the hope of some further discussion before the interactive public discussion period ends on 23rd February. We will then provide a detailed response to reviewer 2 along with a revised manuscript.

We agree with all points raised by reviewer 2 and will address all of these in our de-

tailed comment-by-comment response and revised manuscript after the interactive discussion period has closed.

At this point, we would like to further discuss the main point that reviewer 2 raised, which relates to the comparison between the 2 vs 11-layer hydrology schemes. We agree that the the way it was presented leads to some confusion and difficulty in comparing the two schemes – for example, why we do not compare the 2-layer version to observations. As reviewer 2 stated, we did provide the reasoning behind why we only compare the 11-layer version to observations (and why we perform the CDF matching). However, reading what we wrote in Section 2.3.2 again we realize that perhaps we have not gone into enough detail as to why we did not compare the 2-layer to observations. In the 2-layer scheme, the top layer depth is variable. It can be a maximum of 10cm and it can also disappear completely if the water empties into the bottom layer that makes up the rest of the 2m soil column. Therefore, despite any of the other issues that prevent a comparison of the absolute modeled and observed soil moisture values (and why we use CDF matching), for the 2-layer version we also have the issue that the depth at which we are simulating the upper soil moisture in the 2-layer version is changing over time. Therefore, it makes the comparison to moisture observations extremely difficult – and crucially – this feature of the 2-layer version means our 2-layer model-obs comparison will be different to the 11-layer model-data comparison in which we can at least match the model layer to the depth at which the observations were taken. Therefore, we are still hesitant to compare the upper layer soil moisture simulated by the 2-layer version to observations, even after CDF matching. We did have a version where we normalized the 2-layer model and observed upper layer moisture to just compare their temporal dynamics, but it becomes very confusing when we use a different processing method to compare the 11-layer model to observations. Instead, we chose to have a separate initial results section where we just described the differences between the 2-layer and 11-layer version soil moisture and only discussed the model performance in relation to ET observations. We acknowledge however that the statement that soil moisture appears to be more realistic is unwarranted. We will

change the text so that we only say that the ET is more realistic (given that both versions of the model ET can be directly compared to ET observations). We welcome reviewer 2's thoughts on not including a comparison of the 2-layer model upper layer soil moisture to observations (for reasons given above) and on our proposed changes to the text related to only discussing performance in relation to ET.

Finally, on this point, we have provided a detailed explanation for why we include the 2 vs 11-layer comparison in our initial informal response to reviewer 1 and we would welcome reviewer 2's thoughts on this justification as well. Does reviewer 2 think the manuscript would be vastly improved if we were to remove the comparison between the 2- and 11-layer versions of the model and only examine the 11-layer model? Or would it be enough to de-emphasize this comparison and instead emphasize more the second part in relation to the 11-layer comparison and remaining model-data discrepancies?

We thank reviewer 2 again for their detailed review and we apologize for not leaving much time in the interactive discussion period to reply to this initial informal response to their review. We will provide a much more detailed comment-by-comment response to their review after the interactive discussion period has closed that will take all points raised into account.

---

## Referee Comment (RC3) · Anonymous Referee #2 · 20 Feb 2020

I would like to thank the authors for their openness and the discussion, below I tried to reply to their questions in the informal response.

I can see the difficulties the authors raise with regard to comparing the 2LAY and 11LAY soil moisture values, and also understand why the soil moisture values are not compared to observations for the 2LAY-model. For me, it is not a problem that you cannot use the 2LAY-values, but I just wonder what the point is of comparing 11LAY-results with soil moisture if you cannot do the same for the 2LAY-model. This also depends on the goal of the comparison, because you cannot use it to assess which
of the models is better (which I believe is the main goal of the paper, and also how I interpreted this section). I believe it could serve as an explanation why the ET-values are better, but some textual changes may be needed to clarify this. In the current version, this comparison seems rather important, and relates to some conclusions, whereas it is merely an additional and supportive explanation for some other more important findings.

Regarding the second point of the authors, and I am sorry for not making it easier, but I strongly disagree with reviewer 1 that you should remove the 2-layer versus the 11-layer comparison. This is for me the key-point of the manuscript, and this relates also to my comment in my review that the authors sometimes show already a preference for the 11-layer model. It is not carved in stone that a more detailed model is better, and it should objectively be assessed which one is better. Even though reviewer 1 points out that more detailed Richards' equation approaches often improve LSMs, there is also an important reason bucket-type models are still often used especially in catchment hydrology. The Richards' equation approach does not include macro-pores, which in more sloped areas plays an important role. In addition, the parameterization often assumes a homogene soil, which is also not true. The fact that LSMs often perform better with Richards' approach also relates to how they are parameterized, bucket-type models need actually calibration as the parameters are less physically based, whereas the Richards' approach uses more physically based soil parameters that are often measured. In general, the hydrological schematization in LSMs is in my view still rather poor, even with more detailed Richards' equation approaches, whereas it actually has a strong influence on the outcomes of the models, so I believe it is important that the authors show this. In addition, for a strong modelling experiment, you always need a benchmark, which is here the 2-layer model. Leaving it out leads to a manuscript that is just a model application, and the reader can never see what the 11-layers actually add.

I hope my thoughts are useful, even though it is probably not making it easier. I still look

forward to an improved manuscript and hope to authors find a good way to address all the issues of myself and reviewer 1.

---

## Author Comment (AC1) · 9 Apr 2020

Response to Reviewer #1

MacBean and colleagues compare the land surface model ORCHIDEE against six semi-arid flux sites, using the old 2-layer soil hydrology scheme and the new 11-layer scheme of ORCHIDEE.
The study is certainly done correctly and the comparisons are fine. Specific remarks and questions are below.

We thank anonymous reviewer 1 for providing us with such a thoughtful and useful review. We provide more detailed comments to all of their comments and suggestions below. Please note that responses to the reviewer are in blue and additions to the manuscript are in red. Small changes to existing sentences are given in italics within the original sentence.

However, one asks him/herself why one needs another validation of a Richards model in an LSM, showing that it performs better than on old bucket or 2-bucket version? Specifically the multi-layer soil model of ORCHIDEE was tested quite a number of times already.

We agree to a certain extent with the reviewer's comment and we initially addressed this in our interactive informal response to this review:
https://editor.copernicus.org/index.php/hess-2019-598-SC1.pdf?_mdl=msover_md&_jrl=13&_lcm=oc108lcm109w&_acm=get_comm_file&_ms=81557&c=175959&salt=225704252184511132. We respond with some updated comments here.

It is true that most land surface models do now have a more mechanistic Richards' equation-type approach to modeling soil moisture dynamics. It's also the case that it is hard to compare the 2-layer and 11-layer approach given how different the representation of soil hydrology is and that it is very difficult to compare the 2-layer version to observations (much harder than the 11-layer).

However, despite these considerations we decided to keep the 2-layer vs 11-layer comparison in the first part of the results for this paper following reasons: firstly, we are expecting that not all readers are land surface modelers and that some of those people might either not be familiar with simple bucket models, or they might be users of hydrological or other types of models that still use a simple bucket scheme. For these readers, we wanted to show for a range of semi-arid sites that the bucket model really does not represent the temporal dynamics of the soil moisture or ET well; therefore, they should likely not trust ET predictions in semi-arid from any model that uses these types of soil hydrology schemes.

Secondly, the ORCHIDEE model CMIP5/IPCC AR5 simulations were based on the 2-layer version of the hydrology model. While this was a long time ago now and the CMIP6 simulations are being released, many people are still using CMIP5 to study various aspects of earth system processes, climate change impacts, or to understand model deficiencies. Given the fact that CMIP6 results are ~1 year delayed, we expect that people will continue to use CMIP5 simulations for at least another year. Therefore, we explicitly wanted to mention

that the ORCHIDEE CMIP5 ET predictions might not be as accurate as previously thought for semi-arid regions, with consequences for predictions of other variables.

Finally, we asked anonymous reviewer #2 what they thought about the 2 vs 11 layer comparison and, given the comments of this review, whether they would also be inclined to suggest keeping or discarding the comparison. See our initial interactive response to reviewer #2 here:
https://editor.copernicus.org/index.php/hess-2019-598-SC2.pdf?_mdl=msover_md&_jrl=13&_lcm=oc108lcm109w&_acm=get_comm_file&_ms=81557&c=175961&salt=1073010281052178988. Reviewer #2 replied that they disagree with removing the 2 vs 11 layer comparison. Their reasoning can be read here:
https://editor.copernicus.org/index.php/hess-2019-598-RC3-print.pdf?_mdl=msover_md&_jrl=13&_lcm=oc108lcm109w&_acm=get_comm_print_file&_ms=81557&c=176142&salt=1113425543317000663.

Bearing all these points in mind, we choose to keep the comparison between the 2 vs 11 layer, but in our revised manuscript we propose outlining our reasoning for this comparison more clearly by including the following statement in the introduction (after original lines 120-122):

"Although there have been many previous studies comparing simple bucket schemes versus mechanistic multi-layer hydrology based on the Richards equation, we include such a comparison in the first part of our analysis for the following reasons: a) the simple bucket schemes were the default hydrology in some CMIP5 model simulations and these simulations are still being widely used to understand ecosystem responses to changes in climate; b) variations on the simple bucket schemes are still implemented by design in various types of hydrological models (Bierkens et al., 2015); c) there has not yet been extensive comparisons of these two types of hydrology model for semi-arid regions, and especially not for the SW US; and d) so that the 2LAY can serve as a benchmark for the 11LAY scheme."

Bierkens, M. F. P.: Global hydrology 2015: State, trends, and directions, Water Resources Research, 51(7), 4923–4947, doi:10.1002/2015wr017173, 2015.

We hope this satisfies both reviewers.

But semi-arid ecosystems are interesting because quite a few model assumptions of LSMs get challenged there. Unfortunately the paper does not talk about it nor tries to advance in this direction.

We absolutely agree with the reviewer that many interesting aspects related to hydrology of heterogeneous semi-arid ecosystems were not either a) detailed in the model description and/or b) not elaborated on in the discussion. We have address this issue in detail for each of reviewer #1's comments below.

For example, ORCHIDEE uses tiles or fractions to deal with different land cover within one grid cell. To my knowledge, if a grid cell is vegetated then there is only transpiration (T). Evaporation (E) is from a special bare soil fraction only. There is no below-canopy E, which experiences lower wind speed, higher humidity and a litter layer compared to bare soil. This might have changed in the 11-layer version. Would be interesting to know. If the bare soil fraction mimics below-canopy E, then it is just a modelling concept and should be treated like this.

Reviewer #1 is right that if the grid cell is vegetated then there is only transpiration - but t*his is only the case* for the 2 layer scheme and not for the 11 layer. In the 11-layer scheme, soil evaporation *is allowed* from each PFT, proportionate to the effective bare fraction, which decreases when LAI increases. The effective vegetated fraction is calculated as an exponential function of LAI, and the effective bare fraction is the complement. The same roughness is used in both the effective bare and vegetated fractions, so reviewer 1 is right that in ORCHIDEE the soil evaporation does not depend on below-canopy conditions (i.e. there is no below canopy E).

In the initial manuscript we did mention the first point (that the bare soil fraction increases as LAI decreases) but we only made this point in the discussion (original lines 572 to 575 in section "Issues with modelling vegetation dynamics in semi-arid ecosystems"). However, it was not described as explicitly as we do here and we did not describe it in the model description. Therefore, in the revised manuscript we include the following lines at the end of Section 2.2.1 (the general model description) after we talk about the vegetation soil tiles in the model (original line 190):

"In the 11-layer scheme, both T and E occur in the vegetated soil tiles. T occurs over the effective vegetated fraction, which increases as LAI increases, whereas E occurs at low LAI over the effective bare soil fraction. The effective vegetated fraction is calculated following a modified Beer-Lambert equation describing attenuation of light penetration through a canopy $f\_v^j = f^j (1-e^{((-k\_ext 〚LAI〛\_j ) ) })$, where $f^j$ is the fraction of the grid cell covered by PFT j (i.e. the unattenuated case), $f\_v^j$ is the fraction of the effective fraction of the grid cell covered by PFT j and kext is the extinction coefficient and is set to 1.0. The effective bare soil fraction $f\_b^j$ is the complement to $f\_v^j$."

We further add at the end of Section 2.2.3 (Bare soil evaporation and additional resistance term) that there is no belowground E in ORCHIDEE:

"Note that there is no representation of below canopy E in ORCHIDEE and the same roughness is used for both the effective bare ground and vegetated fractions."

We also add a reference to the relevant model description sections when we discuss this issue in the first section of the discussion ("Issues with modelling vegetation dynamics in semi-arid ecosystems"):

"The connection between vegetation fractional cover and LAI is also a particular issue in sparsely vegetated regions when low LAI effectively means more bare soil is coupled with

the atmosphere *and E increases*. To account for this in ORCHIDEE, the bare soil fraction is slightly increased when LAI is low following a Beer-Lambert law approximation *(see section 2.2.1)*, which is often the case at these sites; however, there are only limited observations to support this model specification."

We also address the issue of below canopy E in the discussion section "ET partitioning (T/ET ratio)" by adding the following after the original final sentence in that section (which was "Nevertheless, in spatially heterogeneous mixed shrub-grass ecosystems it seems likely that missing model processes will need to be accounted for before accurate simulations of T/ET ratios are achieved.")

"One example of this might be the need to include in the model a representation of shrub understory and below canopy E."

Semi-arid ecosystems are probably the only ecosystems where this model structure is valid for soil evaporation. However, the rest of the model structure with fractions comes to its limits. If there is a shrub-encroached grassland, the shrubs (trees in this study) get all crammed into a small tile, shading each other and competing for soil moisture. Or is there a gap fraction in ORCHIDEE? Does it allow for shrub (tree) roots to forage in the grass tile? The grass in semi-arid ecosystems dies off during the year. This changes the LAI as discussed in the paper. But does the grass fraction stay constant? Should LAI rather stay constant in the grass tile but the tile should shrink, leading to more bare soil fraction? I think that one cannot discuss semi-arid ecosystems without talking about vegetation (dynamics). The CO2 fluxes could be interesting in this respect as well. They are omitted in the current paper.

The reviewer is absolutely right that the complexity of semi-arid vegetation dynamics are not well represented in this version of the model - resulting in weaknesses beyond the implementation of the hydrological scheme. No there is no gap fraction in this version of the model and no cross-foraging of tree roots in the grass tile etc. The fraction of vegetation stays constant in the model. All these points are severe limitations and changing these aspects of the model would indeed affect the hydrology. Unfortunately it would not be trivial to change these vegetation dynamics in the model and therefore we have not attempted to do so here. We did investigate the impact of reducing the bare soil fraction. This simple test was in place of having a more dynamic grass vs bare soil cover that changes over the course of the year (which is trickier to implement in ORCHIDEE although we are looking into it). In other words, this lower bare soil fraction test represents the other bookend of two possible ratios of grass to bare soil fraction. The reviewer is also right that this will affect CO2 fluxes. As mentioned in our initial information response to reviewer 1 we are investigating model representation of CO2 fluxes in a separate study. The issues related to CO2 fluxes are greater than can be fixed by changing the soil hydrology and therefore we have separated out these analyses into a separate, forthcoming paper. For this future paper we are also investigating the best way to implement more dynamic seasonal changes in grass cover but it is an ongoing study that is outside the scope of this current study. However, we have added the following sentence into Section 2.4 describing the simulations set-up so as to explain the reasoning for the reduced bare soil fraction test:

"Tests 3 and 5 (reduced bare soil fraction) are designed to account for the fact that grass cover is highly dynamic at intra-annual timescales at the low-elevation sites and therefore during certain seasons (e.g. the monsoon) the grass cover will likely be higher than is represented in the model."

Furthermore, while we did discuss all these issues of vegetation dynamics in the original manuscript discussion  (section entitled "Issues with modelling vegetation dynamics in semi-arid ecosystems"), we appreciate that we could have been clearer about these particular issues. Therefore, we have changed the first sentence of that section to:

"Our analysis has suggested that that biases in low-elevation shrub and grassland site ET might be due to incorrect simulations of seasonal vegetation dynamics; therefore, in order to obtain realistic estimates of ET and its component fluxes, it is important that the model can accurately simulate seasonal changes in leaf area and/or grass versus bare soil fractional cover."

And we have added the following sentence later in the paragraph after the original sentence "While not tested in this study, it is also possible that LSMs contain an inaccurate representation of different semi-arid vegetation *phenology*, including drought-deciduous shrubs and annual versus perennial C4 grasses". The new sentence is:

"The model does yet discern between perennial grasses and annual C4 grasses that only grow during warmest, wettest periods (Smith et al., 1997). It is possible that LSMs need new phenology models that account for annual C4 grass strategies in order to obtain accurate simulations of semi-arid water and carbon fluxes."

Developing new models that account for annual C4 grasses is also beyond the scope of this study unfortunately. We need to conduct separate analyses to develop such models, which will take some time (but we are working on it).

The paper discusses quite a few shortcomings of ORCHIDEE, or even LSMs in general. But there is no assessment of the importance of each point. They all seem to be similar important. I would have loved to see either prioritisation for model development or at least a guidance to the reader how to evaluate model shortcomings. The model might already be fine from an atmospheric perspective, or it might lead to a wet bias in spring.

The reviewer makes a good point here; however, it is hard to know how to prioritize model shortcomings. We did attempt to highlight issues that perhaps haven't been raised before in the final sentence of the conclusion (and this has been further adapted based on changes to the revised version):

"We recommend that future work on improving LSM semi-arid hydrological predictions focuses not only on issues highlighted in previous studies such as dynamic root zone moisture uptake, inclusion of ground water, lateral and vertical redistribution of moisture (e.g.

Whitley et al., 2016; 2017; Grippa et al., 2017) but also on: i) multi-variable calibration of vegetation and hydrology-related parameters across all sites; ii) more data to test modelled snow mass or depth at high elevation sites; iii) more data to better estimate and evaluate the seasonal trajectory of LAI across all sites and the vegetation fractional cover and LAI magnitudes at low elevation sites; and iv) testing of a more mechanistic description of resistance to bare soil evaporation."

We've discussed these points extensively above. We feel that these are the main contributions from this particular study and therefore serve as somewhat of a priority list, but we cannot evaluate how important they are compared to other issues that have been highlighted (e.g. the need for groundwater, dynamic root zone moisture uptake and lateral and vertical redistribution of moisture - which we also mention in the discussion) because we have not evaluated those components; indeed, they are not all implemented in the models yet. This is an age old issue in modeling - knowing which of the issues to focus on - and we appreciate it is frustrating.

Specific remarks are:
• I would change the tile. "Multi-variable" and "flux and storage" is tautologic. "Multi-configuration" is a bit much for two configurations.

The lead author admits she is not the best at formulating manuscript titles and thus agrees with the reviewer on this point. In response to the reviewer's comment, we suggest the title could be changed to:

"Testing water fluxes and storage from two hydrology configurations within the ORCHIDEE land surface model across US semi-arid sites"

• You should only cite one paper in preparation for CMIP6 and not once Ducharne et al. (in prep.) and once Peylin et al. (in prep.).

We have dropped the reference to the Peylin et al. paper in prep. The Ducharne paper is the relevant one for the hydrology.

• There are three personal communications, which are all from co-authors. Which co-author talked to which co-author?

It was the site PIs communicating with NM. However, we agree that given they are all co-authors these "pers. comms." are not needed so we have removed them.

• The description of "Richards and Darcy's equation" is strange. Darcy is part of Richards. The description is strange at two places (l.110 and l.211ff). I think that Richards equation is

known sufficiently so it is only interesting which form is solved, the saturation-based or the head-based form.

Agreed. We have removed the reference to Darcy and instead referred to it as the Richards equation around line 110 (introduction) and changed the sentence around line 211 to:

"The scheme implemented in ORCHIDEE relies on the one-dimensional Richards equation, combining the mass and momentum conservation equations, but is in the form of a Fokker-Planck equation that uses volumetric water content $\theta$ (m3m−3) as a state variable instead of pressure head."

• If LAI was identified to be important why is no local LAI data used? I found local LAI data in Scott and Biederman (2017) for some of the sites.

Actually the LAI data in Scott and Biederman (2017) are from the MODIS satellite with a 1km resolution. Indeed we would love to have local LAI data to validate the model, and it is something we are looking into with a PhD student at the University of Arizona. As we explain in the discussion section on "Issues with modelling vegetation dynamics in semi-arid ecosystems" there are unfortunately no local LAI timeseries we can use at these sites - all the data in the associated papers are derived from satellite measurements, and given the spatial heterogeneity at the site is it impossible to say which vegetation type is dominating the signal at this resolution as LAI doesn't scale linearly (i.e. you can't unmix the signal based on % cover type, and in fact, estimates of % cover type are uncertain given the heterogeneity):
"Similarly, there are not many LAI measurements for grasses and shrubs in these ecosystems; therefore, we have relied on estimating the LAImax parameter from MODIS LAI data. While different satellite LAI products often correspond well to each other in terms of temporal variability, there is often a considerable spread in their absolute LAI values (Garrigues et al., 2008; Fan et al., 2013); therefore, the MODIS LAI data may not be accurate for these ecosystems. In any case, the satellite LAI values represent a mix of different vegetation types and unlike satellite reflectance data it is not possible to linearly unmix the satellite LAI estimates based on fractional cover. More field LAI measurements are needed from different vegetation types (especially annual versus perennial grasses and shrubs) to verify what the likely maximum LAI is for each PFT. " Therefore, unfortunately at this time we cannot use local LAI data. We will revisit this in future studies if (hopefully, when) we get time series of field LAI data.

• Why are different T/ET algorithms used for different sites?

Initially, we used Scott and Biederman (2017) for the low elevation more water-limited shrub- and grass sites because it was deemed that this method is better at detecting T/ET for water limited sites following reasons given in that paper, namely that "Because we do not force the regression through the origin, our approach is more appropriate for water-limited sites, where it is often found that the ET ≠ 0 (i.e., the intercept) for GEP = 0 [Biederman et al.,

2016].". However, the method does not work well at the less water-limited forested sites - there is only a month or two where there are significant linear fits and where those fits yield positive ET axis intercepts. Indeed, Scott and Biederman had no intention of this method being universally used but just found that it worked particularly well for their sites (low elevation shrub and grassland). Thus, for the Fuf sites we used the Zhou method.

However, we appreciate that our original manuscript lacked a lot of detail and explanation when it came to the T/ET ratio estimates: we did not explain why there are two methods, we did not explain the S&B17 method well and we did not explain the Zhou et al. (2016) method at all in the methods. We also did not provide Zhou estimates for US-Vcp. These were oversights by the authors. We have corrected all these issues in the revised manuscript.

As the reviewer says below, there are a number of algorithms in the literature and it is hard to validate them. At the forested sites we only keep the Zhou et al. estimates for the reasons given above and at the lower elevation grass and shrub sites we now give estimates from both Zhou et al. (2016) and Scott and Biederman (2017) to show that indeed there is uncertainty in estimating T/ET ratios based on assumptions in different methods. We detail both of the these methods and our reasoning for having only Zhou at the forested sites and both at the grassland sites in Section 2.3.1 ("Site-level meteorological and eddy covariance data and processing") with the following sentence:

"Estimates of T/ET ratios were derived from Zhou et al. (2016) for the forested sites, and both Zhou et al. (2016) and Scott and Biederman (2017) at the more water-limited low elevation grass- and shrub-dominated sites. Zhou et al. (2016) (hereafter Z16) used eddy covariance tower GPP, ET and vapor pressure deficit (VPD) data to estimate T/ET ratios based on the ratio of the actual or apparent underlying water use efficiency (uWUEa) to the potential uWUE (uWUEp). uWUEa is calculated based on a linear regression between ET and GPP.VPD0.5 at observation timescales for a given site, whereas uWUEp was calculated based on a quantile regression between ET and GPP.VPD0.5 using all the half-hourly data for a given site. Scott and Biederman (2017) (hereafter SB17) developed a new method to estimate average monthly T/ET from eddy covariance data that was more specifically designed for the most water-limited sites. The SB17 method is based on a linear regression between monthly GPP and ET across all site years. One of the main differences between the Z16 and SB17 method is that the regression between GPP and ET is not forced through the origin in SB17 because at water-limited sites it is often the case that ET $\neq$ 0 when GPP = zero (Biederman et al., 2016). The Z16 method also assumes the uWUEp is when T/ET = 1, which rarely occurs in water-limited environments (Scott and Biederman, 2017)."

Based on the fact we now have also have T/ET estimates for US-Vcp and we also have two T/ET estimates for the grass and shrub dominated sites, we have adapted Figure 6 (and its caption) to include both estimates for the grass- and shrub-dominated sites and included the Zhou et al. (2016) method for the US-Vcp site. We have also altered the description of these results in Section 3.3 as described below.

Figure 6: Comparison of modelled and data-derived estimates of mean monthly T/ET ratios for each site. Forest site (US-Fuf and US-Vcp) T/ET estimates are derived using the method

of Zhou et al. (2016 – Z16 – green curve). Monsoon low-elevation grass- and shrub-dominated site T/ET estimated are based on both Zhou et al. (2016) and Scott and Biederman (2017 – SB17 – orange curve). Blue curves show the model ratios at each site. Please see Section 2.3.1 for details on methods for data-derived T/ET estimates.

[Figure]

For the forested sites, we have edited this paragraph: "Further support for the suggestion that modelled E is overestimated comes from examining the T/ET ratios. Although both E and T increase in the US-Fuf 11LAY simulations (compared to the 2LAY – Fig. S3a) – due to the increase in soil moisture (as previously described in Section 3.1 and Figs. 2 and S2a) – the larger increase in 11LAY E compared to T resulted in lower 11LAY T/ET ratios (Fig. S3a). The seasonal trajectory of T/ET ratios at US-Fuf appear to match data-derived estimates following the Zhou et al. (2016) method: the ratio peaks in the Spring before decreasing in July, with monsoon period T/ET values that are on average lower than the spring (Fig. 6). However, the magnitude of T/ET ratios are too low in all seasons given the 100% tree cover at this site with a LAI ~2.4. Whilst low spring 11LAY T/ET ratios may be due

to overestimated E as a result of higher soil moisture and underestimated snow cover, the generally low bias in T/ET ratios may also be due to the fact there is no bare soil evaporation resistance term included in the default 11LAY version."

to include a broader description of issues at the forested sites now we have T/ET estimates for US-Vcp as well as US-Fuf. The edited text now reads:

"Further support for the suggestion that modelled spring E is overestimated comes from comparing the model to estimated T/ET ratios (Fig. 6). Although both E and T increase in the US-Fuf and US-Vcp 11LAY simulations (compared to the 2LAY – Fig. S3a and b) due to the increase in soil moisture (as previously described in Section 3.1 and Figs. 2 and S2a), the stronger increase in 11LAY E compared to T resulted in lower 11LAY T/ET ratios across all seasons (Fig. S3a and b). While the model captures the bimodal seasonality at the forested sites as seen in the Z16 data-derived estimates (Fig. 6), the magnitude of model T/ET ratios appear to be too low in all seasons given the 100% tree cover at these sites with a maximum LAI of ~2.4. Whilst low spring 11LAY T/ET ratios at may be due to overestimated E as a result of higher soil moisture and underestimated snow cover, the generally low bias in T/ET ratios across all seasons at both US-Fuf and US-Vcp may also point to the issue that no bare soil evaporation resistance term is included in the default 11LAY version. This may also explain why the model T/ET ratios do not increase as rapidly as estimated values at the start of the monsoon (Fig. 6). Discrepancies in the timing of T/ET ratio peak and troughs between the model and data-derived estimates at the forested sites could also be due to the fact evergreen PFTs have no associated phenology modules in ORCHIDEE; instead, changes in LAI are just only subject to leaf turnover as a result of leaf longevity, which may be an oversimplification."

One of the main changes to the results following the inclusion of both methods is in the paragraph relating to US-SRM spring T/ET given that the model now lies in between the two estimates for this time period. Therefore, we have replaced this original text: "We can also glean some information on whether T or E (or both) are be responsible for the 11LAY overestimate of springtime ET at US-SRM by comparing modelled T/ET ratios against data-derived estimates. Observed T/ET ratios at the low-elevation sites were derived from independent eddy covariance data following the method of Scott and Biederman (2017) (Fig. 6). The observed spring T/ET at US-SRM is slightly underestimated by the model (Fig. 6). Given that T/ET ratios are underestimated by the model but ET is overestimated by the model, it is probable that spring E at this site is too high. Spring T could also be overestimated at US-SRM due potentially due to an overestimate in LAI (Fig. S5); however, the positive bias in E must be larger than the bias in T. If model LAI at US-SRM is too high during the spring, it is impossible to determine whether the shrub or grass LAI are inaccurate without independent, accurate estimates of seasonal leaf area for each vegetation type; however, in the field the spring C4 grass LAI is typically half that of its monsoon peak (R.L. Scott – pers. comm.) – a pattern not seen in the model (Fig. S6)."

with

"At US-SRM, the modelled spring T/ET ratio overestimates the Z16 estimate and underestimates the SB17 estimate (Fig. 6). The current state of the art is that different methods for estimating T/ET typically compare well in terms of seasonality but differ in absolute magnitude; therefore, the uncertainty in T/ET magnitude during the spring at US-SRM makes it difficult to glean any information on whether T or E (or both) are be responsible for the 11LAY overestimate of springtime ET (Fig. S3c). If the SB17 method is more accurate, then it is probable that modelled spring E at this site is too high. However, if the Z16 estimate is accurate, then it is likely that spring T is overestimated at US-SRM, potentially due to an overestimate in LAI. The model-data bias in spring mean monthly ET is well correlated (0.XX) with spring mean LAI at US-SRM (Fig. S5). If model LAI at US-SRM is too high during the spring, it is impossible to determine whether the shrub or grass LAI are inaccurate without independent, accurate estimates of seasonal leaf area for each vegetation type, which are not available at present; however, in the field the spring C4 grass LAI is typically half that of its monsoon peak – a pattern not seen in the model (Fig. S6). We will test both of these hypotheses (overestimate in either T or E) in Section 3.4."

We have also edited the following original text: "Data-derived T/ET ratios also help to diagnose why the 11LAY model underestimates monsoon ET at the low-elevation shrub sites (US-SRM and US-Whs– Figs. S3 c-d). Fig. 6 shows that the 11LAY model also underestimates monthly T/ET ratios, and furthermore, that the model does not capture the correct temporal trajectory (Fig. 6). Although the earlier summer drop in T/ET ratios in the 11LAY compared to the 2LAY simulations at grass and shrubland sites (Figs. S3 c-f) does result in a better match in ET between the model and the observations (Fig. 3), the 11LAY T/ET ratios are slightly out of phase. Observed T/ET ratios decline in June during the hottest, driest month, whereas model values decrease one month later in July (Fig. 6). Furthermore, the ratios do not increase as rapidly as observed during the wet monsoon period (July – September).
The underestimate in modelled monsoon T/ET ratios across all grassland and shrubland sites (and likely at US-Fuf and US-Vcp) suggests either that transpiration is too low or bare soil evaporation is too high. At the shrubland sites (US-SRM and US- 500 Whs), both monsoon ET and T/ET are underestimated; therefore, for these sites it is plausible that the dominant cause is a lack of transpiring leaf area. Certainly, monsoon model-data ET biases are better correlated with LAI at shrubland sites compared to grassland sites (Fig. S7). The underestimate in modelled monsoon period leaf area could either be: i) an underestimate of maximum LAI for either grasses or shrubs; or ii) due to the fact the static vegetation fractions prescribed in the model do not allow for an increase in vegetation cover during the wet season (e.g. the lack grass growth in the model in interstitial bare soil 505 areas). In contrast, at the grassland sites (US-SRG and US-Wkg) monsoon ET is well approximated by the 11LAY model; thus, the underestimate in T/ET ratios suggests that both the transpiration is too low and the bare soil evaporation too high." to include both T/ET methods, to make the text more understandable, and to provide further explanation of the "out of phase" seasonality in T/ET ratios at the low elevation sites. The new text is:

"At the low elevation grass- and shrub-dominated sites, both data-derived estimates of T/ET agree on their seasonality and sign with respect to the model magnitude during the

monsoon. Given this agreement, both sets of estimated values can help to diagnose why the 11LAY model underestimates monsoon peak ET at the low-elevation shrub sites (US-SRM and US-Whs– Figs. S3 c-d). Fig. 6 shows that the 11LAY model also underestimates both Z16 and SB18 monthly monsoon period T/ET estimates across all low elevation sites. The underestimate in modelled monsoon T/ET ratios across all grassland and shrubland sites suggests either that T is too low or E is too high. At the shrubland sites (US-SRM and US-Whs), both monsoon ET and T/ET are underestimated; therefore, for these sites it is plausible that the dominant cause is a lack of transpiring leaf area. As was the case for spring ET at US-SRM, monsoon model-data ET biases are better correlated with LAI at shrubland sites compared to grassland sites (Fig. S7). In contrast, at the grassland sites (US-SRG and US-Wkg) monsoon ET is well approximated by the 11LAY model; thus, the underestimate in T/ET ratios suggests that both the transpiration is too low and the bare soil evaporation too high.

Furthermore, although the 11LAY does capture the decrease in ET during the hot, dry period of May to June (which is a significant improvement compared to the 2LAY – see Section 3.1), the 11LAY T/ET ratios are slightly out of phase with the estimated values. Both data-derived estimates agree that T/ET ratios at all low elevation sites decline in June during the hottest, driest month (as expected); however, the model T/ET ratios reach a minimum one month later in July (Fig. 6). This one month lag in model T/ET ratios is apparent despite the fact that the ET minimum is accurately captured by the model (Figs. 3b and S3). The modelled T/ET ratios also do not increase as rapidly as both estimates during the wet monsoon period (July – September), which can be explained by the fact that the model E at the start of the monsoon increases much more rapidly than modelled T. Taken together, these results suggest that LAI is not increasing rapidly enough after the start of monsoon rains (see Fig. S6), resulting in low biased T/ET ratios in July. Meanwhile the increase in available moisture from monsoon rains is causing a biased high model E that compensates for the lower T. These compensating errors result in accurate ET simulations. The underestimate in modelled leaf area during the monsoon could either be: i) incorrect timing of LAI growth for either grasses or shrubs and an underestimate of peak LAI; and/or ii) due to the fact the static vegetation fractions prescribed in the model do not allow for an increase in vegetation cover during the wet season (e.g. the model lacks the ability to grow grass in interstitial bare soil areas)."

We have also added the following sentence in the abstract:
"However, discrepancies in the timing of the transition from minimum T/ET ratios during the hot, dry May-June period to high values during the summer monsoon period in July-August could point towards incorrect simulations of seasonal leaf phenology. "

• T/ET is seen as a measurement in the manuscript. But it is not. Any validation is missing in the Scott and Biederman (2017) paper, because it is pretty impossible to validate it. So T/ET should be seen only as an estimate. There are quite some algorithms in the literature to calculate T/ET and it is hard to tell why one should be more correct than the other.

We agree and shouldn't have ever referred to the T/ET ratios as "observations" we have changed all the text throughout to refer to these as "estimates" or "data-derived estimates".

• l.255: what is the subscript j on cj?

Thank you for spotting this. It refers to the PFT. We have added this into the manuscript. We have also changed all other subscripts referring to PFT to *j* and not *v* as was in the original manuscript.

• l.255ff: R(z) is explained but not nroot. If nroot were explained then one does not have to (confusingly) start the sums from 2 because nroot=0 in v=1 and i=1.

nroot is explained in the original manuscript on lines 257-258 (directly after explaining R(z): "In 11 LAY, a related variable is nroot(i), quantifying the mean relative root density of each soil layer i, so that $\sum$ nroot(i) = 1".

• l265ff: Why is the relative water content weighted with nroot? This formulation is an empirical observation and the beta term is never weighted by root length density (or similar) in the data papers (e.g. Keenan et al. (Biogeosci 2009)).

The exponential dependence of beta to soil moisture in the 2 layer scheme can be related to the convolution of SM and root density controls, as demonstrated by de Ronsay et al 1998. The root density control component was then extended by de Ronsay et al 2002 to the multi-layer scheme. Whilst it may not be in the data papers, we believe that an exponential decay of root density must be a common assumption, and therefore that convolution of SM and root density controls for plant water uptake are reasonable formulations. It is certainly a common approach in other LSMs (e.g. De Kauwe et al., 2015). These papers are already cited elsewhere in the model description section, particularly the De Kauwe paper in the new discussion section "Implications for modelling plant water stress" (see comment below) and we also highlight the need for calibrating water stress function parameters as well as parameters related to root zone uptake. But we can add a sentence clarifying this at this point in the manuscript if needed.

De Kauwe, M. G., Zhou, S.-X., Medlyn, B. E., Pitman, A. J., Wang, Y.-P., Duursma, R. A. and Prentice, I. C.: Do land surface models need to include differential plant species responses to drought? Examining model predictions across a mesic-xeric gradient in Europe, Biogeosciences, 12(24), 7503–7518, doi:10.5194/bg-12-7503-2015, 2015.
de Rosnay, P. and Polcher, J.: Modelling root water uptake in a complex land surface scheme coupled to a GCM, Hydrol. Earth Syst. Sci., 2, 239–255, https://doi.org/10.5194/hess-2-239-1998, 1998.

• l.268: Should W be in kg/m3 instead of kg/m2? Why is W used and not volumetric soil moisture theta?

The units are correct here (kg/m2). This takes into account the total water content in each layer of different thickness.

• l270ff: Why is p% = 0.8? There is quite some literature that it should be around 0.4 (e.g. Granier et al. (AFM 2007)), at least for forests?

The water stress function of the 11-layer hydrology scheme was inspired by the bucket model, of Manabe (1969), who used a value of 0.75 for the equivalent parameter to p%, and mentioned a plausible range of 0.7-0.8 based on Alpatev (1954).

A quick look at the literature shows that the range of values that is effectively used in LSMs is between 0.4 and 1 for the place in the WP-FC range at which the water stress function becomes 1 (corresponding no unstressed transpiration), regardless of the shape of the function (see for instance the review by Mahfouf et al 1998, or Verhoef and Gregorio, 2014).

MANABE, S., 1969: CLIMATE AND THE OCEAN CIRCULATION. Mon. Wea. Rev., 97, 739–774, https://doi.org/10.1175/1520-0493(1969)097<0739:CATOC>2.3.CO;2

Alpatev, A. M., "Vlagooborot kul'turnykh rastenil," (Moisture Exchange in Crops), Gidrometeoizdat, Leningrad, 1954, 247 pp.

Mahfouf JF, Ciret C, Ducharne A, Irannejad P, Noilhan J, Shao Y, Thornton P, Xue Y, Yang ZL (1996). Analysis of transpiration results from the RICE and PILPS Workshop, Global and Planetary Change , 13, 73-88, doi:10.1016/0921-8181(95)00039-9

Verhoef, A., and Gregorio, E. (2014). Modeling plant transpiration under limited soil water: Comparison of different plant and soil hydraulic parameterizations and preliminary implications for their use in land surface models, Agricultural and Forest Meteorology, 191, 22-32, https://doi.org/10.1016/j.agrformet.2014.02.009.

As described in other responses to both reviewers, for many other parameters in this model we use the default values to test the default behavior (also to allow a comparison to forthcoming CMIP6 results), and have not performed a full calibration of all these parameters as this would take too long and is therefore outside the scope of this study. In the discussion we have discussed the need for parameter calibration, including the need to optimize "water-limitation parameters". p% also is a universal parameter and not PFT-dependent. We have not investigated the need for PFT-dependence of this parameter but again we would take that into account when doing a parameter calibration.

• l.276f: The references are missing. And only the Keenan et al. paper actually supports this claim. The Zhou et al. papers do something very different and act only on stomatal conductance.

Thank you for pointing out the missing references. We have added these references in. However, we disagree that the Zhou et al. papers do something different and only act on Gs (also following discussion with collaborators on this work). See for example the following text in the 2013 paper: "The results are consistent with other stud- ies showing that both stomatal and non-stomatal processes are affected by drought (e.g. Egea et al., 2011; Keenan et al., 2010). Our analysis shows that non-stomatal limitation is considerable and has in general a greater impact than that of stomatal limitation on pho- tosynthetic rates. Photosynthesis under drought would be greatly overestimated if the decline in apparent Vcmax was not taken into account. Both assimilation rate and stomatal conductance decrease as pre-dawn leaf water potential declines, but assimilation rate usually decreases more – often many times more – than could be explained by a reduction in stomatal conductance (and g1) alone (see Figs. 1 and 2 in Appendix B)."
And from the 2014 paper "We found consistency among the drought responses of g1, gm, Vcmax and Jmax, suggesting that drought imposes limitations on Rubisco activity and RuBP regeneration capacity concurrently with declines in stomatal and mesophyll conductance". The beta functions are different in the Zhou studies (resulting in different shapes of water-limitation function).

Keenan, T., Sabate, S. and Gracia, C.: The importance of mesophyll conductance in regulating forest ecosystem productivity during drought periods, Global Change Biology, 16(3), 1019–1034, doi:10.1111/j.1365-2486.2009.02017.x, 2010.
Zhou, S., Duursma, R. A., Medlyn, B. E., Kelly, J. W. and Prentice, I. C.: How should we model plant responses to drought? An analysis of stomatal and non-stomatal responses to water stress, Agricultural and Forest Meteorology, 182-183, 204–214, doi:10.1016/j.agrformet.2013.05.009, 2013.
Zhou, S., Medlyn, B., Sabaté, S., Sperlich, D., Prentice, I. C. and Whitehead, D.: Short-term water stress impacts on stomatal, mesophyll and biochemical limitations to photosynthesis differ consistently among tree species from contrasting climates, Tree Physiology, 34(10), 1035–1046, doi:10.1093/treephys/tpu072, 2014.

• l.303f: I wondered if this claim means that you have a near perfect energy balance closure?

Energy balance closure at the low elevation sites is typically good, on the order of 10%. At the flagstaff site energy balance closure was 0.69 or greater for 30-minute values, and 0.81 or greater for daily values (Dore et al. 2010). But no, the close matching of annual ET with P indicates mainly these sites have very little runoff and drainage, i.e. most precipitation evaporates or transpires locally (also verified in the cited paper with additional hydrologic measurements).

• l.315f: why are there no site-specific soil characteristics? They must have been done at some point in the past.

In fact this sentence is misleading - these parameters have not all been measured at all sites. The parameters we need are mostly not available. No site has measured all the soil and hydraulic parameters we need (perhaps one or two) given the number and difficulty of

measuring them, and some sites don't have any measurements. So it makes it difficult to only use site-specific parameters for just a few of the values we need and not across all sites. We therefore have taken an approach that we only set site specific parameters if we have them for all sites and the rest we are effectively testing the default model parameters (which has the benefit that we're testing the default model behavior). We have added this sentence in to section 2.4 ("Simulation set-up and post-processing") and refer to this section around the lines the reviewer has highlighted in this comment.

"Due to the lack of available data on site-specific soil hydraulic parameters across the sites studied, we chose to use the default model values that were derived based on pedotransfer functions linking hydraulic parameters to prescribed soil texture properties (see Section 2.2.2). Using the default model parameters values also allows us to test the default behavior of the model."

However, as we point out in the end of results section 3.4, in the discussion section on Bare Soil Evaporation, and in the conclusions, it is possible that calibrating these hydraulic parameters at each site would be beneficial, as done in this study:

Shi, Y., Baldwin, D. C., Davis, K. J., Yu, X., Duffy, C. J. and Lin, H.: Simulating high-resolution soil moisture patterns in the Shale Hills watershed using a land surface hydrologic model, Hydrological Processes, 29(21), 4624–4637, doi:10.1002/hyp.10593, 2015.
We have added that reference to that sentence in the discussion section on bare soil evaporation.

It is also possible that further analyses using pedotransfer functions to determine soil hydraulic parameters from soil texture data at each site would be useful but we have not done this for this study - in part because the pedotransfer functions themselves are uncertain (Mermoud et al., 2006). Some of the authors are involved in ongoing investigations related to this topic. Taking all this into consideration, it's not clear that we would improve the accuracy or reliability of the model by using pedotransfer functions to derive these parameters, and as we said above it is useful (particularly considering ongoing CMIP6 experiments) to test the default behavior of the model. However, we have added the following sentence into the discussion section on bare soil evaporation (after adding the reference to Shi et al., 2015) to highlight that, along with statistical parameter calibration experiments, it may be possible (if needed) to better determine soil hydraulic properties following further investigation into the uncertainty surrounding available pedotransfer functions:

"Future studies could also investigate the impact of uncertainty in the use of pedotransfer functions (e.g. Mermoud et al., 2006) in deriving soil hydraulic parameters from soil texture information. "

Mermoud, A. and Xu, D.: Comparative analysis of three methods to generate soil hydraulic functions, Soil and Tillage Research, 87(1), 89–100, doi:10.1016/j.still.2005.02.034, 2006.

• Fig. 1: Where are the observations?

We have added ET observations but not the observations for soil moisture variables because in this plot these given as total water content (see comment below) to see overall mean changes in the amount of water in the upper and total soil column and therefore have not been re-scaled to match observations (as we outline in Section 2.3.2). Instead, we use the re-scaled soil moisture observations for all other plots. We also propose adding the following in the Figure 1 caption to make this point clear:

"For soil moisture, the absolute values of total water content for the upper layer and total 2m column are shown for both model versions, i.e. the simulations have not been re-scaled to match the temporal dynamics of the observations (as described in Section 2.3.2); therefore, soil moisture observations are not shown. Observations are only shown for ET."

We have also changed the description of how we process soil moisture data in Section 2.3.2 to highlight this point:

"Therefore, with the exception of Fig. 1 in which we examine changes in total water content between the two model versions, for the remaining analyses we do not focus on absolute soil moisture values in the model – data comparison, we specifically investigate how well the model captured the temporal dynamics at specific soil depths."

• Fig. 1: Harmonise scales of ET, Runoff and Drainage, as well as of Upper SM and Total SM so that one can compare the fluxes/stocks. For example, why is Total SM up to 1000? If kg/m3, then Upper SM and Total SM could have the same scale. If kg/m2, they should be scaled according to layer depth.

We have harmonized the scales for all variables with the same units.

The units are kg/m2, not kg/m3. The total SM sums up SM over all the layers (0-2m - as in the y-axis title). The upper layer is only over the top 10cm. The max value shouldn't have been 1000 - this has been adjusted. We can convert these to m3/m3 (volumetric water content instead of total water content) if the reviewer would prefer so the upper layer and total column scales can be more comparable.

Fig. 1: Why is there (almost) no drainage at forested sites with the 11-layer version? Is this realistic? There is only a very small mention for US-Fuf in the text.

It is unfortunate that we don't have more data on runoff and drainage across all these sites, as we mention in the discussion. We do have the following sentence for US-Fuf in the text as the reviewer points out: "The 11LAY limited drainage is also likely to be the case at US-Fuf given that nearly all precipitation at the site is partitioned to ET (Dore et al., 2012).". We don't have any corresponding data for US-Vcp unfortunately. However, in general these semiarid flux sites have very little precipitation that is not accounted for by ET, at the annual scale (i.e.

looking at ET:P ratios).  This means that precip can be much higher than ET for some months (winter) but "catch up" during others (spring, early summer). See Biederman et al. (2017) Table S1. We have included this sentence where we talk about drainage:

"In general, all these semi-arid sites have very little precipitation that is not accounted for by ET at the annual scale (Biederman et al., 2017 Table S1)."

• Fig. 2: I think the titles of the y-axes of row 3 and 4 are swapped.

The y-axes labels of rows 3 and 4 are correct but the description in the caption is the wrong way round - thank you for spotting that. This was also wrong for Fig. S2 so we have corrected the captions for both figures.

• Fig. 4: please put the 2 cm, 20, cm and 50 cm plots on the same scales.

Done, thank you (and for Fig. S4).

• Fig. 5b: Data stays low during much of the snowfall period. This can happen if the data is measured inside a forest whereas the model assumes open space. Much of SnowMIP's model intercomparison, at which ORCHIDEE probably par- ticipated, focussed on open sites. We might not know well the behaviour of our models at forest sites.
It looks like that the data is even decreasing at the beginning of the snowfall period. This could point to soil freezing. Some soil moisture sensors measure only liquid water, so low values are measured during frozen soil conditions. So sites also do not include possible ice phases in their transformations from voltage to soil moisture.
Both processes were not discussed.

We thank the reviewer for pointing these processes out. We looked at the modeled surface temperatures and indeed found that the early winter positive model-data bias (model higher than the data) coincided with negative surface temperatures (which is therefore possibly related to instrument biases or the issue of open sites vs a closed forest setting as the reviewer mentioned). So we have increased the description and discussion of these results and made the paragraphs related to snow biases at the high elevation sites more nuanced. These paragraphs now read:

"In contrast, the temporal mismatch between the observations and the model in the uppermost layer is higher at the forest sites. The US-Fuf and US-Vcp 11LAY simulations appear to compare reasonably well with observations in the upper 2cm of the soil from June through to the end of November (end of September in the case of US-Vcp) (Fig. 4). However, in some years the model appears to overestimate the VWC at both sites during the winter months (positive model-data bias), and underestimate the observed VWC during the spring months (negative model-data bias), particularly at US-Fuf. Although US-Fuf and US-Vcp are semi-arid sites, their high-elevation means that during winter, precipitation falls as snow; therefore, these apparent model biases may be related to: i) the ORCHIDEE snow scheme; ii) incorrect snowfall meteorological forcing; and/or iii) incorrect soil moisture

measurements under a snow pack. During the early winter period the model soil moisture increases rapidly as the snowpack melts and is replenished by new snowfall, whereas the observed soil moisture response is often slower (Fig. 5a and b light blue zones). This often coincides with periods when the surface temperature in the model is below 0°C (Fig. 5 bottom panel), suggesting that in reality soil freezing may be negatively biasing the soil moisture measurements. An alternative explanation is that ORCHIDEE overestimates snow cover (and therefore snow melt and soil moisture) at the forest sites because it is assumed that snow is evenly distributed across the grid cell, whereas in reality the snow mass/depth is lower under the forest canopy than in the clearings.

At US-Fuf, it appears that the model melts snow quite rapidly after the main period of snowfall (Fig. 5a light green zones). Once all the snow has melted, the model soil moisture also declines; however, the observed soil moisture often remains high throughout the spring – causing a negative model-data bias (Fig. 5a). Unlike US-Fuf, a similar negative model-data bias at US-Vcp often coincides with periods when snow is still falling, although the amount is typically lower (Fig. 5b light green zones); however, the model does not always simulate a high snow mass during these periods. These periods coincide with rising surface temperature above 0°C. Although snow cover, mass, or depth data have not been collected at these sites, snow typically remains on the ground until late spring after winters with heavy snowfall, suggesting that the continued existence of a snow pack and slower snow melt that replenishes soil moisture until late spring when all the snow melts. Therefore, the lack of a simulated snow pack into late spring could explain the negative model-data soil moisture bias. To test the hypothesis that the model melts or sublimates snow too rapidly, thereby limiting the duration of the snowpack and also allowing surface temperatures to rise, we altered the model to artificially increase snow albedo and decrease the amount of sublimation; however, these tests had little impact on the rate of snow melt or the duration of snow cover (results not shown). Aside from model structural or parametric error, it is possible that there is an error in the meteorological forcing data. Rain gauges may underestimate the actual snowfall amount during the periods when it is snowing (Rasmussen et al., 2012; Chubb et al., 2015). If the snowfall is actually higher than is measured, it may in reality lead to a longer lasting snowpack than is estimated by the model. To test this hypothesis, we artificially increased the meteorological forcing snowfall amount by ten times and re-ran the simulations. Although this artificial increase is likely exaggerated, the result was an improvement in the modelled springtime soil moisture estimates at US-Fuf (Fig. S5). However, the same test increased positive model-data bias in the early winter increased at US-Fuf, and degraded the model simulations at US-Vcp. This preliminary test suggests that inaccurate snowfall forcing estimates may play a role in causing any negative model-data bias spring soil VWC but more investigation is needed to accurately diagnose the cause of the springtime negative model-data bias."

To better match this text we have updated Figure 5 to only include the pertinent variables (and have added surface temperature) and we have added an extra supplementary figure (S5) to show the results of the increased snow forcing (as per a comment from Reviewer 2):

Figure 5: a) US-Fuf and b) US-Vcp 11LAY (blue curve) daily time series (2007-2010) of model versus re-scaled (via linear CDF matching) observed volumetric soil water content (middle panel SWC – m3m-3) (black curve), compared to simulated snow mass (top panel)

and surface temperature (bottom panel). Snowfall is also shown as grey lines in the SWC time series. In the bottom panel the grey horizontal dashed line shows 0°C threshold.

[Figure]

Figure S5: Linear regressions between spring (March-April) mean monthly LAI (m2m-2) and spring mean monthly ET (mmmonth-1) model-data misfits for each site. The dominant PFT is given in brackets for each site. See Table 1 for PFT acronyms.

[Figure]

We have also added this sentence into the abstract:
"Biases in winter and spring soil moisture at the forest sites could be explained by inaccurate soil moisture data during periods of soil freezing and underestimated snow forcing data."

Finally, we also updated a sentence in the conclusions to reflect both the negative and positive model-data biases in soil moisture at the forested sites could be related to snowfall issues:

"Remaining discrepancies in both overestimated and underestimated winter and spring soil moisture at high-elevation semi-arid forested sites might be respectively related to issues with soil moisture data during periods of soil freezing and underestimated snowfall forcing data causing a limited duration snowpack, with consequent implications for predictions of water availability in regions that rely on springtime snowmelt."

• I.384f: This is a "false friend" to me. Evaporation is water vapour but the Richards equation (as used in ORCHIDEE) does not include vapour transport in the soil. So the model has to compensate for this omittance. This is one of the primary reasons why the Richards solvers need very thin layers at the top of the soil. These layers cannot be seen as physical layers because they have to compen- sate for all the model deficiencies on top of possible litter layers. It is thus doubtful that these first few layers should be compared satellite measurements.

To solve Richards equation, we need thin layers at the atmosphere interface, not because we need to compensate for model deficiencies in lacking vapor transport but because the

moisture gradients are larger (as we discussed in section 2.2.2). Models representing vapor transfer have even thinner discretization.

Concerning the comparison with satellite data, we agree that the non representation of vapor transfers, could lead to an overestimation of soil moisture in the surface layers but could be balanced by the fact that the satellite sounds also a deeper soil in dry soil conditions. But given the fact that the average sensing depth of the microwave instruments is of a few centimeters, the capacity of the model to represent thin layers compared to the 2LAY is a benefit. The challenges and benefits of how to compare model soil moisture with satellite soil moisture are discussed extensively in Raoult et al. (2018) (which we cite here).

• I.396: Isn't this a contradiction to Whitley et al. (2016). You state in the introduction that Whitley et al. (2016) found that T of the vegetation is mostly too low in the models. Is 2-layer ORCHIDEE different so that 11-layer ORCHIDEE can decrease T during the warm season?

Indeed this is a good point. It is not so much that the 2-layer and 11-layer are different here so much as the modification to the formulation of beta (water stress function) has allowed there to be a greater decrease in T during the hot, dry (water limited) periods - as highlighted by the brown shaded zones in Figure 2. We state this in the previous sentence at lines 392-394 in the original manuscript with the following sentence "At the low-elevation shrub and grass sites, the improvement in ET is also related to changes between the two versions in the calculation of the empirical water stress function, b (Figs. 2 and S2 5th panel), which acts to limit both photosynthesis and stomatal conductance (therefore, T) during periods of moisture stress (Section 2.2.4)."

However, we agree that it's worth noting what is, what isn't, similar to the findings of Whitley et al. (2016) in our study given we have highlighted that study in the introduction. Therefore, we have added an small extra section to the discussion with the following text (that also takes the opportunity to discuss more broadly about modeling plant response to water stress):

"**Implications for modelling plant water stress**

Similar to Whitley et al. (2016), the original 2LAY version of the model underpredicted wet monsoon season ET. The peak ET fluxes were generally much better captured in the 11LAY version. However, in contrast to Whitley et al. (2016), the 2LAY simulations overestimated ET during the hottest, driest period between May and June. Our results demonstrated that a modified empirical beta water stress function (used to downregulate stomatal conductance during periods of limited moisture) that takes into account available soil moisture and root density across the entire soil column (Section 2.2.4) helped to better capture dry season ET dynamics. These results are interesting in light of previous studies showing that LSMs employing empirical beta water stress functions show considerable differences in their simulated soil moisture response to during water stressed periods (Medlyn et al., 2016; De Kauwe et al., 2017). These studies argue for more evidence-based formulations of plant response to drought. De Kauwe et al. (2015) also highlight the need for models to

incorporate dynamic root zone soil moisture uptake down profile as the soil dries. It is therefore possible that while the modified beta function used in the 11LAY does help to capture seasonal water stress, as in this study, new mechanistic plant hydraulic schemes that can track transport of water through the xylem (e.g. Bonan et al., 2014; Naudts et al., 2015) may be needed when simulating plant response to prolonged drought periods. However, comparing beta functions versus plant hydraulic schemes under severe water stressed periods was not within the scope of this study. When discussing woody plant responses to drought, it is also worth noting that many LSMs to date are also missing any representation of groundwater (Clark et al., 2015). As described in Section 2.1, the water table is typically very deep (10s to 100s metres) at these sites. Previous modeling studies have shown that only rather shallow water tables (~1m) are likely to significantly increase ET in the SW US (e.g. by >=2.4mmd-1 in Fig. 4g of Wang et al., 2018). However, the fact LSMs typically do not include adequate descriptions of groundwater access could impact their ability to simulate savanna ecosystem dry season water uptake given that drought deciduous shrubs in Mediterranean and semi-arid ecosystems are more resilient to droughts due to their ability to tap groundwater reserves (e.g. Miller et al., 2010). A new groundwater module is being developed for ORCHIDEE and will be tested in future studies."

Bonan, G. B., Williams, M., Fisher, R. A. and Oleson, K. W.: Modeling stomatal conductance in the earth system: linking leaf water-use efficiency and water transport along the soil–plant–atmosphere continuum, Geoscientific Model Development, 7(5), 2193–2222, doi:10.5194/gmd-7-2193-2014, 2014.
De Kauwe, M. G., Zhou, S.-X., Medlyn, B. E., Pitman, A. J., Wang, Y.-P., Duursma, R. A. and Prentice, I. C.: Do land surface models need to include differential plant species responses to drought? Examining model predictions across a mesic-xeric gradient in Europe, Biogeosciences, 12(24), 7503–7518, doi:10.5194/bg-12-7503-2015, 2015.
De Kauwe, M. G., Medlyn, B. E., Walker, A. P., Zaehle, S., Asao, S., Guenet, B., Harper, A. B., Hickler, T., Jain, A. K., Luo, Y., Lu, X., Luus, K., Parton, W. J., Shu, S., Wang, Y. P., Werner, C., Xia, J., Pendall, E., Morgan, J. A., Ryan, E. M., Carrillo, Y., Dijkstra, F. A., Zelikova, T. J. and Norby, R. J.: Challenging terrestrial biosphere models with data from the long-term multifactor Prairie Heating and CO2 Enrichment experiment, Global Change Biology, 23(9), 3623–3645, doi:10.1111/gcb.13643, 2017.
Medlyn, B. E., Kauwe, M. G. D., Zaehle, S., Walker, A. P., Duursma, R. A., Luus, K., Mishurov, M., Pak, B., Smith, B., Wang, Y.-P., Yang, X., Crous, K. Y., Drake, J. E., Gimeno, T. E., Macdonald, C. A., Norby, R. J., Power, S. A., Tjoelker, M. G. and Ellsworth, D. S.: Using models to guide field experiments:a prioripredictions for the CO2response of a nutrient- and water-limited native Eucalypt woodland, Global Change Biology, 22(8), 2834–2851, doi:10.1111/gcb.13268, 2016.
Miller, G. R., Chen, X., Rubin, Y., Ma, S. and Baldocchi, D. D.: Groundwater uptake by woody vegetation in a semiarid oak savanna, Water Resources Research, 46(10), doi:10.1029/2009wr008902, 2010.
Naudts, K., Ryder, J., Mcgrath, M. J., Otto, J., Chen, Y., Valade, A., Bellasen, V., Berhongaray, G., Bönisch, G., Campioli, M., Ghattas, J., Groote, T. D., Haverd, V., Kattge, J., Macbean, N., Maignan, F., Merilä, P., Penuelas, J., Peylin, P., Pinty, B., Pretzsch, H., Schulze, E. D., Solyga, D., Vuichard, N., Yan, Y. and Luyssaert, S.: A vertically discretised canopy description for ORCHIDEE (SVN r2290) and the modifications to the energy, water

and carbon fluxes, Geoscientific Model Development, 8(7), 2035–2065, doi:10.5194/gmd-8-2035-2015, 2015.

Wang F, Ducharne A, Cheruy F, Lo MH, Grandpeix JL (2018). Impact of a shallow groundwater table on the global water cycle in the IPSL land-atmosphere coupled model, Climate Dynamics, 50, 3505-3522, doi:10.1007/s00382-017-3820-9

• l.448ff: There also seems to be a problem with infiltration. At the model attenu- ates precipitation peaks too much at forest sites, while it is almost not attenuating at the grassland sites. Could you explain that please. There seems to be a differ- ence in the model why water can flow quickly to deep layers in grassland but not in forests. Or is it the bare soil fraction?

We thank the reviewer for raising this issue because in fact we omitted one important change of saturated hydraulic conductivity (Ks) with depth, which is an exponential increase in Ks towards the surface to account for the effect of increased soil porosity due to bioturbation by roots. Given tree roots are deeper this increase towards the surface starts lower in the profile, and as Ks increases towards the surface, so does the infiltration capacity. Therefore, infiltration under the forests is likely to be quicker, which we believe explains the smoother profiles at depth under the forested sites (although looking at the full timeseries in Fig. S4a the model doesn't do a bad job at US-Fuf of capturing the largest swings in soil moisture in the deepest layer - the smooth model temporal profile at depth is more of an issue at US-Vcp). This explains the difference in the model behavior between the forest and grass sites; however, it doesn't explain why the model simulations don't capture the observed soil moisture dynamics as well at depth. One reason may be that in the absence of PFTs defined specifically for semi-arid ecosystems we are essentially modeling the trees and shrubs in these ecosystems as temperate trees. One parameter that might be very different is the root density decay factor. Semi-arid shrubs and trees tend to have deeper tap roots than their temperate counterparts to account for limited water availability. Often they also have extensive shallow root systems, which is not something we can account for in ORCHIDEE. And this still doesn't explain the model-data differences at depth at the grass sites.

The forested sites also tend to be silt or clay loams, whereas the grass and shrub sites are more sandy loams. The latter has a higher ks, which results in a slightly faster decrease in the Ks downwards through the soil profile with the equation that accounts for decrease in Ks with soil compaction. However, this would counter the effect of changes in Ks with depth described above due to root zone bioturbation and we expect the effect to be much smaller at depth.

Despite not having a clearer answer to the reviewer's question, we agree we failed to explain or discuss any of the above mentioned points. Including these points would greatly aid a reader in understanding this issue. Therefore, we have adapted the manuscript text in several places to account for this.

In the description of the 11 layer hydrology in Section 2.2.2 we have added the following sentence:

"Ks increases exponentially with depth near the surface to account for increased soil porosity due to bioturbation by roots, and decreases exponentially with depth below 30cm to account for soil compaction (Ducharne et al., in prep)."

In addition, where we initially described the results of the differences in soil moisture at depth (original submission line 452 - Section 3.2), we have added the following:

"The smoother model temporal profile at depth at the forest sites compared to the sites with higher grass fraction is likely related to impact of rooting depth on exponential changes in Ks towards the surface (see Section 2.2.2). As the forests have deeper roots, the increase in Ks starts from a lower depth in the soil profile than the more grass-dominated sites, which in turn allows for a quicker infiltration of moisture to deeper layers and decreased simulated soil moisture temporal variability. However, this description of the model behaviour does not explain the model-data discrepancies."

And at the end of the same paragraph we have modified the original text so it now reads:

"Alternatively, it is possible that the model description of a vertical root density profile, which is used to calculate changes in Ks with depth, is too simplistic for semi-arid vegetation that typically have extensive shallow root systems that are better adapted for water-limited environments. It is also possible that assigning semi-arid tree and shrub types to temperate PFTs, as we have done in this study in the absence of semi-arid specific PFTs, has resulted in a root density decay factor that is too shallow.  Finally, changes in soil texture that in reality may occur much deeper in the soil could alter hydraulic conductivity parameters; in the model however, hydraulic conductivity only changes exponentially with depth owing to soil compaction (see Section 2.2.2)."

Finally, in various discussion section where we have talked either about the need for parameter calibration or issues with lateral redistribution of moisture, we have added the need to calibration root density profile or root zone plant water uptake parameters, and we've added that LSMs do not currently simulate extensive shallow root systems that are typical of semi-arid vegetation that is more adapted to water limited conditions. We hope these additions significantly improve the discussion related to model-data discrepancies in soil moisture at lower depths

• l.454: I was wondering why the model was not tested with more layers, say 100?

11-layers is a compromise between computational cost vs accuracy. In the initial development of the model they tested several different vertical soil discretizations and found that 11 layers was a good compromise for unsaturated soils (de Rosnay et al., 2000). In Campoy et al. (2013) they tested the effect of alternative soil bottom boundary conditions (including impermeable soil bottom and prescribed water table depth). This led them to increase the number of layers to 20 to describe the hydraulic gradients with enough accuracy; however, this is not necessary for unsaturated soil and additional layers

significantly increase the CPU requirement. Also, given 11-layers is the default version used in CMIP6, we based all our simulations on that version and chose not to test a different number of layers. We suggest adding in the following sentence to Section 2.2.2 (describing the 11-layer hydrology) for clarification on this point:

"De Rosnay et al. (2000) tested a number of different vertical soil discretizations in a 2m soil column and decided 11 layers was a good compromise between computational cost and accuracy in simulating vertical hydraulic gradients."

de Rosnay, P., Bruen, M. and Polcher, J.: Sensitivity of surface fluxes to the number of layers in the soil model used in GCMs, Geophysical Research Letters, 27(20), 3329–3332, doi:10.1029/2000gl011574, 2000.

---

## Author Comment (AC2) · 9 Apr 2020

Response to Reviewer #2

Review of Multi-variable, multi-configuration testing of ORCHIDEE land surface model water flux and storage estimates across semi-arid sites in the southwestern US by MacBean et al.

The manuscript by MacBean et al. deals with two different soil schematizations of the ORCHIDEE land surface model. One model set-up consists of a 2-layer soil schematization, whereas the other set-up makes use of an 11-layer soil scheme. In addition, resistance for soil evaporation was varied and bare soil fractions were reduced. The model set-ups were evaluated for several sites in the southwestern US. The authors show that adding a more detailed soil schematization improves the model results, especially regarding total evaporation and high frequency moisture dynamics.

The manuscript is generally well-written, and the figures are clear and of high quality. Most of the statements are supported by the data, and I think the article is interesting, because I agree that the hydrology in LSMs deserves attention. Nevertheless, after reading the article, I have several questions that remain.

We thank the reviewer very much for their useful and comprehensive review and constructive comments. We have attempted to provide detailed responses to all general and specific comments below. Please note that responses to the reviewer are in blue and additions to the manuscript are in red. Small changes to existing sentences are given in italics within the original sentence.

One of the first things the authors observe is that the forested sites show differences in transpiration and soil moisture. The soil schemes are different between the model runs, but rooting depths and rooting profiles are hardly mentioned by the authors. However, different rooting depths for both set-ups will have a strong influence on the findings of the authors. So how are these parameterized and are these different for the different model set-ups?

We have explained the rooting density profiles used for each PFT in Section 2.2.4 in the original text: "Whichever the soil hydrology model, beta depends on soil moisture and on the root density profile $R(z)=\exp(-c_j z)$, where z is the soil depth and cj (in m-1) is the thethe root density decay factor for PFT j. For a 2m soil profile, cj is set to 4.0 for grasses, 1.0 for temperate needleleaved trees and 0.8 for temperate broadleaved trees."

We have added "In both model versions" before "For a 2m soil profile".

We have kept the same rooting depths for both model versions, so this is not influencing the differences between the versions. We agree that changes in rooting depth can change the hydrological fluxes; however, this was not the aim of our paper so we do not want to test that further here. But we agree our discussion on roots was limited. We have also added in one extra point in the explanation of saturated hydraulic conductivity (in Section 2.2.2) that is related to roots following a comment by Reviewer #1 about infiltration differences between the tree and grass PFTs:

"Ks increases exponentially with depth near the surface to account for increased soil porosity due to bioturbation by roots"

Further to the changes made for Reviewer #1's infiltration comment (mentioned above), we have added some text in the results discussion to highlight that the root density decay factor may need to be adapted for semi-arid ecosystem PFTs:

" it is possible that the model description of a vertical root density profile, which is used to calculate changes in Ks with depth, is too simplistic for semi-arid vegetation that typically have extensive shallow root systems that are better adapted for water-limited environments. It is also possible that assigning semi-arid tree and shrub types to temperate PFTs, as we have done in this study in the absence of semi-arid specific PFTs, has resulted in a root density decay factor that is too shallow."

Finally, in various discussion section where we have talked either about the need for parameter calibration or issues with lateral redistribution of moisture, we have added the need to calibration root density profile or root zone plant water uptake parameters, and we've added that LSMs do not currently simulate extensive shallow root systems that are typical of semi-arid vegetation that is more adapted to water limited conditions. We hope these additions significantly improve the discussion related to rooting depths.

Similarly, the authors also often refer to the low and high elevation sites, but these also come with different vegetation types (forested vs grass/shrubland). I think the different vegetation types are much more the reason for the differences between the different sites, so I suggest that the authors distinguish more between the different vegetation types instead of the elevation, especially in the figures.

This is a fair point by the reviewer. We mainly followed this distinction because of the differences in precipitation regime and sources of available moisture throughout the year. The higher elevation sites are partly driven by snowmelt (as we discuss) as well as monsoon rains, whereas the lower elevation sites' moisture availability predominantly comes from monsoon rains. This is fundamental to explaining why forests exist at these higher elevation locations but not at all in the lower elevations. But we agree that for most of the text adding in high or low elevation is not needed, so we have removed a good chunk of those references. And to be clearer as to why we talk about differences in high and low elevation in addition to the type of vegetation we have added the following text into the site description in Section 2.1:

(for the low elevation sites): "The four grass- and shrub-dominated sites (US-SRG, US-SRM, US-Whs and US-Wkg) are located at low-elevation (<1600m) in southern Arizona with mean annual temperatures between 16 and 18°C (Biederman et al., 2017)." and "Moisture availability at these low elevation sites is predominantly driven by summer monsoon precipitation; however, winter and spring rains also contribute to the bi-modal growing seasons at these sites (Scott et al., 2015; Biederman et al., 2017)."

(for the high elevation sites): "Both high elevation sites experience cooler mean annual temperatures of 7.1 and 5.7°C respectively and are dominated by ponderosa pine (Anderson-Teixiera et al., 2010; Dore et al., 2012). The high elevation forested sites have two annual growing seasons with available moisture coming both from heavy winter snowfall (and subsequent spring snow melt) and summer monsoon storms. "

The authors also decided to model the soils with a thickness of 2 m, and mention that for the 11LAY- model drainage occurs as free gravitational flow at the bottom of the soil. This thickness, which is rather arbitrary, will also have a strong influence on the results as presented. The groundwater tables may influence the soil moisture profiles, and I wonder therefore if the authors have some idea on the groundwater tables at these sites. I do not object to this model choice of a 2 meter soil thickness, as you probably have to make an assumption here, but I believe it would be good to reflect on it, especially as the goal of the authors is to get the hydrology right, from which the groundwater is an important aspect and that is now basically assumed to be negligible.

We appreciate the reviewer's comment here but we are inclined to disagree that the ultimate goal is to make the hydrology *exactly* "right" because, as we discuss in the introduction and discussion, there are many aspects of the hydrology models that we already know are not implemented - groundwater being a good example. We have not set out to test every single parameter that contributes to the soil hydrology schemes (including soil depth and rooting density decay factor etc). The resulting manuscript would be too large; although we agree that both these parameters (and many more of the assumptions that go into the model, as well as known missing processes) could affect the model results. We have done our best to caveat and discuss these decisions and limitations in the discussion.

In an earlier version of the manuscript, we did have a small discussion section on the impact of soil depth (and texture). We removed it because many co-authors thought the paper was already long enough and we had to prioritize the points we discussed. However, if the reviewer would like we can add it back in. It read:

"**Soil texture and depth**
Total water content is unquestionably dependent on both the texture and depth of the soil, which is fixed at 2m in the 11-layer discretized hydrology model. However, semi-arid region soils are likely to be shallower with a higher concentration of rock and gravel (Grippa et al., 2017) – both of which are not represented in the ORCHIDEE soil texture classes. These two issues could introduce a bias in the soil moisture magnitude that is not easy to assess with the current observations. The inclusion of a mechanistic surface hydraulic conductivity parameter in the 11LAY version has allowed more water to be partitioned as runoff. However, it is nevertheless possible that too much water is still being held in the soil as a result of an incorrect soil depth and texture; in reality, more water might be partitioned to runoff or drainage. Ultimately, more different types of observations (such as runoff) are needed to test multiple different model versions."

In terms of groundwater at these sites, the depths to the groundwater are much deeper at these sites (10-100s m depth) and therefore groundwater access is not thought to be a large

contributor. We have added in a sentence into Section 2.1 describing the sites to add that groundwater depths are typically 10s to 100s metres deep.

Several previous studies have shown that in ORCHIDEE, only rather shallow WTDs are likely to significantly increase ET. As an example, Fig 4g in Wang et al. 2018 indicates that a forced WT at a depth of 1m from the soil surface can increase ET by more than 2.4 mm/d in SW USA. In this area, complementary work, but not yet published, shows that ET increases of 1% or more can be achieved with WTDs down to 5m or less (Ducharne et al., submitted).

Campoy A, Ducharne A, Cheruy F, Hourdin F, Polcher J, Dupont JC (2013). Response of land surface fluxes and precipitation to different soil bottom hydrological conditions in a general circulation model. JGR-Atmospheres, 118, 10,725–10,739, doi:10.1002/jgrd.50627.

Wang F, Ducharne A, Cheruy F, Lo MH, Grandpeix JL (2018). Impact of a shallow groundwater table on the global water cycle in the IPSL land-atmosphere coupled model, Climate Dynamics, 50, 3505-3522, doi:10.1007/s00382-017-3820-9

Ducharne A, Lo MH, Decharme B, Chien RY, Ghattas J, Colin J, Tyteca S, Cheruy F, Wu WY, Lan CW. Compared sensitivity of land surface fluxes to water table depth in three climate models. Submitted to Journal of Hydrometeorology.

However, in relation to one of Reviewer #1's comments we have proposed adding a few new sentence in the discussion about the need for groundwater to be included to the model:

"When discussing woody plant responses to drought, it is also worth noting that many LSMs to date are also missing any representation of groundwater (Clark et al., 2015). As described in Section 2.1, the water table is typically very deep (10s to 100s metres) at these sites. Previous modeling studies have shown that only rather shallow water tables (~1m) are likely to significantly increase ET in the SW US (e.g. by 2.4mmd-1 in Fig. 4g of Wang et al., 2018). However, the fact LSMs typically do not include adequate descriptions of groundwater access could impact their ability to simulate savanna ecosystem dry season water uptake given that drought deciduous shrubs in Mediterranean and semi-arid ecosystems are more resilient to droughts due to their ability to tap groundwater reserves (e.g. Miller et al., 2010). A new groundwater module is being developed for ORCHIDEE and will be tested in future studies."

The answer to the question of why we decided to use a thickness of 2m is similar to our answer to Reviewer #1's question of why 11 layers (and why not 100). The discretization of the soil column and the depth have been tested in previous studies testing the implementation of the finite difference integration needed to solve the Richards' equations in De Rosnay et al. (2000) and are now set as default parameters in the model. Aside from comparing these two schemes, and some additional tests related to the bare soil fraction and bare soil evaporation resistance term (which we decided to test based on the most obvious model-data discrepancies we found), we do not attempt to test any of the other options that may contribute to differences in water stores and fluxes for the reasons given above - it is too much for one paper. Rather, in the absence of insights that these

parameters may be the main cause of model-data discrepancies, we prefer to leave most of the parameters (such as soil depth) set to the default values that have been set based on these previous studies. This has the additional benefit of providing a reference as to how the default model (used in ongoing CMIP6 simulations) compares to observations for this region. We have added the following on to the end of the sentence that originally detailed that 2m soil depth was used for both versions:

"In this study, the depth of the soil for both schemes is set to 2m based on previous studies that tested the implementation of the soil hydrology schemes (de Rosnay and Polcher 1998; de Rosnay et al., 2000; de Rosnay et al., 2002). "

We have also added the following sentence to the hydrology model description in Section 2.2.2.

"De Rosnay et al. (2000) tested a number of different vertical soil discretizations in a 2m soil column and decided 11 layers was a good compromise between computational cost and accuracy in simulating vertical hydraulic gradients."

de Rosnay, P., Bruen, M. and Polcher, J.: Sensitivity of surface fluxes to the number of layers in the soil model used in GCMs, Geophysical Research Letters, 27(20), 3329–3332, doi:10.1029/2000gl011574, 2000.

There are also two methods used to derive ratios of transpiration/evaporation (Figure 6), but also here I have several questions. First, I wonder what the difference is between the two methods and if it is a fair comparison. There is also no data in the first months, and no data for US-Vcp, why is that? In addition, at US-Fuf, the data-derived estimates show that almost half of the total evaporation is transpiration, even during winter. At the same time, the site is described as having snow, at a high elevation, and one would therefore expect hardly any transpiration in winter here. This is also what the model actually does, it shows a strong reduction during winter. So how reliable are the estimated observations here?

The reviewer is absolutely right that we did not outline the difference between the two methods to derive the T/ET ratios. We also did not explain the S&B17 method well and we did not explain the Zhou et al. (2016) method at all in the methods. We also did not provide Zhou estimates for US-Vcp. These were oversights by the authors. We have corrected all these issues in the revised manuscript but the reasons are explained below.

Initially, we used Scott and Biederman (2017) for the low elevation more water-limited shrub- and grass sites because it was deemed that this method is better at detecting T/ET for water limited sites following reasons given in that paper, namely that "Because we do not force the regression through the origin, our approach is more appropriate for water-limited sites, where it is often found that the ET ≠ 0 (i.e., the intercept) for GEP = 0 [Biederman et al., 2016].". However, the method does not work well at the less water-limited forested sites - there is only a month or two where there are significant linear fits and where those fits yield positive ET axis intercepts. Indeed, Scott and Biederman had no intention of this method

being universally used but just found that it worked particularly well for their sites (low elevation shrub and grassland). Thus, for the Fuf sites we used the Zhou method.

At the forested sites we only keep the Zhou et al. estimates for the reasons given above and at the lower elevation grass and shrub sites we now give estimates from both Zhou et al. (2016) and Scott and Biederman (2017) to show that indeed there is uncertainty in estimating T/ET ratios based on assumptions in different methods. We detail both of the these methods and our reasoning for having only Zhou at the forested sites and both at the grassland sites in Section 2.3.1 ("Site-level meteorological and eddy covariance data and processing") with the following sentence:

"Estimates of T/ET ratios were derived from Zhou et al. (2016) for the forested sites, and both Zhou et al. (2016) and Scott and Biederman (2017) at the more water-limited low elevation grass- and shrub-dominated sites. Zhou et al. (2016) (hereafter Z16) used eddy covariance tower GPP, ET and vapor pressure deficit (VPD) data to estimate T/ET ratios based on the ratio of the actual or apparent underlying water use efficiency (uWUEa) to the potential uWUE (uWUEp). uWUEa is calculated based on a linear regression between ET and GPP.VPD0.5 at observation timescales for a given site, whereas uWUEp was calculated based on a quantile regression between ET and GPP.VPD0.5 using all the half-hourly data for a given site. Scott and Biederman (2017) (hereafter SB17) developed a new method to estimate average monthly T/ET from eddy covariance data that was more specifically designed for the most water-limited sites. The SB17 method is based on a linear regression between monthly GPP and ET across all site years. One of the main differences between the Z16 and SB17 method is that the regression between GPP and ET is not forced through the origin in SB17 because at water-limited sites it is often the case that ET ≠ 0 when GPP = zero (Biederman et al., 2016). The Z16 method also assumes the uWUEp is when T/ET = 1, which rarely occurs in water-limited environments (Scott and Biederman, 2017)."

Based on the fact we now have also have T/ET estimates for US-Vcp and we also have two T/ET estimates for the grass and shrub dominated sites, we have adapted Figure 6 (and its caption) to include both estimates for the grass- and shrub-dominated sites and included the Zhou et al. (2016) method for the US-Vcp site. We have also altered the description of these results in Section 3.3 as described below.

For the forested sites, we have edited this paragraph: "Further support for the suggestion that modelled E is overestimated comes from examining the T/ET ratios. Although both E and T increase in the US-Fuf 11LAY simulations (compared to the 2LAY – Fig. S3a) – due to the increase in soil moisture (as previously described in Section 3.1 and Figs. 2 and S2a) – the larger increase in 11LAY E compared to T resulted in lower 11LAY T/ET ratios (Fig. S3a). The seasonal trajectory of T/ET ratios at US-Fuf appear to match data-derived estimates following the Zhou et al. (2016) method: the ratio peaks in the Spring before decreasing in July, with monsoon period T/ET values that are on average lower than the spring (Fig. 6). However, the magnitude of T/ET ratios are too low in all seasons given the 100% tree cover at this site with a LAI ~2.4. Whilst low spring 11LAY T/ET ratios may be due to overestimated E as a result of higher soil moisture and underestimated snow cover, the

generally low bias in T/ET ratios may also be due to the fact there is no bare soil evaporation resistance term included in the default 11LAY version."

to include a broader description of issues at the forested sites now we have T/ET estimates for US-Vcp as well as US-Fuf. The edited text now reads:

"Further support for the suggestion that modelled spring E is overestimated comes from comparing the model to estimated T/ET ratios (Fig. 6). Although both E and T increase in the US-Fuf and US-Vcp 11LAY simulations (compared to the 2LAY – Fig. S3a and b) due to the increase in soil moisture (as previously described in Section 3.1 and Figs. 2 and S2a), the stronger increase in 11LAY E compared to T resulted in lower 11LAY T/ET ratios across all seasons (Fig. S3a and b). While the model captures the bimodal seasonality at the forested sites as seen in the Z16 data-derived estimates (Fig. 6), the magnitude of model T/ET ratios appear to be too low in all seasons given the 100% tree cover at these sites with a maximum LAI of ~2.4. Whilst low spring 11LAY T/ET ratios at may be due to overestimated E as a result of higher soil moisture and underestimated snow cover, the generally low bias in T/ET ratios across all seasons at both US-Fuf and US-Vcp may also point to the issue that no bare soil evaporation resistance term is included in the default 11LAY version. This may also explain why the model T/ET ratios do not increase as rapidly as estimated values at the start of the monsoon (Fig. 6). Discrepancies in the timing of T/ET ratio peak and troughs between the model and data-derived estimates at the forested sites could also be due to the fact evergreen PFTs have no associated phenology modules in ORCHIDEE; instead, changes in LAI are just only subject to leaf turnover as a result of leaf longevity, which may be an oversimplification."

One of the main changes to the results following the inclusion of both methods is in the paragraph relating to US-SRM spring T/ET given that the model now lies in between the two estimates for this time period. Therefore, we have replaced this original text: "We can also glean some information on whether T or E (or both) are be responsible for the 11LAY overestimate of springtime ET at US-SRM by comparing modelled T/ET ratios against data-derived estimates. Observed T/ET ratios at the low-elevation sites were derived from independent eddy covariance data following the method of Scott and Biederman (2017) (Fig. 6). The observed spring T/ET at US-SRM is slightly underestimated by the model (Fig. 6). Given that T/ET ratios are underestimated by the model but ET is overestimated by the model, it is probable that spring E at this site is too high. Spring T could also be overestimated at US-SRM due potentially due to an overestimate in LAI (Fig. S5); however, the positive bias in E must be larger than the bias in T. If model LAI at US-SRM is too high during the spring, it is impossible to determine whether the shrub or grass LAI are inaccurate without independent, accurate estimates of seasonal leaf area for each vegetation type; however, in the field the spring C4 grass LAI is typically half that of its monsoon peak (R.L. Scott – pers. comm.) – a pattern not seen in the model (Fig. S6)."

with

"At US-SRM, the modelled spring T/ET ratio overestimates the Z16 estimate and underestimates the SB17 estimate (Fig. 6). The current state of the art is that different methods for estimating T/ET typically compare well in terms of seasonality but differ in absolute magnitude; therefore, the uncertainty in T/ET magnitude during the spring at US-SRM makes it difficult to glean any information on whether T or E (or both) are be responsible for the 11LAY overestimate of springtime ET (Fig. S3c). If the SB17 method is more accurate, then it is probable that modelled spring E at this site is too high. However, if the Z16 estimate is accurate, then it is likely that spring T is overestimated at US-SRM, potentially due to an overestimate in LAI. The model-data bias in spring mean monthly ET is well correlated (0.XX) with spring mean LAI at US-SRM (Fig. S5). If model LAI at US-SRM is too high during the spring, it is impossible to determine whether the shrub or grass LAI are inaccurate without independent, accurate estimates of seasonal leaf area for each vegetation type, which are not available at present; however, in the field the spring C4 grass LAI is typically half that of its monsoon peak – a pattern not seen in the model (Fig. S6). We will test both of these hypotheses (overestimate in either T or E) in Section 3.4."

We have also edited the following original text: "Data-derived T/ET ratios also help to diagnose why the 11LAY model underestimates monsoon ET at the low-elevation shrub sites (US-SRM and US-Whs– Figs. S3 c-d). Fig. 6 shows that the 11LAY model also underestimates monthly T/ET ratios, and furthermore, that the model does not capture the correct temporal trajectory (Fig. 6). Although the earlier summer drop in T/ET ratios in the 11LAY compared to the 2LAY simulations at grass and shrubland sites (Figs. S3 c-f) does result in a better match in ET between the model and the observations (Fig. 3), the 11LAY T/ET ratios are slightly out of phase. Observed T/ET ratios decline in June during the hottest, driest month, whereas model values decrease one month later in July (Fig. 6). Furthermore, the ratios do not increase as rapidly as observed during the wet monsoon period (July – September).
The underestimate in modelled monsoon T/ET ratios across all grassland and shrubland sites (and likely at US-Fuf and US-Vcp) suggests either that transpiration is too low or bare soil evaporation is too high. At the shrubland sites (US-SRM and US- 500 Whs), both monsoon ET and T/ET are underestimated; therefore, for these sites it is plausible that the dominant cause is a lack of transpiring leaf area. Certainly, monsoon model-data ET biases are better correlated with LAI at shrubland sites compared to grassland sites (Fig. S7). The underestimate in modelled monsoon period leaf area could either be: i) an underestimate of maximum LAI for either grasses or shrubs; or ii) due to the fact the static vegetation fractions prescribed in the model do not allow for an increase in vegetation cover during the wet season (e.g. the lack grass growth in the model in interstitial bare soil 505 areas). In contrast, at the grassland sites (US-SRG and US-Wkg) monsoon ET is well approximated by the 11LAY model; thus, the underestimate in T/ET ratios suggests that both the transpiration is too low and the bare soil evaporation too high." to include both T/ET methods, to make the text more understandable, and to provide further explanation of the "out of phase" seasonality in T/ET ratios at the low elevation sites. The new text is:

"At the low elevation grass- and shrub-dominated sites, both data-derived estimates of T/ET agree on their seasonality and sign with respect to the model magnitude during the

monsoon. Given this agreement, both sets of estimated values can help to diagnose why the 11LAY model underestimates monsoon peak ET at the low-elevation shrub sites (US-SRM and US-Whs– Figs. S3 c-d). Fig. 6 shows that the 11LAY model also underestimates both Z16 and SB18 monthly monsoon period T/ET estimates across all low elevation sites. The underestimate in modelled monsoon T/ET ratios across all grassland and shrubland sites suggests either that T is too low or E is too high. At the shrubland sites (US-SRM and US-Whs), both monsoon ET and T/ET are underestimated; therefore, for these sites it is plausible that the dominant cause is a lack of transpiring leaf area. As was the case for spring ET at US-SRM, monsoon model-data ET biases are better correlated with LAI at shrubland sites compared to grassland sites (Fig. S7). In contrast, at the grassland sites (US-SRG and US-Wkg) monsoon ET is well approximated by the 11LAY model; thus, the underestimate in T/ET ratios suggests that both the transpiration is too low and the bare soil evaporation too high.

Furthermore, although the 11LAY does capture the decrease in ET during the hot, dry period of May to June (which is a significant improvement compared to the 2LAY – see Section 3.1), the 11LAY T/ET ratios are slightly out of phase with the estimated values. Both data-derived estimates agree that T/ET ratios at all low elevation sites decline in June during the hottest, driest month (as expected); however, the model T/ET ratios reach a minimum one month later in July (Fig. 6). This one month lag in model T/ET ratios is apparent despite the fact that the ET minimum is accurately captured by the model (Figs. 3b and S3). The modelled T/ET ratios also do not increase as rapidly as both estimates during the wet monsoon period (July – September), which can be explained by the fact that the model E at the start of the monsoon increases much more rapidly than modelled T. Taken together, these results suggest that LAI is not increasing rapidly enough after the start of monsoon rains (see Fig. S6), resulting in low biased T/ET ratios in July. Meanwhile the increase in available moisture from monsoon rains is causing a biased high model E that compensates for the lower T. These compensating errors result in accurate ET simulations. The underestimate in modelled leaf area during the monsoon could either be: i) incorrect timing of LAI growth for either grasses or shrubs and an underestimate of peak LAI; and/or ii) due to the fact the static vegetation fractions prescribed in the model do not allow for an increase in vegetation cover during the wet season (e.g. the model lacks the ability to grow grass in interstitial bare soil areas)."

We have also added the following sentence in the abstract:
"However, discrepancies in the timing of the transition from minimum T/ET ratios during the hot, dry May-June period to high values during the summer monsoon period in July-August could point towards incorrect simulations of seasonal leaf phenology. "

In terms of winter values at US-Fuf (and now US-Vcp), my co-author (Russ Scott) left out months where GPP is very low because both estimation procedures rely on the relationship between ET and GPP, very low and low variability GPP (in the winter) results in a poor relationship between these two quantities. We have added the following sentence explaining this into Section 2.3.1:

"T/ET ratio estimates are omitted in certain winter months when very low GPP and limited variability in GPP results in poor regression relationships."

Thus, the data-derived estimates are not given for US-Fuf during the winter months when there is a lot of snow so we are not relying on the T/ET estimates for this period. And we agree with the reviewer that the model is likely right on simulating low T/ET values during this period.

The authors often argue that snow is not correctly modelled, and I think the statement of the authors on page 14, lines 442-444 is important here. Snow usually falls within a temperature range around 0 degrees Celsius, and the authors mention that the results improved by changing the temperature threshold, but these results are not shown, so please add these results.

We were initially reluctant to add these snow test results because a) we didn't show the results of the other snow-related tests we did (described in the original lines 436-438) and b) because there are already a lot of figures in this paper and the figure for this snow forcing test was deemed to be of lower importance. However, we have now added this test to the supplementary (Figure S5 - please see below). Please note also that we have slightly lengthened and added to the description of these snow-related results in Section 3.2 (and changed Figure 5) following some suggestions from Reviewer 1. We hope that the description and discussion of these particular results is more detailed and nuanced. The edited text is:

"In contrast, the temporal mismatch between the observations and the model in the uppermost layer is higher at the forest sites. The US-Fuf and US-Vcp 11LAY simulations appear to compare reasonably well with observations in the upper 2cm of the soil from June through to the end of November (end of September in the case of US-Vcp) (Fig. 4). However, in some years the model appears to overestimate the VWC at both sites during the winter months (positive model-data bias), and underestimate the observed VWC during the spring months (negative model-data bias), particularly at US-Fuf. Although US-Fuf and US-Vcp are semi-arid sites, their high-elevation means that during winter precipitation falls as snow; therefore, these apparent model biases may be related to: i) the ORCHIDEE snow scheme; ii) incorrect snowfall meteorological forcing; and/or iii) incorrect soil moisture measurements under a snow pack. During the early winter period the model soil moisture increases rapidly as the snowpack melts and is replenished by new snowfall, whereas the observed soil moisture response is often slower (Fig. 5a and b light blue zones). This often coincides with periods when the surface temperature in the model is below 0°C (Fig. 5 bottom panel), suggesting that in reality soil freezing may be negatively biasing the soil moisture measurements. An alternative explanation is that ORCHIDEE overestimates snow cover (and therefore snow melt and soil moisture) at the forest sites because it is assumed that snow is evenly distributed across the grid cell, whereas in reality the snow mass/depth is lower under the forest canopy than in the clearings.
At US-Fuf, it appears that the model melts snow quite rapidly after the main period of snowfall (Fig. 5a light green zones). Once all the snow has melted, the model soil moisture also declines; however, the observed soil moisture often remains high throughout the spring – causing a negative model-data bias (Fig. 5a). Unlike US-Fuf, a similar negative model-data

bias at US-Vcp often coincides with periods when snow is still falling, although the amount is typically lower (Fig. 5b light green zones); however, the model does not always simulate a high snow mass during these periods. These periods coincide with rising surface temperature above 0°C. Although snow cover, mass, or depth data have not been collected at these sites, snow typically remains on the ground until late spring after winters with heavy snowfall, suggesting that the continued existence of a snow pack and slower snow melt that replenishes soil moisture until late spring when all the snow melts. Therefore, the lack of a simulated snow pack into late spring could explain the negative model-data soil moisture bias. To test the hypothesis that the model melts or sublimates snow too rapidly, thereby limiting the duration of the snowpack and also allowing surface temperatures to rise, we altered the model to artificially increase snow albedo and decrease the amount of sublimation; however, these tests had little impact on the rate of snow melt or the duration of snow cover (results not shown). Aside from model structural or parametric error, it is possible that there is an error in the meteorological forcing data. Rain gauges may underestimate the actual snowfall amount during the periods when it is snowing (Rasmussen et al., 2012; Chubb et al., 2015). If the snowfall is actually higher than is measured, it may in reality lead to a longer lasting snowpack than is estimated by the model. To test this hypothesis, we artificially increased the meteorological forcing snowfall amount by ten times and re-ran the simulations. Although this artificial increase is likely exaggerated, the result was an improvement in the modelled springtime soil moisture estimates at US-Fuf (Fig. S5). However, the same test increased positive model-data bias in the early winter increased at US-Fuf, and degraded the model simulations at US-Vcp. This preliminary test suggests that inaccurate snowfall forcing estimates may play a role in causing any negative model-data bias spring soil VWC but more investigation is needed to accurately diagnose the cause of the springtime negative model-data bias."

To better match this text we have updated Figure 5 to only include the pertinent variables (and have added surface temperature) and we have added an extra supplementary figure (S5) to show the results of the increased snow forcing (as per a comment from Reviewer 2):

Figure 5: a) US-Fuf and b) US-Vcp 11LAY (blue curve) daily time series (2007-2010) of model versus re-scaled (via linear CDF matching) observed volumetric soil water content (middle panel SWC – m3m-3) (black curve), compared to simulated snow mass (top panel) and surface temperature (bottom panel). Snowfall is also shown as grey lines in the SWC time series. In the bottom panel the grey horizontal dashed line shows 0°C threshold.

[Figure]

Figure S5: Linear regressions between spring (March-April) mean monthly LAI (m2m-2) and spring mean monthly ET (mmmonth-1) model-data misfits for each site. The dominant PFT is given in brackets for each site. See Table 1 for PFT acronyms.

[Figure]

We have also modified this sentence in the discussion:
"More specifically, more information on snow cover, depth or mass, particularly under closed forest canopies, would be useful to  diagnose potential sources of bias in the snowfall simulations. "

We have also added this sentence into the abstract:
"Biases in winter and spring soil moisture at the forest sites could be explained by inaccurate soil moisture data during periods of soil freezing and underestimated snow forcing data."

Finally, we also updated a sentence in the conclusions to reflect both the negative and positive model-data biases in soil moisture at the forested sites could be related to snowfall issues:

"Remaining discrepancies in both overestimated and underestimated winter and spring soil moisture at high-elevation semi-arid forested sites might be respectively related to issues with soil moisture data during periods of soil freezing and underestimated snowfall forcing data causing a limited duration snowpack, with consequent implications for predictions of water availability in regions that rely on springtime snowmelt."

In addition, the reasoning of the authors regarding the snow modelling relates to the overestimation of ET at US-Fuf for 11LAY, but this does not happen for 2LAY. At the same time, US-Vcp also shows an underestimation and has snow, so it does not seem to be a consistent problem here.

We did mistakenly say that the overestimation of spring ET was for Fuf *and* Vcp - we have corrected that now to only refer to Fuf. But we agree with the reviewer that it is an incomplete explanation (and doesn't help to explain Vcp). We did try to emphasize this in the original text by moving on later in the paragraph to use T/ET ratios to try to explain all ET issues at both sites. At both these sites the T/ET ratios are lower than the estimated values (we see this now we have Vcp included in these estimates); thus, we go to on say that this could be due to a lack of T or a possible overestimation of E at both sites due to the lack of the bare soil resistance term and/or issues with LAI and the phenology. We test the former further hypothesis in Section 3.4. So, while it was our intention in Section 3.2 to say the underestimate in spring *soil moisture* at Fuf and Vcp was due to incorrect snowfall (and we have updated that text - see above), the link between an underestimated snowpack and overestimated spring ET at Fuf in Section 3.3 is just one of the hypotheses we put forward for the errors in spring ET. It was not our intention to say that snowfall *is* definitively the factor that contributes to overestimated spring ET at Fuf - but more that it is one possible explanation  (and we believe it does not read that way given we say "The lack of a persistent snowpack in the model during this period could explain the positive bias in spring ET because in reality the presence of snow would suppress bare soil evaporation"). We did not give this as a definitive cause of the ET in the abstract and conclusions. Many interacting factors likely go into why spring ET is overestimated at Fuf (and indeed, not at Vcp), which we try to emphasize. Unfortunately it is difficult to test all of these hypotheses - we have tested one in Section 3.4. We have added this sentence in the conclusions (after the sentence about the possible role of snowfall issues in soil moisture model-data biases detailed in the answer to the previous comment) to clarify that there are multiple possible reasons why there are ET discrepancies at the forest sites, not just reasons related to snowfall:

"However, biases in soil moisture at both the forested sites do not translate into the same biases in modelled ET at the forest sites, suggesting other factors such as issues in evergreen phenology/LAI simulations or the lack of resistance to bare soil evaporation may also play a role."

Do the two model set-ups use the same snow module and are the parameterizations the same for the different sites?

Yes, they are. But even though the snow model is the same in the 2 configurations, the different hydrology simulations at the two sites then impacts the soil thermal processes differently because the soil properties (heat capacity and thermal conductivity) depend on soil water content. Also we don't expect complete snow coverage all the time at each site (we can see in Fig. 5 snow comes and goes throughout the winter period), so the overall surface temperature may be different, leading to different snow melt, snowpack etc at the two sites.

As suggestion, it could also help the authors to look at remotely sensed snow cover products such as MODIS10A. These products are relatively easy and could provide already a quick check if the snow temporal dynamics are captured in the model.

We would love to have data that could help us test our hypotheses the model is underestimating snow pack. Indeed we said this in original lines 470-471 (directly after the sentence quoted above): "To accurately diagnose this issue, we would need further information on snow mass or depth". However, as we mentioned, we considered that we would need information on snow mass or depth (to validate the top panel in Figs. 5 a and b), not snow cover (given these are site simulations and we're not examining spatial heterogeneity), which is what the MOD10A product is. There are satellite products of snow water equivalent that might be more useful in validating snow mass/depth but as far as we understand these products are only available at very coarse spatial resolutions (>=25km). However, after considering the reviewer's suggestion we agreed that the MOD10A snow cover at 500m, while not helping us with snow amount, would help with evaluating snow duration and indeed it did to some extent. Therefore, we have included it in the new Fig S5 which also shows the result of the increased snow forcing test we did, which we have now included as per the reviewer's justified request.

My most important point relates however to the fact that the article misses sometimes a bit focus regarding the goal of the authors, which is comparing a simple two-layer scheme with a more complex scheme in order to improve the hydrology. A couple of times the authors only look at the 11LAY- results, or do not use observations to assess if there are any improvements. For example, the authors only compare 11LAY with the soil moisture measurements (Fig. 4,5, paragraph 3.2). I do understand why, as the authors explain this in paragraph 2.3.2, but I am not sure if there is any point in evaluating 11LAY-results with soil moisture data, if you can not do the same for 2LAY. After reading paragraph 2.2.2 I still think the authors could at least compare also the temporal dynamics in the 2LAY-model, as this is what the authors do anyway with equation 5.
Similarly, a large part of paragraph 3.1 gives a description on the differences between the two model set-ups, and discusses Figure 1. Nevertheless, without any idea on how reality looks like, it is hard to really get an understanding on what is actually better. So I am not sure if this part of the paragraph really adds something, unless the authors add some observations. The authors do have soil moisture data and flux tower data, so I suggest to add these to Figure 1.
One of the main conclusions is also that the high frequency soil moisture dynamics are more realistic for the 11LAY-model. This conclusion is however not supported by the data as shown, there is no figure in the manuscript and supplementary material that actually compares both 11LAY and 2LAY soil moisture values with observations, so you can unfortunately not state that 11LAY is clearly better here. The conclusion that surface runoff is more realistic (P21.L669) came even as a bigger surprise to me, I believe there is no data on surface runoff in the manuscript, or I must have completely missed this.

We appreciate the reviewer's comments. Indeed, we debated whether or not to add observations to Figure 2 when comparing the 2 vs 11 layer upper soil moisture, and in an earlier version we did have such a comparison. For the 2LAY version we only have the option to compare either the upper layer moisture (0-10cm) or the total column of 2m (or bottom 1.9m). In figure 2 we really wanted to look at the upper layer moisture given this is predominantly a plot about what is happening for ET (and its component fluxes and relevant

processes) and the upper layer moisture comes from. However, the issue with the 2LAY upper layer moisture is that that layer can disappear entirely (as we describe in Section 2.2.2), which is why it has this very noisy temporal profile that can decrease to zero. Even by looking at the temporal dynamics of the 2 vs 11 layer upper layer compared to the ET we can see that the ET temporal dynamics are mostly related to this upper layer moisture. We also thought of having a further soil moisture plot that shows the much smoother total column soil moisture temporal dynamics (to highlight again that the ET temporal dynamics are dominated by the upper layer, and not the total column, but that would have added a 7th panel to Figure 2, which we felt was too much (but we can add that in if the reviewer thinks that would help explain this point.

With all this in mind, we thought a) that comparing the 2 layer upper moisture to the observations is tricky because of this issue that the layer can disappear entirely, and b) that it takes away from the main point, which is that the temporal dynamics of the 2lay reflect this issue that the layer can disappear entirely and are therefore almost by design not realistic when comparing to observations. Furthermore, because of the reasons we highlight in Section 2.3.2, we do not wish to compare absolute soil moisture values in any given layer directly to observations (and we are happy the reviewer understands these points), therefore we need to do the linear CDF matching. We can absolutely do this for the 2 layer upper layer soil moisture (as well as the 11 layer equivalent) and compare to the observations that most represent this 0-10cm interval (see the adapted figure below), but we want to stress that this is not as direct a comparison as we make for the 11-layer comparison in Figure 4 and is somewhat more subject to the issues we describe above (essentially, less of an "apples to apples" comparison than we have for Figure 4). However, we have done the CDF matching and adapted the figure, and we agree with the reviewer that it certainly helps to make our case that the 11 layer does indeed better capture the observed temporal characteristics (the fluctuations are much more realistic). We hope they see our point more clearly now and we propose keeping this revised figure 2 and will make any necessary changes in the text.

[Figure]

However, we choose not to add observations into Figure 1 because this is the only figure we have where we look at the overall changes in the *absolute values* of total soil moisture (as opposed to re-scaling the model to match the observations using linear CDF matching (in original equation 5 and as explained in Section 2.3.2). As we also explain in Section 2.3.2 the observations come from different depths at each site and it is hard to know over which depth the different soil moisture probes measure. Furthermore, we do not have observations below 75cm (and much shallower at some sites - Table 2). Therefore, we do not have estimates of how much water content there is in the total soil column and thus we cannot put the observed total column moisture in Figure 1. Even if we were to convert the total column soil moisture to volumetric soil moisture, we still do not know a column average volumetric soil moisture content. We think it would be heavily biased if we were to simply average over the limited depths we have. We have added the following sentence into Section 2.3.2 for further clarification of this point:

"Given the maximum depth of the soil moisture measurements is 75cm (and is much shallower at some sites) we cannot use these measurements to estimate a total 2m soil column volumetric soil moisture content."

If we do add the re-scaled soil moisture observations into the upper layer soil moisture comparison plot in Fig. 2 (bottom panel - see above), we will also modify the following sentence of this section:

"Instead, we only used these measurements to evaluate the 11LAY model and 2LAY upper layer soil moisture (calculated for 0-10cm) because, unlike the 2LAY model, with the 11LAY version of the model we have model estimates of soil moisture at discrete soil depths."

We have also added this sentence to the caption of Figure 1:

"For soil moisture, the absolute values of total water content for the upper layer and total 2m column are shown for both model versions, i.e. the simulations have not been re-scaled to match the temporal dynamics of the observations (as described in Section 2.3.2); therefore, soil moisture observations are not shown. Observations are only shown for ET."

Finally, we agree with the reviewer about the claim that surface runoff is more realistic, given we do not actually show any data in the manuscript. The fact that claim appears to be overstated is perhaps more due to our lack of properly articulating what we meant here, and the lack of referencing other studies when discussing the runoff and drainage results (although data is still limited). We did refer to two studies from US-Fuf and US-SRM that discuss low drainage results and the fact that Precipitation is mostly accounted for by ET.

In the revisions (and in response to another comment by Reviewer #1 about whether limited drainage at the forested sites was plausible), we have also added this sentence: "In general, all these semi-arid sites have very little precipitation that is not accounted for by ET at the annual scale (Biederman et al., 2017 Table S1)."

Table S1 in Biederman also shows that most precipitation is accounted for by ET across all these sites; therefore, although we don't explicitly have runoff and drainage data we feel these data do serve to highlight that the original 2LAY estimates of total runoff were *likely* too high and that the 11LAY values appear to be more plausible.

Given these points, we could modify this sentence in the conclusions in the following way:

"Associated changes in the calculations of runoff, soil moisture infiltration, and bottom layer drainage also appear to result in more plausible (lower) estimates of total runoff (surface runoff plus drainage) at the forest sites given that across all these semi-arid sites, most precipitation is accounted for by ET at the annual scale."

However, if the reviewer feels this is still too exaggerated a claim for the conclusions we will remove the sentence entirely.

Concluding, the manuscript is interesting, but the authors should make sure they build a systematic case why one hydrological schematization should be preferred over another. I have sometimes the feeling the authors have a preference for the 11LAY-scheme, but I think it is important to objectively assess the performance of both set-ups. I hope my comments are useful for the authors and look forward to an improved manuscript.

We are glad the reviewer finds the manuscript interesting and appreciate their thoughtful comments. We hope that by addressing their comments (above and below) we have helped to clarify our objectives, to better align the results with those objectives, and to provide conclusions that better support the results. In particular, we hope that the modifications we propose to figure 2 help to support one of our main conclusions that the 11LAY does a better

job in terms of capturing the ET temporal dynamics, and that it is not simply that we prefer the 11LAY version. We also hope that the discussion we provided serves to highlight that we realize there are many remaining caveats (model issues, missing processes) in how we currently model hydrology using the mechanistic 11LAY model, but that dealing with all of these issues is beyond the scope of the current paper.

Minor comments

P1.L36. Results better → results in a better?

Changed - thank you.

P2.L62. A evaporation → an evaporation

Changed - thank you.

P3.L79 have been rarely been → have rarely been

Changed - thank you.

P4.L115. Define PFT

Done - thank you.

P6. L187. What do you mean with soil tile? The spatial distribution of different soils within a grid cell?

No it corresponds to the number of water columns for which each separate water flux is calculated. We have modified this sentence to read: "Independent water budgets are calculated for each "soil tile", which define separate water columns within a grid cell.". We hope that with this modification and the original following sentences of "In the 2-layer scheme, soil tiles correspond to PFTs; therefore, a separate water budget is calculated for each PFT within the grid cell. In the 11-layer scheme there are three soil tiles: one gathering all tree PFTs, one gathering grasses and crops, and the third as bare soil." that the meaning of soil tile is now clearer.

P6.L189. "all three PFT's" –> It is mentioned before that there are 12, so why three now?

This actually reads "all **tree** PFTs"

P6.L191. Related parameters) → remove ")"

Changed - thank you.

P7.L210. At al → et al

Changed - thank you.

P7.L217. At al → et al

Changed - thank you.

P8.L227. Seems a bit arbitrary to me, why these numbers?

These are very classical values, often given as -33kPa and -1500kPa, or -0.33 and -15 bars, see for instance Rawls et al. (1982) and Verhoef and Gregorio (2014)

Rawls, W. J., Brakensiek, D. L., & Saxtonn, K. E. (1982). Estimation of soil water properties. Transactions of the ASAE, 25(5), 1316-1320. Cited 1894 times according to Google Scholar.

Verhoef, A., and Gregorio, E. (2014). Modeling plant transpiration under limited soil water: Comparison of different plant and soil hydraulic parameterizations and preliminary implications for their use in land surface models, Agricultural and Forest Meteorology, 191, 22-32, https://doi.org/10.1016/j.agrformet.2014.02.009.

Bonan (2002) gives the same potential for wilting point, but -1m for field capacity (which is very close to -3.3m given the wide range of soil water potential in an unsaturated soil: from 0 to much less than -150m).

Bonan, G. (2015). Ecological climatology: concepts and applications. Cambridge University Press. Cited 1285 times in Google Scholar.

We have added a reference to Ducharne et al. (in prep.) here that explains this point (extensive description of the latest version of the ORCHIDEE soil hydrology). We can add references to the above if needed.

P8.L229. Has been test → have been tested
Changed - thank you.

P8.L256. The the root density → the root density
Changed - thank you.

P8.L256-257. Why these values? What are they based on? Eq3. Please define and describe also h_t and d

$h_d^t$ is one variable that is "dry soil height of the topmost soil layer" (originally defined in lines 259-260).

These values were selected to get a higher root density for forests than for low vegetation and have not been calibrated against field data. We have added/modified lines in the discussion on the need for calibration of these parameters, e.g.:

In the discussion section on Issues with modelling vegetation dynamics in semi-arid ecosystems:

"Alternatively, it may be that other model parameters and processes involved in leaf growth – for example phenology, root zone plant water uptake, and photosynthesis-related parameters – are inaccurate and in need of statistical calibration (e.g. MacBean et al., 2015). "

And in the discussion section on bare soil evaporation: "
"It is possible that the bare soil resistance is only part of the solution, and that the simulation of ET and its component fluxes could be fixed with both a more realistic representation of semi-arid phenology or vegetation fractional cover at both grass and shrub dominated sites (as discussed above) and/or a statistical calibration of relevant vegetation, root density, soil hydraulic parameters (e.g. Shi et al., 2015)."

However, there are limited root density decay factor parameters, so these parameters would have to be calibrated by means of indirect observations (such as ET). Although it is beyond the scope of the present study, we do plan to conduct future studies. Calibration of such parameters has not yet been attempted, and based on the data assimilation experience of NM, investigating how best to optimize new processes/parameters will take time. Thus, these studies will be presented in future papers.

P8.L267. Is T here transpiration? Please define.
We do define that in Section 2.2.1. We can define it again here.

Eq5. Please define your variables
Done - thank you.

P12.L351. Higher compared to the other sites? It is not higher than the 11LAY-scheme.
We thank the reviewer for spotting the error in this sentence. It now reads: "In the 2LAY simulations, the upper layer soil moisture is similar across all sites; whereas, in the 11LAY simulations the difference between the high elevation forest sites and low elevation grass and shrub sites has increased."

P12.L380. I do not see any values going to 0 in Figure S1 for VWC in the upper 2m. Basically 2LAY seems to drain the upper layer faster.
The sentence actually refers to the upper layer (top 10cm) not the upper 2m. The whole soil column is 2m deep. The 2LAY upper layer (top 10cm) does decrease to 0 and this can be seen in Figure S2 (and Fig. 2, which we refer to in this sentence (not Fig. S1): "Whereas the 2LAY
upper layer soil moisture simulations at all sites fluctuate considerably between field capacity and zero throughout the year – including during dry periods with no rain – the temporal dynamics of the 11LAY upper layer moisture simulations correspond more directly to the timing of rainfall events (see Fig. 2 bottom panel for an example at 3 sites in 2009 and Fig. S2 for the complete time series for each site)."

P12.L383-384. I do not think you can conclude 11LAY is better based on the data as shown, there are no observations shown of soil moisture in Fig. 2.

We have added observations into Fig. 2 (presented above) and we have also modified all the Fig. S2 plots. Please see the above discussion (in the reviewer's main comments on this point).

P14.L421. Fig 4 →Fig. 4
Changed - thank you.

P14.L422. So which sites in fig4 do you mean? It's easier to add the names, then the reader knows where to look.
We have added in "(US-SRM, US-SRG, US-Whs, US-Wkg)" at the end of the sentence.

P14.L445-448. Where can I see this? Please make sure you back up your conclusions by showing the evidence.
The reviewer is right - we meant to add "(data not shown)" because the paper is already very dense and this is a relatively small test by comparison. But we agree that this should be shown and so we have added a figure to the supplementary material (new Fig S5). We also included the MOD10A snow cover product results for this site as per a suggestion from the reviewer (see above). Please note that we have also added further detail and nuance to the description of the forest site results related to snow as per a comment from Reviewer 1 - these can be found in the response to Reviewer 1.

[Figure]

P15.L460-480. I was a bit confused by the term evaporation E, whereas you also discuss evapotranspiration ET (which are often used interchangeably), but you mean here interception evaporation, correct? For clarity it might be good to add a subscript Ei and talk about interception evaporation.

We apologize, when we described what we include in ET in the original lines 182 to 184, we put the "E" in the wrong place (incorrectly placing it next to "evaporation from water intercepted by the canopy" instead of "bare soil evaporation". Those lines have been modified and now read: "Evapotranspiration, ET, in the model is calculated as the sum of

four components: 1) evaporation from bare soil, E; 2) evaporation from water intercepted by the canopy; 3) transpiration, T, (controlled by stomatal conductance); and 4) snow sublimation (Guimberteau et al., 2012b)."

We hope this is now clear, because when talking about plants we necessarily need to talk about plant transpiration, and therefore ET is not to be confused with E, which just refers to bare soil evaporation (at least, this is how we refer to it in LSMs that fully couple hydrology and biogeochemistry).

We have also made changes to the abstract to be clear as to what E and T refer to.

P15.L467. You mention before that US-Vcp underestimated ET, instead of overestimated.

Thank you for spotting this mistake. We had already spotted the incorrect inclusion of US-Vcp here and have removed it.

P16.L480. Are be responsible –> are responsible?

Changed - thank you.

P17.L517. You do not show that T/ET fractions are better with the reduced bare soil fraction.

The reviewer is right that we haven't shown these in Fig. 7 and that this statement is too broad and imprecise. We have now added the T/ET data-derived estimates into Fig. 7 and updated the caption. Given we now also have a more nuanced description of the use of the T/ET estimates in evaluating the model (described above), we have also further modified this sentence and included additional sentences to i) emphasize that we are talking about mean changes across all the sites; ii) to highlight differences between the spring and summer months; and most importantly, iii) to give further weight to the suggestion that the main issue might be more related to the model's ability to capture the seasonal changes in leaf area/vegetation cover (as opposed to just the amount of vegetation that is present throughout the year):

"However, although the T/ET ratios reduced the negative model biases compared to the data-derived estimates in the summer monsoon period, the model now overestimates ET (Figs. 7 and S8). However, while the decrease of the bare soil fraction (increase in C4 grasses) may have partially accounted for the negative bias in T/ET ratios at the start of the monsoon, the changes did not correct the phase discrepancy between the estimated and modelled T/ET seasonal trajectories: the estimated T/ET still declines to a minimum in June (as expected during the hot, dry period), whereas the model declines one month later. Furthermore, the spring ET model-data bias is further exacerbated by the increase in bare soil fraction and the mean spring estimated T/ET ratios and ET are a closer match to the original 11LAY version (Figs. 7 and S8). Putting the latter two points together, this new analysis gives further weight to the suggestion put forward in Section 3.3. that the model is not capturing the correct increase in leaf area at the start of the monsoon – not just that there is a lack in the overall amount of transpiring leaf area. Thus, there is potentially more of a problem with the model phenology schemes and/or the model's ability to capture dynamic changes in  seasonal vegetation cover than there is with the prescribed fractional vegetation cover. We discuss these issues more in Section 4."

P17.L523. TeNE-forest?
We have removed the forest.

P17.L529. Spring → spring
Changed - thank you.

P19.L592. ORCHIEE → ORCHIDEE
Changed - thank you.

P21.L669. I am not sure how you can conclude this without runoff data and never evaluating it.

We have proposed a change to this sentence - please see the reviewer's general comments above.

Table3. Please note that RMSE also has a unit
Changed - thank you.

Figure 3. The unit is mmm-1, I believe you mean mm/month, but please make this clearer.
We have added "Units are mm per month (mmm$^{-1}$)" to the caption and all other figures captions that have the same issue.

Figure 6. Why not include also the 2LAY-estimates? There are two methods used to estimate the ratios for the high and low elevation sites, is this a fair comparison then? Why is there no data for the first months? Why no data for US-Vcp?
We have changed this plot according to the discussion above in the reviewer's main comments, including adding the two methods and the addition of Vcp. We have also replied above about why there are no data for the first (winter) months. The 2LAY estimates are not included because this plot is referred to in the section in which we are describing remaining discrepancies between the 11LAY and the observations (and *not* the differences between the 2 and 11LAY); therefore, we only plot the 11LAY here for clarity in describing the results and to not have too many curves to distinguish between. The comparison between the 2LAY and 11LAY T/ET estimates are shown in Fig. S3.

Figure 7. Why would you average over all the sites? This is just removing information, please show all sites individually, there is no point in lumping this together.
This was a collective decision on the part of the co-authors to show a summary here and then show each individual site in the supplementary (Fig. S8) to avoid an excessive number of figures in the main manuscript. We would like to stick with this decision.

Figures S5 and S6. Please add units and a legend. And as these are regressions, why are there no data points shown? I only see a regression line, so I am not sure how to interpret these figures.
We assume the reviewer is talking about Figures S5 and S7, not S6? We have added units - thank you to the reviewer for pointing these out. However, in the original figures the data points were there - not just the regression lines, so we're not sure why these weren't

showing in the documents the reviewer received. We are not sure what the reviewer means by a legend as only one dataset is shown? We already have titles for each of the subplots (which correspond to each site).

Data availability:
Where are the model results shared?
The model results will be shared on my GitHub Page: https://github.com/nmacbean. We will wait to set up a specific repository for the article and to upload the simulations to the repository with a readme file (and eventually, the published paper) until the revisions have been accepted (in case we need to do more work on the paper).

---

## Author Comment (AC3) · 9 Apr 2020

I would like to thank the authors for their openness and the discussion, below I tried to reply to their questions in the informal response.

Thank you very much for considering our informal response and getting back to us so quickly. These additional comments helped to clarify our edits to the manuscript and our responses to the formal reviews.

I can see the difficulties the authors raise with regard to comparing the 2LAY and 11LAY soil moisture values, and also understand why the soil moisture values are not compared to observations for the 2LAY-model. For me, it is not a problem that you cannot use the 2LAY-values, but I just wonder what the point is of comparing 11LAY- results with soil moisture if you cannot do the same for the 2LAY-model. This also depends on the goal of the comparison, because you cannot use it to assess whichof the models is better (which I believe is the main goal of the paper, and also how I interpreted this section). I believe it could serve as an explanation why the ET-values are better, but some textual changes may be needed to clarify this. In the current version, this comparison seems rather important, and relates to some conclusions, whereas it is merely an additional and supportive explanation for some other more important findings.

We agree with the reviewer and in fact we did originally have a comparison to the 2LAY moisture but removed it for reasons we detail in the response to their formal (original) review. We have now proposed adding in a comparison to soil moisture observations to Figure 2, which compares the 2 vs 11LAY model. Please see our full response and updated figure in the response to the formal review.

Regarding the second point of the authors, and I am sorry for not making it easier, but I strongly disagree with reviewer 1 that you should remove the 2-layer versus the 11- layer comparison. This is for me the key-point of the manuscript, and this relates also to my comment in my review that the authors sometimes show already a preference for the 11-layer model. It is not carved in stone that a more detailed model is better, and it should objectively be assessed which one is better. Even though reviewer 1 points out that more detailed Richards' equation approaches often improve LSMs, there is also an important reason bucket-type models are still often used especially in catchment hydrology. The Richards' equation approach does not include macro-pores, which in more sloped areas plays an important role. In addition, the parameterization often as- sumes a homogene soil, which is also not true. The fact that LSMs often perform better with Richards' approach also relates to how they are parameterized, bucket-type mod- els need actually calibration as the parameters are less physically based, whereas the Richards' approach uses more physically

based soil parameters that are often mea- sured. In general, the hydrological schematization in LSMs is in my view still rather poor, even with more detailed Richards' equation approaches, whereas it actually has a strong influence on the outcomes of the models, so I believe it is important that the authors show this. In addition, for a strong modelling experiment, you always need a benchmark, which is here the 2-layer model. Leaving it out leads to a manuscript that is just a model application, and the reader can never see what the 11-layers actually add.

We thank the reviewer for outlining further reasoning for keeping the 2 vs 11 layer comparison. To address both reviewers concerns/suggestions on this matter, we propose outlining our reasoning for this comparison more clearly by including the following statement in the introduction (after original lines 120-122):

"Although there have been many previous studies comparing simple bucket schemes versus mechanistic multi-layer hydrology based on the Richards equation, we include such a comparison in the first part of our analysis for the following reasons: a) the simple bucket schemes were the default hydrology in some CMIP5 model simulations and these simulations are still being widely used to understand ecosystem responses to changes in climate; b) variations on the simple bucket schemes are still implemented by design in various types of hydrological models (Bierkens et al., 2015); c) there has not yet been extensive comparisons of these two types of hydrology model for semi-arid regions, and especially not for the SW US; and d) so that the 2LAY can serve as a benchmark for the 11LAY scheme."

Bierkens, M. F. P.: Global hydrology 2015: State, trends, and directions, Water Resources Research, 51(7), 4923–4947, doi:10.1002/2015wr017173, 2015.

We do completely agree that it is not necessarily the case that a more complex model is needed. We hope that by addressing the reviewer's original comments and suggestions (including adding soil moisture observations to figure 2) that we have made our case for why we think the 11LAY does a better job at capturing the temporal dynamics of the upper layer (root zone) soil moisture and evapotranspiration. We have tried explicitly not to go beyond that specific conclusion regarding any preference for the 11 layer. We also hope that it is clear from our analyses on remaining model discrepancies, as well as other topics that we have highlighted in the discussion, that we agree that there are still many issues (missing or inadequately represented processes) in the more mechanistic versions included in LSMs that still need to be addressed. For example we have mentioned, as the reviewer discussed above, the fact that soil texture and most hydraulic parameters are fixed both vertically in the soil column and that spatial heterogeneity is not well captured. We also mention the need for parameter calibration. We did not include an exhaustive list of all the LSM hydrology model issues simply because these comparisons are still point-based, whereas many of the issues that remain are related to modeling spatially distributed hydrological budgets, which is beyond the scope of our present study.

I hope my thoughts are useful, even though it is probably not making it easier. I still look forward to an improved manuscript and hope the authors find a good way to address all the issues of myself and reviewer 1.

We sincerely thank the reviewer for their second round of feedback. It was certainly both insightful and useful in making a decision on this issue.

---

## Referee Report (RR1)

Review of **Testing water fluxes and storage from two hydrology configurations within the ORCHIDEE land surface model across US semi-arid sites** by *MacBean et al.*

The new version of the manuscript by MacBean et al., who compares two different soil schematizations for the ORCHIDEE land surface model, shows many improvements compared to the initial submission. I am happy that my comments were proven useful, and noticed the authors addressed most of them adequately. I also appreciated the open attitude of the authors and the scientific discussion.

I think the manuscript is making a better comparison now between the two model set-ups. The changes in Figure 2 and the additional performance measures help a lot here. I still think the authors could focus the manuscript a bit more, as often a discussion of the results is already included in the Results section, whereas an actual Discussion section is present in the paper. This is however a minor comment, and hope the authors just go over the manuscript once more to sharpen these details a bit. I also still have quite some minor issues, and think the authors should address these as well before the paper is published.

Regarding some specific questions of the authors, a discussion on soil texture and depth would definitely be appreciated by me, but I am also already quite happy with the additional discussion on the need for groundwater. I would also appreciate a short and condensed version of the text in the author's response on soil texture and depth, maybe just a few lines, but I leave this up to the authors. The statements on runoff in the conclusions are also much more moderate now and relate more to previous discussions in the manuscript.

I hope the authors find these comments useful again, and look forward to a final version of the manuscript.

**Minor comments**
P4-5.L112-159. This paragraphs contain quite some details, and would probably fit better in the Methods section.
P9.L269. In the introduction bare soil evaporation was introduced as E, here it is Eg, later in the manuscript it is E again, and in some figures BS evap, so please check consistency throughout the manuscript.
P11.L341. Please define GPP.VPD 0.5
P13.L393. Reduced bare soil fraction → maybe also good to mention how much you reduce this.
P13.L419-421. This sentence seems a bit odd to me.
P13.L425. The low-elevation shrub and grass sites → here you mean the remaining four correct? It is maybe helpful to repeat the site names here.
P14.L439. Lower drainage flux → I think you should refer to lower total runoff in general, as the surface runoff is higher.
P14.L445.onl → only
P14.L448. How is it possible to have such a variation in the upper soil moisture during dry periods?
P14.L450. 3 → three
P14.L450. In much better fit → in a much better fit
P16.L500-501. Where do I exactly see this? The second panel of US-SRM and SRG?
P16.L516. I do not see a light green zone, do you mean light blue? It is also not very visible in the figure.
P17.L543-545. As the forests...temporal variability. → I do not follow this reasoning completely. When the hydraulic conductivity is higher due to roots and preferential flows, you would expect a higher temporal variability, not lower.

P18.L561. 11LAY observed ET →do you mean modelled?

P19.L591. spring mean LAI → is this observed LAI?

P20.L632-633. This decrease...at all sites (Fig. 7) → Figure 7 only shows that this happens for all sites averaged together, but not for each site individually. So I do not think you can say this based on Figure 7.

P20.L636-637. The mean estimated...11LAY version (Figs.7) →Figure 7 is not about all sites, correct?

P20-21.L650-651. Why would there be a high sensitivity for transpiration when the bare soil resistance is changed and there is hardly a bare soil fraction? The bare soil evaporation is also rather similar in Figure 8, but I would expect that that would change instead of the transpiration.

P21.L651.TeNE PFT T → please write this out, it is not very clear like this.

P27.L856. Discrepancies semi-arid → discrepancies of semi-arid?

P27.L858. Timing transpiring → timing of transpiring?

Fig3. Please add a space between mmm-1, and write out month, because even though explained in the caption, it is not a correct unit like this and a bit confusing. See also the HESS-recommendations on units: https://www.hydrology-and-earth-system-sciences.net/for_authors/manuscript_preparation.html

Fig5. The lightblue parts are not very visible, maybe use a different color for the marking. Please add this also in the caption.

Fig7. The authors commented they wanted to keep the figure as it is, but averaging results like this just does not make any sense. In this way, results can actually look pretty good "on average", but two big overestimations can completely compensate for two big underestimations, and results that are actually rubbish can be accepted as good. Obviously, this is not the case here, and fortunately I can check this in the supplement, but I think one should be careful by showing results in this way as it can be perceived as misleading. I think it is better to show an example instead, or create envelopes with the full range of results.

Fig.S5. The caption is not correct.

---

## Author Response (AR2)

Dear Natasha MacBean and co authors,

your manuscript was sent out to a second round of reviews and the assessment was generally positive, except for one point. Please attend to the issue raised by Reviewer #2 on the explanation of the tiling procedure. The reviewer has explained the issue really well and I agree. Please also have another look through the manuscript and try to shorten it wherever possible without compromising the content. Given those minor changes the manuscript is ready to be published in HESS. Please submit a response together with a manuscript with highlighted changes for a last assessment by the editor.

At this point, I would also like to thank the two reviewers for their contribution to improving the manuscript.

I am looking forward to the final version!
Best regards,
Anke Hildebrandt

Dear Professor Hildebrandt,

Thank you very much for your response and guidance for final changes before acceptance. We are grateful to both reviewers for their insightful reviews. They have undoubtedly resulted in an improvement to this manuscript.

We have followed yours and Reviewer #2's suggestions on the tiling and modified the text in Section 2.2.1 accordingly. This includes re-structuring that section so the description of the soil tiling comes immediately after the description of PFT fractions to make both more comprehensible to the reader and adding in additional text for clarification. We hope that the description of how the soil tiling works in the model is now much clearer.

We have also edit and shortened the text where possible. These edits included some suggestions from the 1st review. Please see the edits in the tracked change version. And finally, we have addressed all the minor comments of the 1st review. We have provided brief responses to the reviewer comments.

We hope that this revised manuscript has adequately addressed all comments. We would be delighted to see the study published in HESS!

Best wishes,
Natasha

Review of **Testing water fluxes and storage from two hydrology configurations within the ORCHIDEE land surface model across US semi-arid sites** by *MacBean et al.*

The new version of the manuscript by MacBean et al., who compares two different soil schematizations for the ORCHIDEE land surface model, shows many improvements compared to the initial submission. I am happy that my comments were proven useful, and noticed the authors addressed most of them adequately. I also appreciated the open attitude of the authors and the scientific discussion.

I think the manuscript is making a better comparison now between the two model set-ups. The changes in Figure 2 and the additional performance measures help a lot here. I still think the authors could focus the manuscript a bit more, as often a discussion of the results is already included in the Results section, whereas an actual Discussion section is present in the paper. This is however a minor comment, and hope the authors just go over the manuscript once more to sharpen these details a bit. I also still have quite some minor issues, and think the authors should address these as well before the paper is published.

→ We are very grateful to the reviewer for re-reviewing our study and we thank them for pointing out further issues and inconsistencies. Their reviews have been very valuable and have undoubtedly resulted in a strong improvement to the manuscript. We are glad that they are pleased with the revised manuscript. We have addressed their point on soil texture and the other minor comments listed below. These changes can be seen in the revised tracked changes word document.

Regarding some specific questions of the authors, a discussion on soil texture and depth would definitely be appreciated by me, but I am also already quite happy with the additional discussion on the need for groundwater. I would also appreciate a short and condensed version of the text in the author's response on soil texture and depth, maybe just a few lines, but I leave this up to the authors. The statements on runoff in the conclusions are also much more moderate now and relate more to previous discussions in the manuscript.

→ We have added 1 sentence to the end of section 3.2 where we discuss changes in soil texture with depth.

I hope the authors find these comments useful again, and look forward to a final version of the manuscript.

**Minor comments**

P4-5.L112-159. This paragraphs contain quite some details, and would probably fit better in the Methods section.

→ We agree and have put some very detailed parts of this section in the methods.

P9.L269. In the introduction bare soil evaporation was introduced as E, here it is Eg, later in the manuscript it is E again, and in some figures BS evap, so please check consistency throughout the manuscript.

P11.L341. Please define GPP.VPD 0.5

P13.L393. Reduced bare soil fraction → maybe also good to mention how much you reduce this.

P13.L419-421. This sentence seems a bit odd to me.

P13.L425. The low-elevation shrub and grass sites → here you mean the remaining four correct? It is maybe helpful to repeat the site names here.

P14.L439. Lower drainage flux → I think you should refer to lower total runoff in general, as the surface runoff is higher.

P14.L445.onl → only

P14.L448. How is it possible to have such a variation in the upper soil moisture during dry periods?

P14.L450. 3 → three

P14.L450. In much better fit → in a much better fit

P16.L500-501. Where do I exactly see this? The second panel of US-SRM and SRG?

P16.L516. I do not see a light green zone, do you mean light blue? It is also not very visible in the figure.

P17.L543-545. As the forests...temporal variability. → I do not follow this reasoning completely. When the hydraulic conductivity is higher due to roots and preferential flows, you would expect a higher temporal variability, not lower.

P18.L561. 11LAY observed ET →do you mean modelled?

P19.L591. spring mean LAI → is this observed LAI?

P20.L632-633. This decrease...at all sites (Fig. 7) → Figure 7 only shows that this happens for all sites averaged together, but not for each site individually. So I do not think you can say this based on Figure 7.

P20.L636-637. The mean estimated...11LAY version (Figs.7) →Figure 7 is not about all sites, correct?

P20-21.L650-651. Why would there be a high sensitivity for transpiration when the bare soil resistance is changed and there is hardly a bare soil fraction? The bare soil evaporation is also rather similar in Figure 8, but I would expect that that would change instead of the transpiration.

P21.L651.TeNE PFT T → please write this out, it is not very clear like this.

P27.L856. Discrepancies semi-arid → discrepancies of semi-arid?

P27.L858. Timing transpiring → timing of transpiring?

Fig3. Please add a space between mmm-1, and write out month, because even though explained in the caption, it is not a correct unit like this and a bit confusing. See also the HESS-recommendations on units: https://www.hydrology-and-earth-system-sciences.net/for_authors/manuscript_preparation.html

Fig5. The lightblue parts are not very visible, maybe use a different color for the marking. Please add this also in the caption.

Fig7. The authors commented they wanted to keep the figure as it is, but averaging results like this just does not make any sense. In this way, results can actually look pretty good "on average", but two big overestimations can completely compensate for two big underestimations, and results that are actually rubbish can be accepted as good. Obviously, this is not the case here, and fortunately I can check this in the supplement, but I think one should be careful by showing results in this way as it can be perceived as misleading. I think it is better to show an example instead, or create envelopes with the full range of results.

Fig.S5. The caption is not correct.

2nd round: Response to Report #2 (Referee #1)

Comments on revision of MacBean et al. for HESS.

The authors surely answered to all the questions raised by the reviewers. I have to admit that I was a bit overwhelmed (and a bit annoyed) with the amount of text: 58 pages rebuttal letter vs 27 pages of paper text. Sorry if I could not gather every little detail in the paper and in the letter.

So the only thing that I would still like to be changed is the explanation of the tiling in the 11-layer version (see below). I find it essential to understand the implications of the manuscript but the model structure is not clear to me from the text. While the text explains well the vertical structure within the soil column, it is not explaining well the tiling, the different vegetated and non-vegetated fractions. There are so many fractions that I got lost. Perhaps the word "sub-fraction" could be used sometimes.

l. 246ff (of the track changes version): sorry I could not understand the logic even that there is a formula. What are soil tiles? And what does it mean that all PFTs are gathered in one tile? Are they sharing the same soil? Are they all tapping into the same soil? How do you check that you do not take too much water from it?

→ We thank the reviewer for their review and these useful comments. We agree with the reviewer that the tiling description could have been clearer. We have now restructured to section 2.2.1 so the description of soil tiles comes directly after the description of PFTs in the hope that this makes the description of what is going on easier to understand. We have also added a couple of sentences to clarify these points.

I have understood: one grid cell, say 10 km x10 km, is divided into fractions with different PFTs, say 2 PFTs, Temperate broadleaf deciduous tree ($f\_v1$) and C4 grass ($f\_v2$). Is there always another fraction Bare soil ($f\_b$) so that $f\_v1+f\_v2+f\_b=1$ or is it simply $f\_v1+f\_v2=1$? Each PFT fraction has its own bare soil sub-fraction ($f\_b1$ and $f\_b2$) calculated with a fraction of vegetation cover, often denoted $f\_c$. I would not call this not a modified Beer-Lambert equation because $k\_ext$ is not an extinction coefficient. $f\_c$ is rather a gap probability taking into account only between-trees gap probability (and not within-tree gap probability). I would then write directly $f\_cj = 1-exp(-LAI\_j)$ or $fb\_j = f\_vj * exp(-LAI\_j)$. Is there then only one soil underneath the vegetated and bare soil part of this PFT fraction? Are they acting on the same soil water? Or are there two soils, one for vegetated and one for bare soil? If so, how do you preserve water if LAI changes?

→ We appreciate these points weren't clear and we have revised the text accordingly (in addition to the changes mentioned above - please see the track changes in Section 2.2.1). We hope it is much clearer now. We've also referred to "sub-fractions" as suggested by the reviewer above. We haven't changed the notation because the same notation will be used in the upcoming paper by Ducharne et al that will document the latest hydrology developments; therefore, we use the same for consistency.

So I find the formulation in the paper misleading that there is E and T from a vegetated surface. It is rather how the bare soil fraction is defined, on which only E happens whereas only T happens over the effectively vegetated surface. The difference is indeed the energy input to the surface. Bare soil gets direct sunlight and will have hence much higher surface temperatures than the soil below a canopy. So in this respect, it did not change between the 2- and 11-layer versions. It is rather how the bare soil fraction is defined that changed.

→ It is not that E and T are from a vegetated *surface*; rather, both can occur from a vegetated soil *tile*. As we hope is more clear with the revised version - three separate water budgets are calculated based on the three soil tiles (tiles represent soil columns). Within those soil tiles, T is occurring for each PFT based on its LAI and E is happening only when there's low LAI (e.g. in Winter). We appreciate this gets complex and it is difficult to explain; however, we have tried to keep with the terminology used in the ORCHIDEE model and therefore in other ORCHIDEE papers (and in the forthcoming Ducharne paper that describes all of the latest hydrology model in detail). We hope that the modifications we have made, together with the use of "sub-fraction" for describing the effective vegetated and non-vegetation (sub-)fractions of the vegetated soil *tile*, will be much clearer now.

But this might be wrong because you might use only one surface temperature per PFT fraction. Then what is calculated is actually not a bare soil fraction but a below-canopy E assuming a direct connection between above- and below canopy, i.e. no canopy air space.

→ We see the reviewer's point that the E from the effective bare soil over the vegetated tile could be seen this way; however, it is not exactly a below-canopy E because we do not resolve layers in the canopy, so we prefer not to describe it as such. We have added this in the section on the bare soil evaporation resistance term in section 2.2.3.

As you might have understood by now, I could really not understand how all the tiling is done and the real assumptions in the 11-layer version. I looked into the reference of Guimberteau et al. (2014) but that did not help. I did not look into all the Rosnay papers. And I think that the water fluxes cannot be understood without explaining how heat is calculated. I find it essential, though, to understand tiling and the relation between en

[revised manuscript text omitted]

**a) US-Fuf**

[Figure]

45  **b) US-Vcp**

[Figure]

**c) US-SRM**

[Figure]

**d) US-Whs**

[Figure]

**e) US-SRG**

[Figure]

**f) US-Wkg**

[Figure]

110

115

120

125 **Figure S2: Complete daily time series of upper layer soil moisture, surface water fluxes and related variables between the 2LAY (green curve) and 11LAY (blue curve) simulations for all sites – equivalent to Fig. 2. At each site, top panel: LAI; 2nd panel: ET compared to observations (black curve); 3rd panel: bare soil evaporation; 4th panel: transpiration; 5th panel: empirical water limitation function ($\beta$) that scales photosynthesis and stomatal conductance; bottom panel: model soil moisture (re-scaled via linear CDF matching) expressed as volumetric soil water content (SWC) in the uppermost 10cm of the soil compared to observations (black**
130 **curve). Precipitation is shown in the grey bars in the bottom panel for each site. Sites in following order: a) US-Fuf; b) US-Vcp; c) US-SRM; d) US-Whs; e) US-SRG; f) US-Wkg. Precipitation is shown in the grey lines in the bottom panel for each site.**

**a) US-Fuf**

[Figure]

[Figure]

135

140

**b) US-Vcp**

[Figure]

[Figure]

160  **c) US-SRM**

[Figure]

[Figure]

 **d) US-Whs**

[Figure]

[Figure]

[Figure]

180

185

**e) US-SRG**

[Figure]

[Figure]

195

200

**f) US-Wkg**

[Figure]

[Figure]

210

215

**Figure S3: Monthly mean seasonal cycle for each site comparing the 2LAY (green curve) and 11LAY simulations (blue curve) with observations (black curve). Top left: ET; top right: T/ET ratios; bottom left: transpiration, T; bottom right: bare soil evaporation, E. Units in mmd⁻¹. Sites in following order: a) US-Fuf; b) US-Vcp; c) US-SRM; d) US-Whs; e) US-SRG; f) US-Wkg. Units are mm per month (mm month⁻¹).**

225      **a)     US-Fuf**

[Figure]

[Figure]

230

235

¶
¶
¶

**b) US-Vcp**

[Figure]

[Figure]

245

250

255

**c)   US-SRM**

[Figure]

[Figure]

265

270

275

**d)  US-Whs**

[Figure]

[Figure]

295    **e) US-SRG**

[Figure]

[Figure]

300

305

310

**f) US-Wkg**

[Figure]

[Figure]

315

320

325

¶
¶

**Figure S4: Daily simulated volumetric soil water content (SWC – m³m⁻³) across all site years (re-scaled via linear CDF matching)**
**compared to observations at each site for three depths (upper, middle, lower) in the soil profile – equivalent to Fig. 4. The soil depths**
335 **and their corresponding model layers are given in Table 3. Precipitation is shown in the grey lines in the bottom panel for each site.**

a) **US-Fuf**

[Figure]

340

345

**b)  US-Vcp**

[Figure]

**c) US-SRM**

[Figure]

**d)  US-Whs**

[Figure]

400

405

410

415 **f) US-Wkg**

[Figure]

**Figure S5:** Comparison of US-Fuf daily simulated soil water content (SWC – bottom panel – $m^3m^{-3}$) across all site years (re-scaled via linear CDF matching) and snow mass (middle panel) for the original 11LAY model version (blue curve) and a repeat simulation with snowfall forcing multiplied by a factor of 10 (red dashed curve) compared to observations (black curve). MODIS snow cover (%) observations (MOD10A1 v6 – downloaded from the National Snow and Ice Data Center: https://nsidc.org/) are shown in the top panel to ilustrate that the model may simulate a too rapid melting of snow (too short snowpack duration).

[Figure]

**Figure S6: Linear regressions between spring (March-April) mean monthly LAI (m²m⁻²) and spring mean monthly ET (mm month⁻¹) model-data misfits for each site. The dominant PFT is given in brackets for each site. See Table 1 for PFT acronyms.**

[Figure]

455

460

465

**Figure S7: Plots comparing ET and LAI for C4 grasses (C4G) and mesquite shrubs (Temperate Broadleaved Deciduous – TeBD – PFT in ORCHIDEE) monthly mean seasonal cycles at US-SRM for the 2LAY (green curve) and 11LAY (blue curve) model versions in comparison to observations (black curve).**

[Figure]

**Figure S8: Linear regressions between monsoon (July-September) mean monthly LAI (m²m⁻²) and monsoon mean monthly ET (mm month⁻¹) model-data misfits for each site. The dominant PFT is given in brackets for each site. See Table 1 for PFT acronyms.**

[Figure]

[Figure]

[Figure]

510

515

520 **Figure S10: Monthly mean seasonal cycle for all sites comparing the default 11LAY simulations (blue curve) with a simulation that included an additional bare soil evaporation resistance term (red curve). ET is compared to observations (black curve). In all subfigures – top left: ET; top right: T/ET ratios; middle left: transpiration, T; middle right: bare soil evaporation, E; bottom left: mean column soil water content, SWC; bottom right: total leaf area index, LAI. Units are mm per month (mm month⁻¹).**

a) **US-Fuf**

[Figure]

525

530

535

**b) US-Vcp**

[Figure]

[Figure]

545

550

555

560    **c)  US-SRM**

[Figure]

[Figure]

565

570

575

**d) US-Whs**

[Figure]

[Figure]

**e) US-SRG**

[Figure]

[Figure]

**f) US-Wkg**

[Figure]

[Figure]

615

620

625

**Figure S11: Monthly mean seasonal cycle for all** low elevation grass and shrub **sites comparing the default 11LAY simulations (blue curve)**, **with a simulation that** decreased the bare soil (BS) fraction and **increased the C4 grass fraction** (yellow curve)**, and a simulation that** decreased the BS fraction as well as includ**ing** **an additional bare soil evaporation resistance term (red curve). In all subfigures – top left: mean** column volumetric **soil moisture** content (SWC)**; top right:** modelled versus observed (black curve) evapotranspiration, **ET**; **bottom left: transpiration**, **T**; **and** **bottom right: bare soil evaporation**, **E. Units are mm per month (mm m**onth**⁻¹).**

[Figure]

[Figure]

---

## Author Response (AR3)

3rd round response to editor

Dear authors,
thank you for the new version of the manuscript. It is almost ready for publication. I have only two minor points.

Some of the responses to the comments by Reviewer #1 are missing in the letter. It seems most comments were addressed, but a couple are still open. Could you please respond to those below (see below, pages and line numbers refer to the previous manuscript version 4).

Also, be aware that the shades in Fig 5 do not show up in the pdf. Could you please have an eye on this in the copy-editing process?

I am looking forward to the final version of your manuscript. Please send along with it a short note, where you incorporated changes or why you decided not the change the manuscript.

Sincerely,
Anke Hildebrandt

Dear Professor Hildebrandt,

Thank you very much for your response and for pointing out these errors and omissions. I'm sorry I should have provided a point by point response last time to explain what we had done.

I have corrected the two points in the revised manuscript (and please see the response below).

Also my apologies for not checking the shading in the final pdf I submitted. I believe the shading disappeared because of the way I saved the word doc to pdf. I have saved it differently now and the shading appears, but I will check this in the copy-editing process as you suggest.

Thank you once again for your careful review. We greatly appreciate it.

Best wishes,
Natasha MacBean (on behalf of all the co-authors)

(1)
P20.L632-633. This decrease...at all sites (Fig. 7) → Figure 7 only shows that this happens for all sites averaged together, but not for each site individually. So I do not think you can say this based on Figure 7.
P20.L636-637. The mean estimated...11LAY version (Figs.7) →Figure 7 is not about all sites, correct?

(The Figures 7 and S9 are indeed on „all low-elevation grass- and shrub-dominated sites“, not all sites, please adapt the sentence to reflect this)

Thank you for pointing this out. Yes that is correct. We had added "low elevation grass and shrub sites" to the second sentence you mention above (in v5 L652-653) but yes my apologies we had forgotten to amend the first sentence. That now reads "This decrease in bare soil fraction increased ET and T/ET ratios during the monsoon period at all *low elevation grass- and shrub-dominated sites* and also increased ET during spring at the Santa Rita sites (Fig S9; mean across low elevation sites in Fig. 7)"

(2)
P20-21.L650-651. Why would there be a high sensitivity for transpiration when the bare soil resistance is changed and there is hardly a bare soil fraction? The bare soil evaporation is also rather similar in Figure 8, but I would expect that that would change instead of the transpiration.

I agree with the reviewer here and I also was confused by this to begin with when I first saw these results, so I agree that in our original revised version (v4) we should have explained this bettter. We did attempt to explain this in our re-revised version by changing this sentence (v4):
"The lack of bare soil at the high-elevation forested sites resulted in a higher sensitivity of the TeNE PFT T to the addition of the bare soil evaporation resistance term (Fig. 8 – left column)."

to the following in v5:

"The lack of bare soil at the high-elevation forested sites *(therefore, E only occurs over the bare soil sub-fraction of the vegetated soil tiles – see section 2.2.1)* resulted in a higher sensitivity of the Temperate Needleleaved Evergreen forest transpiration to the addition of the bare soil evaporation resistance term (Fig. 8 – left column). The reduction in E during the winter allowed for higher soil moisture content (Figs. S10 a and b) and therefore a greater T (and E) during the spring and summer."

But I can see that is still confusing and is not helped by being a lengthy sentence. Also, I think the word "sensitivity" here might be unnecessarily confusing. In our latest re-submitted version I have changed the above two sentences so this section now reads as follows:

[revised manuscript text omitted]

<table><tr><td>Deleted: T</td></tr><tr><td>Deleted: in</td></tr><tr><td>Deleted: this study</td></tr><tr><td>Formatted: Font: Not Italic</td></tr><tr><td>Deleted: g</td></tr><tr><td>Formatted: Font: Not Italic</td></tr><tr><td>Deleted: g</td></tr><tr><td>Formatted: Font: Not Italic</td></tr><tr><td>Deleted: g</td></tr><tr><td>Formatted: Font: Not Italic</td></tr><tr><td>Formatted: Font: Not Italic</td></tr><tr><td>Deleted: g</td></tr><tr><td>Formatted: Font: Not Italic</td></tr><tr><td>Deleted: g</td></tr><tr><td>Deleted: g</td></tr><tr><td>Formatted: Font: Not Italic</td></tr><tr><td>Formatted: Font: Not Italic</td></tr><tr><td>Deleted: g</td></tr><tr><td>Formatted: Font: Not Italic</td></tr><tr><td>Formatted: Font: Not Italic</td></tr></table>

[revised manuscript text omitted]
; 2$^{nd}$ panel: ET compared to observations (black curve); 3$^{rd}$ panel: bare soil evaporation; 4$^{th}$ panel: transpiration; 5$^{th}$ panel: empirical water limitation function ($\beta$) that scales photosynthesis and stomatal conductance; bottom panel: model soil moisture (re-scaled via linear CDF matching) expressed as volumetric soil water content (SWC) in the uppermost 10cm of the soil compared to observations (black curve). Precipitation is shown in the grey lines in the bottom panel for each site. (Note: full time series across all years are shown for all site in Figs. S2a-f). Light brown shaded zones show periods of maximum plant water limitation ($\beta$) at Santa Rita and consequent troughs in T and SWC.**

¶

[Figure]

[Figure]

**Figure 3: Evapotranspiration (ET) monthly mean seasonal cycle comparing the 2LAY (green curve) and 11LAY (blue curve) simulations with observations (black curve). Individual site simulations have been averaged over the high-elevation tree dominated sites (left panel) and across all the low-elevation grass- and shrub-dominated sites (right panel). Units are mm per month (mm month⁻¹).**

[Figure]

[Figure]

**Figure 4: Daily simulated volumetric soil water content (SWC – $m^3 m^{-3}$) in 2009 (re-scaled via linear CDF matching) compared to observations at each site for three depths (upper, middle, lower) in the soil profile. The soil depths and their corresponding model layers are given in Table 3. Precipitation is shown in the grey lines in the bottom panel for each site.**

[Figure]

**Figure 5: a) US-Fuf and b) US-Vcp 11LAY (blue curve) daily time series (2007-2010) of model (re-scaled via linear CDF matching) versus observed volumetric soil water content (middle panel SWC – m³m⁻³) (black curve), compared to simulated snow mass (top panel) and soil temperature from the corresponding 2cm soil thermal layer (bottom panel). Snowfall is also shown as grey lines in the SWC time series. In the bottom panel the grey horizontal dashed line shows 0°C threshold. Light blue shaded zones show periods where the model overestimates the observations; light green shaded zones show periods where the model underestimates the observations.**

a)   US-Fuf

[Figure]

[Figure]

US-Fuf

11 layer hydrol    SITE DATA

¶
¶

**b) US-Vcp**

[Figure]

[Figure]

**Figure 6: Comparison of modelled and data-derived estimates of mean monthly T/ET ratios for each site. Forest site (US-Fuf and US-Vcp) T/ET estimates are derived using the method of Zhou et al. (2016 – Z16 – green curve). Monsoon low-elevation grass- and shrub-dominated site T/ET estimated are based on both Zhou et al. (2016) and Scott and Biederman (2017 – SB17 – orange curve). Blue curves show the model ratios at each site. Please see Section 2.3.1 for details on methods for data-derived T/ET estimates.**

[Figure]

**Figure 7: Monthly mean seasonal cycle for ET, T/ET ratios, T and E averaged across all low-elevation grass- and shrub-dominated sites comparing the default 11LAY simulations (blue curve) with a simulation in which bare soil fraction is decreased (C4 grass cover increased (yellow curve). ET is compared to observations (black dashed curve) and T/ET ratios are compared to the data-derived estimates from Scott and Biederman (2017 – orange dashed curve) and Zhou et al. (2016 – green dashed curve). Units are mm per month (mm month⁻¹).**

[Figure]

[Figure]

**Figure 8: Monthly mean seasonal cycle for evapotranspiration (ET), transpiration, $T$, and bare soil evaporation, $E$, averaged across all high-elevation forest sites (left column) and low-elevation monsoon grass- and shrub-dominated sites (right column) for the default 11LAY simulations (blue curve) compared to a simulation that included an additional bare soil evaporation resistance term (red curve). ET is also compared to observations (black curve). Units are mm per month (mm month⁻¹).**

[Figure]

[Figure]